# Budget of the total nitrogen in the Yucatan Shelf: driving mechanisms through a physical-biogeochemical coupled model

Sheila N. Estrada-Allis[1], Julio Sheinbaum[1], Joao M. Azevedo Correia de Souza[2], Cecilia Enríquez Ortiz[4], Ismael Mariño Tapia[3], and Jorge A. Herrera-Silveira[3]

[1]Physical Oceanography Department, CICESE, Ensenada, Baja California, Mexico
[2]MetOcean Solutions / MetService. Raglan. New Zealand.
[3]Departamento Recursos del Mar, CINVESTAV, Merida, Yucatan, Mexico.
[4]ENES-Merida, Facultad de Ciencias, Campus Yucatan, UNAM, Mexico.

**Correspondence:** Sheila N. Estrada-Allis (sheila@cicese.mx)

**Abstract.** Continental shelves are the most productive areas in the seas with strongest implications for global Total Nitrogen (TN) cycling. The Yucatan shelf is the largest shelf in the Gulf of Mexico (GoM), however, its general TN budget has not been quantified. This is largely due to the lack of significant spatio-temporal *in situ* measurements and the complexity of the shelf dynamics, including coastal upwelling, Coastal-Trapped Waves (CTWs) and influence of the Yucatan Current (YC) via bottom Ekman transport and dynamic uplift. In this paper, the TN budget in the Yucatan shelf is quantified using a nine-year output from a coupled physical-biogeochemical model of the GoM. Results indicate that the main entrance of inorganic nitrogen is through its southern (continental) and eastern margins. The TN is then advected to the deep oligotrophic Bay of Campeche and central GoM. It is also shown that the inner shelf (bounded by the 50 m isobath) is "efficient" in terms of TN, since all the Dissolved Inorganic Nitrogen (DIN) imported into this shelf is consumed by the phytoplankton. Submarine groundwater discharges (SGDs) contribute 20% of the TN, while denitrification removes up to 53 % of TN that enters into the inner shelf. The high-frequency variability of the TN fluxes in the southern margin is modulated by fluxes from the YC due to enhanced bottom Ekman transport when the YC leans against the shelf-break (250 m isobath) on the eastern margin. This current-topography interaction can help to maintain the upwelling of Cape Catoche, uplifting nutrient-rich water into the euphotic layer. The export of TN at both western and northwestern margins is modulated by CTWs with a mean period of about 10 days in agreement with recent observational and modelling studies.

## 1 Introduction

Continental shelves are the most productive areas in the ocean, widely recognized to play a critical role in the global cycling of nitrogen and carbon (e.g., Fennel, 2010; Liu et al., 2010) with direct implications for human activities, such as fisheries, tourism, and marine resources (Zhang et al., 2019).

The importance of nitrogen budgets in shelves has motivated numerous observational and modelling studies of different shelves in the world (e.g., Fennel et al., 2006; Xue et al., 2013; Ding et al., 2019; Zhang et al., 2019). Their significance lies in that nutrient supply fuels primary productivity which in turn impacts the socio-economical and recreational activities in those

regions. Furthermore, the exchange of nitrogen between the shelf and the deep ocean influences the carbon cycle (Huthnance, 1995), and it is strongly correlated with other shelf processes such as: acidification, eutrophication, red tides, hypoxia/anoxia zones, pCO2 and sediment denitrification (Fennel et al., 2006; Seitzinger et al., 2006; Enriquez et al., 2010).

In the GoM, (Figure 1a), with a horizontal extension of almost 250 km, the Yucatan Shelf (YS) (Figure 1b and c) is one of the largest shelves in the world. It has 340 km of littoral extension, representing 3.1% of Mexico's littoral zone. The Yucatan state in Mexico occupies the 12th place in volume catches and the 6th place in production value of fisheries in the country. The fishery production is increasing every year with a growth of 72% from 2008 to 2017 (Anuario de Pesca 2017, 2017).

Nutrient fluxes are intrinsically related with the productivity and nitrogen cycling of the shelves. However, sources and sinks of nutrients are highly uncertain and difficult to quantify. This is partly due to the large spatial and temporal variability associated with the cross-shelf and along-shelf regional nutrient budgets and the difficulty to measure them. Biogeochemical coupled modeling systems are a useful tool to quantify the shelf-open ocean nutrient exchange, taking into account the different spatial and temporal scales involved in the biogeochemical cycle (Walsh et al., 1989; Fennel et al., 2006; Hermann et al., 2009; Xue et al., 2013; Damien et al., 2018; Zhang et al., 2019).

The physical mechanisms that drive and modulate the cross-shelf transport of nutrients and biogenic material are also poorly known. Shelves are rich dynamical areas in which several processes can coexist at different spatio-temporal scales. Ekman divergence, CTWs, current interactions with the shelf break, mesoscale structures, vertical mixing and topographic interactions, among others, are processes that may uplift nutrient-rich waters from the deep ocean into the photic zone of continental shelves (e.g., Cochrane, 1966; Merino, 1997; Roughan and Middleton, 2002, 2004; Hermann et al., 2009; Shaeffer et al., 2014; Jouanno et al., 2018).

In this regard, the YS is a complex system due to the coexistence of different physical processes relevant in its dynamics. One of the first studies in the area is that of Merino (1997) who reported the uplift of nutrient-rich Caribbean waters from 220-250 m deep, reaching the YS at the "notch area" (small yellow box in Figure 1), likely due to the interaction of the Yucatan Current (YC) with the YS. The zonal Caribbean Current of the Cayman Sea turns northwards when reaching the Yucatan Peninsula forming the strong western boundary YC that flows through the Yucatan Channel, located between the eastern slope of the YS and Northwestern Cuba (see yellow line in Figure 1a). Once inside the GoM, the YC becomes the Loop Current (LC) (Candela et al., 2002) which interacts with the slope of the YS on its eastern side (Cochrane, 1966; Merino, 1997; Ochoa et al., 2001; Sheinbaum et al., 2002) favoring the outcrop of deep nutrient-rich waters to shallower layers over the shelf. However, the mechanisms responsible for this upwelling and its variability remain unclear.

The wind pattern over the YS is characterized by the trade Winds (easterly winds) throughout the year, with recurrent northerly wind events during autumn and winter caused by cold atmospheric fronts with relatively short duration (Gutierrez-de Velasco and Winant, 1996; Enriquez et al., 2013). The easterly winds drive a westward circulation over the inner-shelf (Enriquez et al., 2010; Ruiz-Castillo et al., 2016). They are also responsible for the upwelling along the zonal Yucatan coast due to divergent Ekman transport (Figure 2). This upwelling is present year-round along the north and northeast coast of the YS, with intensifications from late spring to autumn (Zavala-Hidalgo et al., 2006).

Besides the wind-induced upwelling near the coast, there is also upwelling produced by the interaction of the YC with the eastern YS which is considered the principal mechanism that brings deep nutrient-rich waters over the YS. Observational studies suggest high intrusions of upwelled waters during spring and summer which are suppressed during autumn-winter (Merino, 1997; Enriquez et al., 2013). This seasonal variability is not easy to explain since the YC near the YS does not show such clear seasonal signal and is dominated by higher frequency mesoscale variations (Sheinbaum et al., 2016), so several mechanisms have been proposed to understand it. For example, (Reyes-Mendoza et al., 2016) show how northerly winds can suppress the upwelling at Cape Catoche. Since these cold front northerly winds are active during autumn-winter, they could explain in part the seasonality of the cold water intrusions. But other mechanisms appear to be important too: CTWs (Jouanno et al., 2016), topographic features and bottom Ekman transport (Cochrane, 1968; Jouanno et al., 2018), extension and intensity of the Loop current (Sheinbaum et al., 2016) and encroachment and separation of the YC and LC from the shelf (Jouanno et al., 2018; Varela et al., 2018). External (off-shelf) sea level conditions may also generate pressure gradients that oppose the upwelling and explain its seasonality (Zavala-Hidalgo et al., 2006).

Regarding freshwater inflow, a significant source to the YS is related to submarine groundwater discharge (SGD) due to the karstic geological formation of Yucatan Peninsula (Pope et al., 1191; Gallardo and Marui, 2006), coastal lagoons (Herrera-Silveira et al., 2004), and springs (Valle-Levinson et al., 2011). Due to the complexity of mechanisms and scarcity of observations, the total discharge of SGD into the YS is not well known.

Coupled hydrodynamic-biogeochemical models can be used to establish the TN routes in the marine environment (Fennel et al., 2006). Xue et al. (2013), proposed the first model for TN dynamics in the GoM shelves but excluding the YS. To the best of the authors' knowledge, there are no studies describing the nutrient flux pathways in the YS, so the present work represents the first attempt of a quantitative analysis to understand the biogeochemical cycles and their modulation by physical process at one of the most important socio-economical areas of the southern GoM.

We use a coupled physical-biogeochemical model of the whole GoM to study the nitrogen budget in the YS. The biogeochemical cycles of the YS are poorly known in the GoM and controversies remain regarding its physical dynamics besides the long-term undersampling of biogeochemical variables (Zavala-Hidalgo et al., 2014; Damien et al., 2018), as well as the presence of SGD with unknown fluxes. The main objectives of this study include: (i) quantification of the Total Nitrogen (TN) budget within the inner and outer YS; (ii) investigation of the sources and sinks of nitrogen in the continental shelf and (iii) analysis of the physical mechanisms that modulate the cross-shelf TN transport.

## 2   Model set-up and observational data

### 2.1   Physical model

The physical model is a GoM configuration of the Regional Ocean Modeling System (ROMS) which is a hydrostatic primitive equations model that uses orthogonal curvilinear coordinates in the horizontal and terrain following (*sigma*) coordinates in the vertical (Haidvogel and Beckmann, 1999). A full description of the model numerics can be found in Shchepetkin and McWilliams (2005) and Shchepetkin and McWilliams (2009). Horizontal grid resolution is $\sim$ 5 km, with 36 modified *sigma*

layers in the vertical. We used a new vertical stretching option (Azevedo Correia de Souza et al., 2015) that allows higher resolution near the surface. The numerical domain, which covers the whole GoM, is shown in the bathymetry map in Figure 1a. The model was run for 20 years (1993 to 2012), from which we use 9 years (2002 to 2010) in the present analysis in order to be time-consistent with observational satellite data.

The bathymetry is provided by a combination of the "General Bathymetric Chart of the Oceans" (GEBCO) database (http://www.gebco.net/) with data collected during several cruises in the GoM. The initial and open boundary conditions for temperature, salinity and velocity come from the GLORYS2V3 reanalysis which contains daily averaged fields(Ferry et al., 2012). The model is also forced with hourly tides obtained from the Oregon State University TOPEX/Poseidon Global Inverse Solution (TPXO) (Egbert and Erofeeva, 2002). Hourly atmospheric forcing comes from the "Climate Forecast System Reanal-

ysis" (CFSR) (Dee et al., 2014). These include cloud cover, 10 m winds, sea level atmospheric pressure, incident short and long wave radiation, latent and sensible heat fluxes, and air temperature and humidity at 2 m. These variables are provided at $\approx$38 km horizontal resolution and are used to estimate surface heat fluxes in the model using bulk formulae (Fairall et al., 2003). The model uses a recursive three-dimensional MPDATA advection scheme for tracers, a third-order upwind advection scheme for momentum and a turbulence closure scheme for vertical mixing from Mellor and Yamada (1982).

## 15   2.2   Biogeochemical model

The biogeochemical model is described in Fennel et al. (2006), and is based on the Fasham et al. (1990) model which takes Nitrogen based nutrients as limiting factor. The model is solved for seven state variables, namely: Nitrate ($NO_3$), Amonium ($NH_4$), Phytoplankton ($Phy$), Zooplankton ($Zoo$), Chlorophyll ($Chl$), and two pools of detritus: Large Detritus ($LDet$) and Small Detritus ($SDet$). Details of the model algorithm and coupling to ROMS can be found in Fennel et al. (2006). An

important aspect of this model is a better simulation of denitrification processes at the sediment-ocean interface in the bottom of the continental shelves.

Initial and boundary conditions for the biogeochemical variables were obtained from an annual climatology of $NO_3$, $NH_4$ and $Chl$. The climatology was calculated using all available profiles with the highest quality control from the World Ocean Database (Boyer et al., 2013), and profiles obtained from the XIXIMI cruises carried out by CICESE. The DIVA optimal

interpolation (Troupin et al., 2012) scheme was used to interpolate the individual profiles in the climatology to the model grid. DIVA takes into account the coast line geometry, sub-basins and advection to reduce errors due to artifacts in the interpolation.

The XIXIMI cruises provided profiles of nutrients and chlorophyll in the southern GoM which helps to reduce the bias between the northern and southern part of the GoM. The cruises encompass the region between 12°N and 26°N and -85°W and -97°W, and were carried out within the scope of the "Consorcio de Investigación del Golfo de México" (CIGoM) project

(Gulf of Mexico Research Consortium project in English).

Close inspection of the shelf dynamics through maps of the temporally averaged velocity field $U=(\overline{u},\overline{v})$ (Figure 1b), where the overline denotes the temporal mean, and Mean Kinetic Energy $MKE=0.5(\overline{u}^2+\overline{v}^2)$ (Figure 1c) allows to delimit the shelf into two areas. The first is the inner shelf, delimited by the 50 m isobath where the strongest YS velocities develop (Figure

1b) and where most of the MKE is enclosed (Figure 1c). The second area is the outer shelf between the 50 m and the 250 m isobaths, with the latter isobath representing the shelf break.

The TN examined in this study is taken as the sum of the Dissolved Inorganic Nitrogen (DIN) and the Particulate Organic Nitrogen (PON), with DIN = $NO_3$ + $NH_4$, and PON = $Phy + Zoo + SDet + LDet$, (Xue et al., 2013). The cross-shelf nitrogen
fluxes are calculated as:

$$Q_{50m,250m} = \int\limits_{-50,-250}^{\eta} \mathbf{u}_{cross}\ N\ dz \qquad (1)$$

where $\mathbf{u}_{cross}$ is the velocity component normal to the 50 m or 250 m isobaths, $\eta$ is the model sea level and $N$ can be any component of the TN. The TN cross-shelf fluxes are computed for the North, East, South and West boundaries for both inner and outer shelf, as indicated in the upper right corner box of Figure 1a. Accordingly, the total budget is obtained as the integral
over the area of the shelf and over the depth of the water column for both the inner and outer shelves. The budget also includes the loss to denitrification and to burial in the sediments, which are taken into account for the quantification of the TN budget as sinks of nitrogen.

The initial concentration and boundary conditions at the edges of the GoM model domain (Figure 1a) of the biogeochemical variables: $NH_4$, $Phy$, $Zoo$, $Chl$, and pools of detritus, are set to a small and positive value of 0.1 mmolN m$^{-3}$ following
Fennel et al. (2006, 2011) and Xue et al. (2013). As mentioned in these references, the model quickly adjusts internally to proper variable values within days to weeks. Moreover, these boundaries are far away from the YS and therefore the fluxes across the inner and outer YS determined internally in the model are not impacted by possible inconsistencies at the GoM open boundaries. Given the lack of data for Mexican rivers and ground water fluxes, the same approach is followed for freshwater inputs as done also by Xue et al. (2013). The biological model parameters used in this study are those shown in Table 1
of Fennel et al. (2006), except for the vertical sinking rates which were reduced about 10%, to fit the depth of the Deep Chlorophyll Maximum (DCM) observed with the APEX profiling floats (see Figure A4). The model does not include an explicit compartment for nitrogen in the form of DON although it can be included as in the work of Druon et al. (2010) which adds semi-labile DOC and DON as state variables to the original Fennel et al. (2006) model. They comment on the difficulties of validating the model with observations and highlight open questions even in the definition of both DOC and DON pools (see
also Anderson, 2015). Considering these difficulties and uncertainties, our approach is to use, initially, more basic models to understand their capabilities and build/employ more comprehensive ones upon them later on; so the inclusion of DON and/or DOC compartments is left for future studies.

## 2.3   Freshwater sources

Two riverine systems account for 80% of the freshwater discharge into the GoM, the Mississippi/Atchafalaya system with
18,000 m$^3$s$^{-1}$, and Usumacinta/Grijalva system with 4500 m$^3$s$^{-1}$ (Dunn, 1996; Yáñez Arancibia and Day, 2004; Kemp et al., 2016) (see appendix B). Freshwater contributions to water volume, salinity, temperature and DIN concentration are included

as grid-cell sources into the model. Apart from the two main systems, a total of 81 freshwater sources are included, taking into account freshwater discharges in the Florida, Texas and Yucatan shelves from years 1978 to 2015. For the US rivers the daily data were obtained from the U.S. Geological Survey (USGS) (https://www.usgs.gov/) and the Gulf of Mexico Coastal Ocean Observing System (GCOOS) (https://products.gcoos.org/).

Although the YS has no rivers, freshwater inputs play a key role impacting the local ecosystem (Herrera-Silveira et al., 2002). These inputs come from SGD linked to the "cenotes" ring (sink holes) system inland. The freshwater flux, temperature, salinity, and nutrient concentrations for these sources are not well known. Monthly climatological values were calculated for the Mexican rivers and SGD systems, using temporally scattered information found in the literature (e.g., Rojas-Galaviz et al., 1992; Milliman and Syvitski, 1992; Poot-Delgado et al., 2015; Conan et al., 2016) and a data collection effort within Mexican

institutions led by Dr. Jorge Zavala-Hidalgo (*personal communication*) and from the GOMEX IV cruise of CINVESTAV (Centro de Investigación y Estudios Avanzados in Merida Yucatan) within the CIGOM project. During this cruise, a total of 71 profiles of $NO_3$, potential temperature, salinity and chlorophyll were collected at standard depths from 2-20 November 2015. The localization of the profiles is shown in Figure 2. Therefore, fluxes from US rivers forcing the model present inter-annual variability but Mexican freshwater sources only include a climatology due to lack of information (see appendix B for more

details).

The nitrogen concentration for freshwater sources is essentially DIN. For most of the northern rivers (e.g., Mississippi and Atchafalaya), PON is also considered where available (Fennel et al., 2011; Xue et al., 2013). For the remaining freshwater sources, including the SGD system of YS, the PON contribution is set as a constant small value of 1.5 mmolN m$^{-3}$ due to lack of data.

## 3   Model evaluation for the YS

The model dynamics and its biogeochemistry are validated to guarantee the simulation is able to reproduce basic features of the observations in the GoM, particularly in the YS. Model statistics including biases of physical and biological variables are computed to have some idea of their impact on the estimation of the TN budget over this shelf. Since this is a basin-scale coupled model, a general evaluation of the results and their statistics is carried out considering sea surface temperature, mixed

layer depth, mean kinetic energy, surface chlorophyll and deep chlorophyll maximum over the whole Gulf of Mexico with emphasis on the YS. The results are presented in Appendix A.

### 3.1   YS *In situ* data comparison

Recall that upwelling into the YS is more intense during spring-summer and weaker in autumn-winter (Ruiz-Castillo et al., 2016; Merino, 1997). While the model presents upwelling during all the simulated months, this seasonal behavior is represented

in the model climatologies shown in Figure 2. The figure also shows the position of oceanographic stations occupied during the GOMEX IV oceanographic cruise and delimitation of three areas of particular interest: the inner shelf, the outer shelf, and the upwelling region at Cape Catoche. The climatology of the YS bottom temperature (Figure 3) shows that cold waters enter

into the shelf during spring in agreement with the enhancement of chlorophyll concentrations (Figure 2b). The zonal vertical cross-sections show that the isotherm of 22.5 °C, which traces the upwelled water (Cochrane, 1968; Merino, 1997), outcrops into the shelf during spring (Figure 3c). This is not the case in fall (Figure 3d), and the upwelling is weaker (Figure 2d).

A point-by-point comparison between the model results and the *in situ* observations is shown using only data for November
months from 2002 to 2010 in the model, for compatibility with the observation dates (Figures 4, 5, 7 and 6). Since the simulation is for different years we only expect to reproduce basic features of these observations. The range of temperatures at different depths shown by the model agrees well with those observed during the GOMEX IV (Figure 4). The mean temperature of the observations is 25.5 ±2.9 °C, while the model mean temperature is 24.3 ±3.7 °C. The bias of -1.3 °C is deemed acceptable considering the model mean is a 9 year mean whereas the mean from observations is from just one month and a different year.
A critical area to be evaluated is the upwelling region (see dashed box in Figure 2a), the bias there is -1.1 °C with a root mean square error of 1.68 °C. This means that the model tends to be slightly colder than the observations even inside upwelling waters.

The model mean salinity is 36.5 ±0.2 which matches the 36.5 ±0.2 from observations (Figure 5). Whilst surface salinity in the model is relatively in good agreement with observations (Figure 5a), differences become more important at deeper layers
(Figures 5b and 5c). The root-mean-square error of model salinity (0.23) is low as well as the bias (-0.04) which tends to underestimate the salinity observations. These low differences are also found in the bias for the upwelling area, although the model overestimates the salinity by 0.21 there. The model is able to represent main characteristics of the Caribbean Subtropical Underwater coming from the Caribbean Sea (Merino, 1997) and the Gulf Common Water from the GoM (e.g., Enriquez et al., 2013) within the YS. The warm and high salinity Yucatan Sea Water at the surface described in Enriquez et al. (2013) is present
in the model too, although temperatures do not exceed 31°(not shown) as in observations.

For $Chl$, the model results fall within the range obtained from fluorometer observations in the inner-shelf, outer-shelf and upwelling areas (Figure 6). The mean observed $Chl$ (0.52 ±0.58 mgChl m$^{-3}$) is slightly larger than the model results (0.44 ±0.42 mgChl m$^{-3}$) but within the one standard deviation range, with a bias of -0.08 mgChl m$^{-3}$ and a root mean square error of 1.16 mgChl m$^{-3}$. Notice that there is agreement in $Chl$ concentration between model and observations in the three layers
between 150 m depth and the surface (Figures 6a, b and c). In the upwelling area the model has lower concentrations than observations with a bias of -0.39 mgChl m$^{-3}$ and a root mean square error of 1.39 mgChl m$^{-3}$, although the bias is relatively low, it needs to be taken into consideration for the TN budget. Additionally, a comparison with observed mean chlorophyll vertical profiles over the YS is presented in Figure A9 of Appendix A. Profiles have similar structure but model tends to underestimate the DCM.
To evaluate the temporal behavior of the model $Chl$, time series of the surface chlorophyll averaged over the shelf are compared to similar time series from satellite surface chlorophyll from MODIS (see Figure 8c, and Appendix A for a description of the satellite product) during the simulation period. Mean values of satellite surface $Chl$ are 0.38 ±0.09 mgChl m$^{-3}$ and 0.36 ±0.13 mgChl m$^{-3}$ in the model. Besides reproducing temporal mean and variability of the surface chlorophyll, the model is able to reproduce a positive trend present in the nine years of satellite data. No trend is present in any of the biogeochemical
forcings of the model and determining which physical mechanisms produce it requires further investigation (see below).

The simulated nutrient concentration depicts similar order of magnitude values ( $3.1 \pm 4.6$ mmolN m$^{-3}$) as the observed profiles ( $3.7 \pm 5.2$ mmolN m$^{-3}$) (Figure 7). Surface nutrient concentrations are underestimated by a 1.7 (mmolN m$^{-3}$) compared to observed profiles (Figure 7a). At subsurface depths (25 - 55 m), the model tends to underestimate the NO$_3$ concentrations; however, in the upwelling area, model NO$_3$ concentrations are closer to the observed values with a bias of -0.7 mmolN m$^{-3}$ and larger standard deviations for both model ($4.0 \pm 5.0$ mmolN m$^{-3}$) and observations ($4.81 \pm 6.33$ mmolN m$^{-3}$) (Figure 7b). The temporal variability of the modeled NO$_3$ is larger than the observed NO$_3$ at the surface and bottom as shown by the largest standard deviation in Figure 7b. Below 55 m the modeled and observed NO$_3$ are in good agreement in both, the outer shelf and the upwelling area (Figure 7c). Again, these model results are deemed consistent with observations and are in the range of other values reported in the literature (Merino, 1997). Comparison of similar budgets from other shelves in the GoM can be made (e.g., Xue et al., 2013) though clear interpretation of similarities and differences between them may be difficult given the differences in dynamics and nitrate sources and sinks controlling the budgets on each shelf. One could easily compute budgets per unit area or length for a more sensible comparison among different shelves but in the literature only total budgets are available (see table 2 in Xue et al. (2013)). In that regard, the inner and outer shelf areas are ~74 km$^2$ and ~91 km$^2$, respectively. The TN concentrations for each shelf can be extracted by averaging over the nine year simulation the integrated values of Figure 8, with $4.61 \pm 0.83 \times 10^{16}$ mmolN, and $7.42 \pm 0.89 \times 10^{16}$ mmolN, for the inner and outer shelf respectively.

In addition, the model sea level elevation and surface ocean currents are compared against altimeter products in appendix A, where YC variability and transport from the model are compared with data from three moorings located on the slope close to the eastern YS rim described in Sheinbaum et al. (2016).

## 4    Total Nitrogen budget and cross-shelf transports in the YS

Time series of spatially averaged TN over the YS suggest a positive trend over the nine simulated years. The trend is seen in both the inner and the outer shelves (Figures 8a and b). This, perhaps, could be expected given the positive trend in both model and satellite surface $Chl$ mentioned before (Figure 8c). Varela et al. (2018) report a cooling trend of the inner YS and suggest may be associated with an eastward shift of the YC. We searched for possible connections between the trends in chlorophyll and TN and indices measuring the position and strength of the YC in the model but found no correlation. This is an interesting problem currently under investigation and to be reported elsewhere.

In the inner shelf there are similar total integrated values of DIN and PON (Figure 8a). This indicates the presence of a very "efficient" biogeochemical cycle in the inner shelf (see explanation below). By contrast, in the outer shelf, DIN values are larger than PON (Figure 8b) probably because the integration in the outer shelf includes a large volume below the euphotic zone. Temporal series of integrated TN show a combination of low frequency variability associated with the seasonal cycle as well as interannual variability, but longer period integrations are required to properly investigate the latter.

To understand the high TN variability in the YS, quantification of the cross-shelf fluxes becomes necessary. Their impact on the TN budget and the physical mechanisms modulating such fluxes are investigated next.

Cross-shelf fluxes are quantified for the two compartments, the inner and outer shelves (Figure 1b), and for all the boundaries of each compartment. A schematic view of the main incoming and outgoing pathways of cross-shelf TN fluxes is shown in Figure 9. The yearly averages of the spatially integrated cross-shelf fluxes are shown in Table 1.

For the inner shelf, both PON and DIN are imported through its northern and eastern boundaries and exported through the

west and south borders. Inner shelf acts as a source of PON for the Campeche Bay at the southwest margin. The major source of TN for the inner shelf is from the outer shelf via the Cape Catoche upwelling, representing 80% of the total, while freshwater sources contribute the other 20%. Although the latter is a relatively large source of nitrogen, its relevance seems to be confined to the NW part of the inner-shelf. In general, there is compensation between the DIN and PON concentrations in the inner-shelf (Figure 8a) due to an efficient biogeochemical cycle whereby almost all the DIN imported into the shelf is consumed by

the phytoplankton and thus converted into PON. The efficiency relies on the shallowness of the inner shelf ($\sim$50 m depth), because, if strong mixing conditions are present, organic matter will distribute throughout the shallow water column. This is enhanced during winter, when vertical wind-driven mixing and convective processes are strong enough to reach the sea bottom. Additionally, vertical shear likely generated by bottom friction can lead to instabilities and vertical mixing able to break the stratification and carry nutrients to the euphotic zone. During summer months, vertical mixing is weaker (not shown). Turns

out that, in the model, vertical velocities in the inner shelf are quite intense and upward throughout the year ($\sim$ 5 m day$^{-1}$) carrying nutrients to the euphotic layer. The cause of these vertical velocities is under investigation using a higher resolution model configuration.

By contrast, in the outer shelf, the largest inputs of PON and DIN are advected from its southeastern corner. The eastern boundary is a source of DIN but a sink of PON for the outer shelf. Therefore, the budget reveals that the PON exported to the

inner shelf is produced in the outer shelf and not advected from Caribbean waters. The contribution of TN from the inner shelf to the outer shelf represents only 1.5% of the total inputs.

Over the outer shelf the fluxes of nutrients and organic matter are driven by a westward wind driven circulation (Ruiz-Castillo et al., 2016) and exported to the deep GoM and the Campeche Bay through the north and west borders respectively. This represents a source of DIN, $Phy$ and $Zoo$ to these oligotrophic regions.

In that regard, the model reveals a quasi-permanent thin filament of $Chl$ that is advected from the northwest corner of the outer shelf to the west of the Campeche Bay (Figure 10a). A vertical section of the cross shelf fluxes along the 250 m isobath in the western YS (TN, Figure 10b) shows that while the export of organic matter to the open sea is concentrated in the surface layers (Figure 10d), the bottom layer presents a net DIN export (Figure 10c). The climatological average over nine years of simulated $Chl$ show that this filament is intensified during winter times (not shown), although it is present during the whole

simulation period. Sanvicente-Añorve et al. (2014) studied the larval dispersal for coral reef ecosystems in the southern GoM. They show that the northwestern corner of the outer YS acts as a sink region for larvae. Similar to other coral reef systems, they attributed the sink to the influence of circulation patterns that lead to a unidirectional dispersion pattern during the whole year. Our model results seem to support this idea.

Denitrification is a form of anaerobic microbial respiration in which nitrate and nitrite are finally reduced to molecular

nitrogen (N$^2$). It represents a major sink for bioavailable nitrogen. The spatio-temporal average rate of denitrification for the

YS is of 1.11 $\pm$0.13 mmolN m$^{-2}$ d$^{-1}$. Our results suggest that denitrification removes up to the 53% of the TN in the inner shelf, a significant percentage that agrees with estimates from other shelves in the GoM (Xue et al., 2013). On the other hand, denitrification in the outer shelf only removes 9% of the TN. Our results also indicate that denitrification rates tend to increase with time for both inner and outer shelves (not shown), in concert with TN concentrations (Figures 8a and b). This is expected since denitrification is a reduction process, hence an increase in nitrate concentration means more available DIN to be reduced to N$^2$.

## 4.1 Physical modulation of cross-shelf TN flux by CTWs

Many physical process coexist at different spatio-temporal scales in the YS that modulate the cross-shelf transport of nutrients and organic matter. We suggest that at least two processes are responsible for such modulation: CTWs and interaction of the YC with the eastern shelf break.

CTWs can be generated by wind forcing over irregular bottom topography along the coast and have been the subject of investigation for a long time (e.g., Clarke, 1977). In the GoM, CTWs are forced by alongshore winds and then travel anti-clockwise with the coast on its right until they reach the western portion of the Yucatan Peninsula, mainly associated with cold fronts (Dubranna et al., 2011; Jouanno et al., 2016). CTWs have a signature in sea level that is well captured in relatively high resolution models such as the one used in the present study ($\sim$ 5 km). In their modelling study, Jouanno et al. (2018) suggest that CTWs may influence the Yucatan upwelling pulses. In this study, the presence of CTWs is corroborated and its effect on the modulation of cross-shelf nutrient fluxes at the west margin of the YS is exposed.

The presence of CTWs in the model simulations is evidenced in the hovmöller diagram along the 50 m isobath shown in Figure 11. Phase speeds are in the range of [2 - 4] m s$^{-1}$ in agreement observations (Dubranna et al., 2011) and other models (Jouanno et al., 2016).

The cross-shelf TN in the YS western inner shelf boundary exhibits high-frequency variability. The daily climatology of the wavelet power spectrum of wind-stress, sea level anomaly (SLA) and western boundary cross-shelf TN temporal series for the inner shelf of Figure 12, show that both, along-shelf wind stress and changes in SLA may be linked with the cross-shelf TN variability in the inner shelf. The three variables show maximum energy during winter times when CTWs are expected to be more intense, and the wind increases its magnitude due to the incursion of the "Nortes" (cold front winds).

It is worth mentioning that the wavelet power spectrum for the whole 2002-2010 period (not shown) depicts an interesting intensification of cross-shelf flow (and nutrient fluxes) during 2003,2004,2009 and 2010 which coincides with Niño-Modoki events (Ashok et al., 2007; Ashok and Yamagata, 2009). The possibility of such connection deserves further investigation.

To further examine the relationship between these physical and biogeochemical variables, results from a cross-correlation spectral analysis are shown in Figure 13 for the time series used in the wavelet analysis of Figure 12. The variability of along-shelf wind-stress and cross-shelf TN fluxes shows significant coherence in the 8-10 day period band at nearly zero phase lag (Figure 13). Coherence between cross-shelf TN fluxes and SLA is also coherent in the same band (peaks at 8 and 8.4 days) but 180 degrees out of phase. This is consistent with offshore Ekman transport produced by along-shelf northerly winds triggering nutrient and organic matter fluxes across the western boundary of the YS and negative SLA at the coast.

Propagation of CTWs is evident in the Hovmöller diagram of Figure 11 and most certainly modulates the cross-shelf TN transport. The coherent 8-10 day period band (and also at other higher frequencies, e.g 5-6 day period) is in agreement with those reported in the literature for CTWs in the GoM (e.g., Jouanno et al., 2016). Since the coherence analysis is carried out here using time-series of spatially averaged quantities (from 20°30' N to almost 22°N, approximately 100 km), possible
phase-lags are probably masked.

## 4.2 Influence of the Yucatan Current in the coastal upwelling

Observational studies suggest that favorable-upwelling winds at the northern Yucatan coast are present all year round (Ruiz-Castillo et al., 2016; Pérez-Santos et al., 2010). Cold SSTs on the YS vary seasonally and are particularly characterized by a cold water band on the inner YS very close to the coast that appears in spring and continues until the beginning of autumn
(Ruiz-Castillo et al., 2016; Zavala-Hidalgo et al., 2006). Pérez-Santos et al. (2010), using ten years of sea surface wind data from QuikSCAT, show that Ekman transport is the main contributor to the upwelling over the north YS (93 %), with Ekman pumping only contributing 7%.

This upwelling regime requires a supply of cold and rich-nutrient deeper waters from the open ocean to maintain the observed biological productivity on the YS.

The main import of TN to the YS is through the southeast and eastern YS boundaries via mechanisms related to the dynamics of the Yucatan and Loop currents and their interaction with the YS shelf-break, such as intensification, separation and/or encroachment from the coast, bottom boundary layer transport, advection, instabilities, eddies and the presence of particular topographic features (e.g.submarine canyons. The reader is referred to (Roughan and Middleton, 2002), for a discussion of upwelling mechanisms on the East Australian Current that appear to be relevant here too as several local studies indicate
(Cochrane, 1966; Merino, 1997; Zavala-Hidalgo et al., 2006; Enriquez et al., 2010, 2013; Enriquez and Mariño Tapia, 2014; Carrillo et al., 2016; Jouanno et al., 2016, 2018).

On the other hand, the export of TN to the deep GoM through the YS northern margin can also be related to advection by the YC and associated mesoscale structures (Roughan and Middleton, 2002; Carrillo et al., 2016; Enriquez and Mariño Tapia, 2014).

Correlation analysis between the strength of the cross-shelf flow from the YC and TN, PON and DIN fluxes at the eastern margin, all vertically integrated, show high values at seasonal time scales (Figure 14). The time series are filtered by a 30 day moving average window to remove high frequency variability. The square of the correlation coefficients ($r^2$) for TN, DIN, and PON against the vertically integrated YC are indicated on top of the Figure 14. These results indicate that TN fluxes are well correlated with the strength of the current.

To investigate the possible role of the position and trajectory of the YC and its closeness to the YS in the upwelling (Enriquez et al., 2010, 2013; Jouanno et al., 2018) we computed an index measuring the closeness of the YC core to Cape Catoche and the Notch areas in the model, which are two places where water tends to upwell (Merino, 1997; Jouanno et al., 2018). The index depicts no seasonality that could be directly connected to strong(weak) upwelling during spring (autumn). This is an indication

that seasonality of the inflow of rich nutrient water into the YS is probably more influenced by other processes as discussed in Reyes-Mendoza et al. (2016), such as cold front winds that can stop the upwelling or other non-local perturbations.

One of the important mechanisms suggested since Cochrane (1966) to be responsible for the YS eastern boundary upwelling and the nutrient flux towards the coast is bottom Ekman layer transport produced by interaction of the YC with the upper slope and shelf break. The stress exerted by the intense along-shore velocity of the YC on the topography generates an Ekman spiral at the bottom boundary layer and a net depth integrated transport to the left i.e., a cross-shelf transport towards the shelf in the boundary layer. For example, Shaeffer et al. (2014) using glider observations find that bottom Ekman transport can explain up to the 71% of the bottom cross-shelf transport variability on the southeastern Australian shelf produced by the East Australian Current.

Here we present modeling evidence that bottom Ekman layer transport could be the precursor of the upwelling in Cape Catoche. The Bottom Ekman Transport ($U_{bE}$, m$^2$ s$^{-1}$) can be taken as $U_{bE} = -\tau_{by}/(\rho_o f)$, where $\tau_{by}$ is the bottom stress computed by $\tau_{by} = \rho_o C d v_b \sqrt{u_b^2 + v_b^2}$, with $Cd$=1×10$^{-3}$ the drag coefficient, $u_b$ and $v_b$ are the bottom velocities at the 250 m isobath, $f$ the Coriolis frequency and $\rho_o = 1025$ kg m$^{-3}$ the reference potential density of the sea water. The analysis shows that the time-mean $U_{bE}$ is toward the shelf (defined positive here), and is well correlated with the bottom cross-shelf water transport ($r^2 = 0.71$, $ci = 95\%$) calculated directly (Figure 15a). The Ekman transport is calculated from the theoretical formula (i.e. stress divided by the coriolis frequency) whereas the direct transport is calculated using the bottom velocities and integrating on the last grid cell. We should mention here that the bottom grid cell at this depth has a vertical size of $\sim$20 m. Using standard formulas to estimate the width of the Ekman layer (e.g., Cushman-Roisin and Beckers, 2011; Perlin et al., 2007) from bottom velocities or stresses and stratification we obtain values $\sim$ 10-30 m, therefore the layer is not really resolved by the model grid. The correlation is also large over time ($r^2 = 0.78$, $ci = 95\%$) as shown in Figure 15b. Figure 15c shows that the vertically integrated TN transport averaged over nine simulated years and over latitude is towards the coast at 250 m depth, that is, at the bottom-most model layer which is considered here as the bottom Ekman layer.

Comparison between bottom layer Ekman transport and the time mean vertically integrated TN transport across the eastern 250 m isobath indicates that bottom Ekman transport is responsible for 65 % of the TN that is entering the shelf. The mechanisms that explain the remaining flux need to be further investigated and are probably related to meanders, eddies, topographic features and other processes. Moreover, bottom Ekman transport can be arrested by stratification and may not be dominant everywhere along the YS east coast as has been documented in other western boundary upwelling regions (e.g., Roughan and Middleton, 2002, 2004). Our goal here was only to estimate the size of the TN fluxes related to the bottom Ekman layer and determine its relative importance.

## 5   Model uncertainties

The bias of the model with respect to observations described in section 3 (see also appendix A) is analyzed in order to examine how uncertainties may impact the TN budget calculation.

The model tends to overestimate/underestimate $Chl$/SST in winter and underestimate/overestimate $Chl$/SST in summer. This bias produces more intense upwelling at Cape Catoche during spring than in summer. In fact, upwelling waters are still present during winter (Figures 2) but not in observations. The filtered seasonal time series of bottom Ekman transport shown in Figure 15b (black line), depict the same pattern: they indicate that water from the Caribbean Sea entering into the YS (via

bottom Ekman transport) increases during spring towards the summer, decreases during autumn and increases again during winter. This is in agreement with Figure 3.

In the water column, the model underestimates $NO_3$ concentration a maximum of 15% and is also about 5% colder than the observed vertical profiles. These biases can impact the growth of phytoplankton whose maximum growth rate (Eppley, 1972) depends on temperature and nutrient concentration (Fennel et al., 2006). However, since phytoplankton only represents 15%

of the TN, the overall impact on TN of these biases is estimated to be less than 3%. The main point we are trying to make is that, although there are model biases, the main processes that control the TN budget in the YS are well captured by the model simulation particularly the Cape Catoche upwelling, which together with the southeastern boundary, represent the main entrance of TN to the YS.

## 6    Summary and concluding remarks

The TN budget, main nutrient transport pathways and their modulation by physical process over the Yucatan shelf have been investigated using a nine year simulation from a ROMS physical-biogeochemical coupled model for the GoM. Our work provides a first general view of the shelf physical-biogeochemical coupled system, schematized in Figure 16.

The results indicate that TN, especially DIN, enters the Yucatan outer-shelf through the southeastern and eastern margins. The TN is then driven by a westward shelf current and then exported to the deep GoM and Campeche Bay through the northern

and western boundaries, respectively. In the inner-shelf, the biogeochemical nitrogen-based cycle seems to be very efficient for $NO_3$ remineralization/consumption by the phytoplankton converting most of the DIN to PON. The freshwater sources represent an important contribution of about 20% to the DIN concentration, although it is restricted to the northwest of the Yucatan peninsula. Denitrification represents the main sink of nutrients for the inner shelf, removing more than the 50% of the nitrogen. The inner shelf contributes to the TN of the outer shelf at its western edge, but this contribution is less than 2%,

indicating weak fluxes from the inner to the outer Yucatan shelf. By contrast, the outer shelf is the main nitrogen supplier of the inner shelf, particularly of PON from the eastern margin. A quasi-constant filament in the outer shelf western border represents an important source of both organic and inorganic nitrogen for the oligotrophic Campeche Bay.

Surface Ekman layer dynamics at the western and northwestern shelf borders play an important role in the transport of nutrient and organic matter to the Campeche Bay and deep central GoM. Part of the high-frequency variability of the TN fluxes

at the western YS boundary are correlated and in phase with the along-shelf wind-stress modulating the variability of TN across the western shelf of Yucatan in the 5-10 period band. These high-frequency TN fluxes are also correlated with changes in SLA at similar periods, which are also typical of CTWs found in the GoM. Coherence is 180 degrees out of phase and consistent

with negative SLA resulting from offshore Ekman transport. This exchanges are enhanced during winter due to cold frontal atmospheric systems "Nortes".

The advection by the YC dominates the nutrient concentration import to the YS through the southeastern border. This advection, together with the influence of mesoscale structures, control the export of nutrients to the deep GoM at the northern margin. A different process modulates the flux of nutrients at the eastern YS margin. The YC flowing parallel to the slope plays an important role in the intrusion of DIN into the shelf. Initial estimates carried out here suggest that, in the model, bottom Ekman layer transport explains the deep TN flux through the eastern YS boundary. There is a positive mean transport (into the shelf) over the nine simulated years along the eastern shelf break so friction generated between the YC and the shelf break produces a net bottom Ekman transport towards the shelf. This produces the upwelling at Cape Catoche on the eastern shelf, but it is not the only process at work: external and remote forcings appear to control its seasonality (e.g. winds, CTWs); besides, other upwelling mechanisms such as divergence/convergence from current separation/encroachment, eddy-current interactions with topographic features (e.g submarine canyons) may be important too and must be analyzed in future research.

*Data availability.* Data from the model simulation used in this study are available upon request to the corresponding author

## Appendix A: Model evaluation

### A1    Basin scale model evaluation

This study is focused on the Yucatan shelf region, whose hydrodynamics and biogeochemical outputs were previously validated before the analysis. This appendix provides a summary of this validation to provide evidence that the basic features of the whole GoM circulation is correctly represented in the model. The time-mean eddy kinetic energy (EKE, $m^2 s^{-2}$) map computed from AVISO geostrophic velocities (http://www.marine.copernicus.eu) is used to compare it with the EKE from the model (Figures A1a and b) for 17 years (1995-2012). The model is able to capture the main features of the eddy field exhibited by the altimeter product as well as the main structure of the Loop Current. In particular, the mean and standard deviation of the eddy kinetic energy field are reasonably captured by the model. The model produces a hook-like pattern of EKE in the western part of the GoM, between 24 and 28 °N, that is more evident in the standard deviation of model EKE (Figures A1c and d). This pattern is not so clear in the AVISO maps but has been identified using lagrangian data (e.g., Gough et al., 2019). It is associated with the GoM western boundary current that isolates the western continental shelf from the open ocean. EKE is higher in the model, particularly at the Yucatan channel and Florida Straits, probably due to the higher resolution of the model (∼5 km) compared to the altimeter product (∼28 km).

Seasonal climatology of the sea surface temperature (SST) are also compared with the Aqua-MODIS satellite products (http://modis-atmos.gsfc.nasa.gov) (Figure A2). The model SST shows a good agreement with seasonal cycles exhibited by the satellite data. The overall bias for the deep GoM SST (depths > 1000 m) is in the range [-0.21, +0.21] °C. Larger differences are found near the coast. The model tends to underestimate the coastal SST during winter and overestimates it during summer.

Nevertheless, these differences are less than 0.5 °C, and on average differences are on the order of 0.05 °C with a standard deviation of 0.4 °C. The relatively good agreement between model and data is perhaps not very surprising considering that observed air temperatures are provided to the model to compute heat fluxes using bulk formulae. At the same time, no flux correction is applied in the model so it is important to confirm that there is no drift in the simulation.

In order to evaluate the mixed layer depth, a total of 2629 ARGO floats profiles, available in the period between 1995-2012, are compared with the mixed layer depth given by the model in the deep GoM. This is an important quantity in terms of biogeochemical behavior since the Gulf is an oligotrophic region in which the vertical advection of nutrients controls primary production to the photic layer (Fennel et al., 2006; Xue et al., 2013; Damien et al., 2018). The biogeochemical cycles are partly controlled by the difference between the deep and dark nutrient-rich waters and the upper ocean layer where the availability

of light promotes the growth of phytoplankton and hence zooplankton. Figure A3 shows that the model can reproduce the seasonal cycle of the mixed layer in the GoM, with deepening during winter and shallowing during summer seasons (Damien et al., 2018; Portela et al., 2018). The model depicts shallower mixed layer depths during summer and deeper during winter than the Argo observations. The higher variability of the observed data is likely related to mesoscale structures and submesoscale process which can locally deepen/shallow the mixed layer (e.g., Boccaletti et al., 2007; Fox-Kemper et al., 2008; Levy et al.,

2012) not fully represented by the model. Despite the differences found, the bias between observations and model mixed layer depths are on the order of 1.4 m.

The lack of spatio-temporal biological data sets to validate biogeochemical models in the GoM is a well-known problem (Walsh et al., 1989; Damien et al., 2018). Only satellite-derived surface chlorophyll concentration is available with enough spatial and temporal cover but only at the surface. These observations give us a general overview of the chlorophyll temporal

and spatial distribution patterns at basin scales. Monthly mean time series (2002-2010) of chlorophyll-a concentration from Aqua-MODIS and SeaWiFS 9 km and 4 km (when available) satellite products are used for a basin-scale model evaluation. The temporal series averaged for the whole deep GoM (i.e., excluding high productive coastal areas with less than -1000 m depth) show a good agreement between the coupled model and the observations. The model tends to overestimate the $Chl$ in winter and underestimate it in summer (Figure A5). Despite some exceptional years (e.g., 1999), the modeled chlorophyll

concentration values fall in the range exhibited by the satellite products. Mean satellite $Chl$ is 0.1448 $\pm$ 0.04 mgChl m$^{-3}$ in contrast with mean modelling $Chl$ values of 0.1433 $\pm$ 0.09 mgChl m$^{-3}$.

Observations of the vertical chlorophyll structure are available from eight APEX profiling floats with 537 profiles of $Chl$ from 0 to 2000 m every ten days within the GoM (Pasqueron de Fommervault et al., 2017) (Figure A4). A more detailed description of this database is provided by Hamilton et al. (2017), and the $Chl$ data calibration is explained in Pasqueron de

Fommervault et al. (2017). The resulting profiles give valuable information to evaluate biogeochemical models through the water column, in contrast to the surface only information from satellite measurements (Pasqueron de Fommervault et al., 2017; Damien et al., 2018). The comparison shows that the model is able to reproduce the depth of the DCM measured by the floats. The DCM seasonal cycle is also well represented by the model. It is interesting to note the high dispersion in the data, revealing the large $Chl$ variability found in the deep GoM.

## A2 Regional chlorophyll model evaluation

In addition to the comparison of the surface chlorophyll temporal series with satellite products (Figure 8c), *in situ* spatially averaged vertical profiles of chlorophyll from three GOMEX cruises carried out during November 2015, August 2016 and July 2018 are also compared with the model chlorophyll profiles averaged for all the July, August and November from 2002 to 2010. The observed profiles superimposed in blue, are shown in Figure A6. The result shows that the model is also able to reproduce the large variability of the observed data. Highest values of chlorophyll from model profiles are found at the surface layers, between 5 and 15 m depth. Values higher than 6 mgChl m$^{-3}$ represent only the 0.64% of the total simulated points, while for observations, the percentage is about 0.06% and are also located at the surface between 10 and 35 m depth.

## A3 Regional altimetry and ocean currents comparison

The variance of the Absolute Dynamic Topography (ADT) from AVISO, which is the sea surface height above the geoid obtained as the sum of the sea level anomaly (SLA) and the mean dynamic topography, is compared with the variance of the sea level of the model output (Figure A7). Observed and model ssh variance have good resemblance. There are slight differences in the northern coast of the YS. Remember, however, that the accuracy of the altimeter observations is reduced in shallow areas (Vignudelli et al., 2011).

The variability and magnitude of the current over the shelf is also compared against the GlobCurrent product (www.globcurrent.org) (Rio et al., 2014). Since the current velocity over the YS is a westward wind-driven flow (Ruiz-Castillo et al., 2016), a comparison with only the geostrophic velocity contribution might not represent the whole state of the velocity field. In this regard, the GlobCurrent product is the result of combining geostrophic altimeter velocity with the addition of the wind-forced Ekman velocity contribution under ocean mixed layer model assumptions. The results are shown in Figure A8. The model correctly represents the mean surface current magnitude and direction over the shelf, highest differences are found close to the Yucatan coast (Figure A8a and b). The variability ellipses (Figure A8c) show that the current variability over nine years from the model agree with those from observations. Near the northern Yucatan coast, values are lower in both model and data. However, the model ellipses are zonally oriented in contrast to the meridional orientation of the ellipses from the satellite product. The other important difference is found at the west coast of the YS, where the model exhibits a southwestward oriented ellipses whereas the satellite shows a westward orientation. This might influence the direction of the TN fluxes at the west YS boundary, a subject which is addressed in section of Model Uncertainties. Similarly, as the previous comparison with the AVISO product, significant differences are found near-coast but there are probably significant errors in the data Vignudelli et al. (2011). In contrast, the YC is well represented by the model in terms of its spatio-temporal variability, although its magnitude is overestimated, which again is probably an effect of better model spatial resolution.

## A4 Yucatan Current evaluation

The CICESE-CANEK mooring sections monitoring the flow in the region duiring 2009-2011 is shown in Figure 1a (yellow zonal line). The current velocity normal to the three mooring transects shown in Figure 8 of Sheinbaum et al. (2016) during

years 2009 to 2011 is used for validation. They observe that the YC (YUC transect) was located between the surface and 800 m depth, which agrees with our model results shown in Figure A9a. The core of the YC is located over the West Yucatan slope, and its mean of 1.18 m s$^{-1}$ is in a very good agreement with observations (Sheinbaum et al., 2002, 2016). The model also shows that the highest standard deviation is at the surface on the western side of the channel with a value of 0.3 m s$^{-1}$ (Figure A9d), in contrast with the 0.4 m s$^{-1}$ found by Sheinbaum et al. (2016). They argue that this variability is due to changes in the current position and the counter-flow. At deeper layers (below 900 m), the model shows that the current flows towards the GoM at the center of the section. On both, the western and eastern side of the section, the model is able to reproduce the southward flow as shown in Sheinbaum et al. (2016).

For sections PE and PN (Figure A9b and c), the model exhibits mean velocities of 0.24 $\pm$ 0.24 m s$^{-1}$ and 0.36 $\pm$0.29 m s$^{-1}$, these values are lower than reported by Sheinbaum et al. (2016) of 0.6 $\pm$ 0.7 m s$^{-1}$ and 0.4 $\pm$0.6 m s$^{-1}$ (Figures A9e and f). One should consider that the model has high variability. Moreover, these sections may or may not be influenced by the core of the Loop Current. Sheinbaum et al. (2016) estimate a reduction of about 30-50% in the maxima of the mean when the Loop Current core is not passing over the sections moorings. The southward flow over the slope of section PE below 1000 m is well represented by the model (Figure A9b), as well as the flow across the whole PN section (Figure A9c).

## Appendix B: Freshwater sources inputs

As already described in subsection 2.3, freshwater, and nitrogen input from 81 major rivers and freshwater systems are included in the coupled model simulation. The Mississippi and Atchafalaya riverine systems are the largest fluvial source in the GoM (red and blue points in Figure A10a). Their nitrogen delivers tripled in the last decades and are meaningfully correlated with the coastal DIN concentration in the northern GoM (Xue et al., 2013). Nutrient and transport of this system generally peaked in spring-summer in agreement with the time series inputs shown in Figure A10b.

The Usumacinta-Grijalva rivers system (green points in Figure A10a) is the most important freshwater source in the southern GoM (Xue et al., 2013). The highest riverine discharge of this system, accompanied by an enhancement of nutrient concentration, occurs during winter and decreases during spring (Figure A10c). In contrast to the northern riverine sources, the data available for the southern freshwater sources of the GoM are scarce or undersampled. In order to obtain southern freshwater inputs, a time series is built based on the composite of temperature, salinity, volume transport and nutrient concentration at the location of the freshwater sources or near it. The information is obtained from values reported in the literature (Milliman and Syvitski, 1992; Herrera-Silveira et al., 2002, 2004; Yáñez Arancibia and Day, 2004; Hudson et al., 2005; Herrera-Silveira and Morales-Ojeda, 2010; Poot-Delgado et al., 2015; Kemp et al., 2016; Conan et al., 2016) and from observational hydrographic stations near the Yucatan coast (subsection 2.3). In general, the information reported does not cover a large time series. Therefore, all the information is used to build a climatology which will serve as freshwater model input. An example of this climatological inputs is depicted in Figure A10c and d for the Usumacinta/Grijalva rivers and the Yucatan freshwater sources (magenta point in Figure A10a) (Yáñez Arancibia and Day, 2004; Poot-Delgado et al., 2015; Conan et al., 2016). The

YS receive freshwater and nutrient inputs from spring and runoff from mangrove areas, lagoons, and cenotes. High nutrient concentrations are reported for YS lagoons (e.g., Dzilam Lagoon) (Herrera-Silveira and Morales-Ojeda, 2010).

As one can notice, the inter-annual variability is visible in northern riverine systems, while is absent in the southern freshwater sources due the lack of information. Moreover, it is essential to note that the small-scale variability in most of the GoM rivers structure is not fully resolved by the horizontal resolution of our model configuration.

*Competing interests.* The authors declare that they have no conflict of interest

*Acknowledgements.* This research is funded by the National Council of Science and Technology of Mexico - Mexican Ministry of Energy-Hydrocarbon Trust, project 201441. This is a contribution of the Gulf of Mexico Research Consortium (CIGoM). The authors would like to thank the NASA Goddard Space Flight Center, Ocean Ecology Laboratory, Ocean Biology Processing Group. Moderate-resolution Imaging Spectroradiometer (MODIS) Aqua Chlorophyll Data; 2014 Reprocessing. NASA OB.DAAC, Greenbelt, MD, USA. doi: 10.5067/AQUA/MODIS/ L3M/CHL/2014 and to the Sea-viewing Wide Field-of-view Sensor (SeaWiFS) Chlorophyll Data; doi: 10.5067/ORBVIEW-2/SEAWIFS/ L3M/CHL/2014. The Ssalto/Duacs altimeter products were produced and distributed by the Copernicus Marine and Environment Monitoring Service (CMEMS) (http://www.marine.copernicus.eu). We acknowledge the provision of supercomputing facilities by CICESE. The authors are grateful to Dr Victor Camacho-Ibar (UABC) and Dr Sharon Herzka (CICESE), who provided nutrient data measured during the XIXIMI-IV cruise (funded by INECC-SEMARNAT, FOINS-CONACYT, Mexico). The APEX floats dataset were produced in the framework of the "Lagrangian Study of the Deep Circulation in the Gulf of Mexico", funded by the Bureau of Ocean and Energy Management, USA. Argo data were collected and made freely available by the International Argo Program and the national programs that contribute to it. (http://www.argo.ucsd.edu, http://argo.jcommops.org). The Argo Program is part of the Global Ocean Observing System. Argo float data and metadata from Global Data Assembly Centre (Argo GDAC). SEANOE. http://doi.org/10.17882/42182.

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

**Table 1.** Nutrient budget in molN yr$^{-1}$ for the inner (50 m isobath) and outer (250 m isobath) Yucatan shelf, computed at each boundary (N, W, E and S, see Figure 1a) using cross-shelf velocities. The flux of nutrients is integrated through the water column and temporally averaged using the period 2002-2010 to compute the budget from daily model fields. Positive (negative) values represent sources (sinks) of nutrients. Denitrification is always a nitrogen removal process.

| Boundary | PON | DIN | TN | Fresh water/Inner[a] |
|---|---|---|---|---|
| Inner-shelf budget (x10$^{10}$ molN yr$^{-1}$) | | | | |
| N | 0.34 | 1.63 | 1.97 | 0.76 |
| W | -0.72 | -0.02 | -0.73 | 0.72 |
| E | 2.35 | 1.68 | 4.32 | 0 |
| S | -2.29 | -0.05 | -2.34 | 0 |
| Denitrification | -3.34 | | | |
| Trend[b] | -0.64 | | | |
| Outer-shelf budget (x10$^{10}$ molN yr$^{-1}$) | | | | |
| N | -11.46 | -7.42 | -18.88 | -1.97 |
| W | -1.85 | -9.87 | -11.72 | 0.72 |
| E | -0.28 | 7.65 | 7.36 | -4.03 |
| S | 11.17 | 27.74 | 38.92 | 0 |
| Denitrification | -3.34 | | | |
| Trend[b] | -0.66 | | | |

[a]Fresh water sources are considered only for the inner-shelf. Inner can be taken as a source or sink of nitrogen only for the outer-shelf.

[b]The positive trend of total nitrogen observed in the temporal series during nine years is also taken into consideration to close the budget.

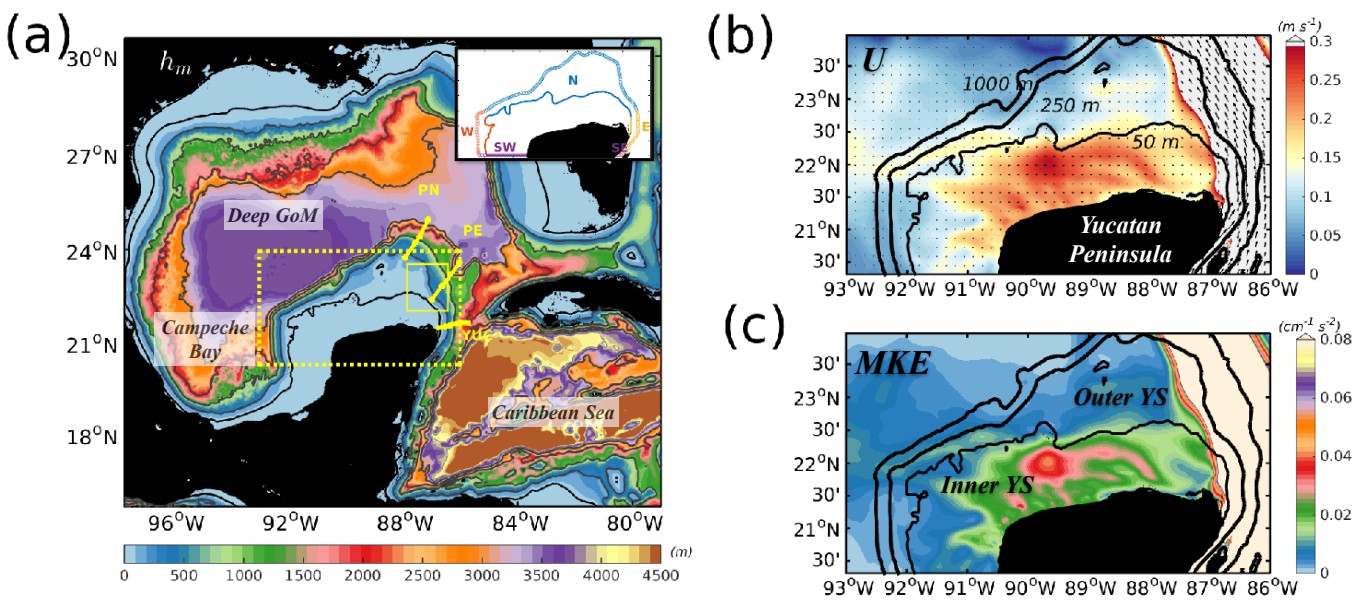

**Figure 1.** Bathymetry ($h_m$, m) of the whole model domain. Isobaths: 50, 250, 1000, 2000, 3000 and 4000 m are also shown in gray contours. The small box at the upper right corner shows the North, East, South and West boundaries used to compute the inner and outer shelf TN cross-shelf fluxes. The yellow dashed box delimits the study area of the Yucatan shelf, where (b) is the surface temporally averaged velocity field ($U$, m s$^{-1}$) with magnitude in color and vectors representing the direction; and (c) is the surface Mean Kinetic Energy (MKE, cm$^2$ s$^{-2}$) computed for the year 2010. The smallest yellow box in (a) shows the "notch" area (see text) and the three yellow lines are the mooring locations for transects YUC, PN and PE. Labels help identify the Deep Gulf of Mexico, Campeche Bay and Caribbean Sea regions in (a). The inner and outer Yucatan Shelf are shown in (c).

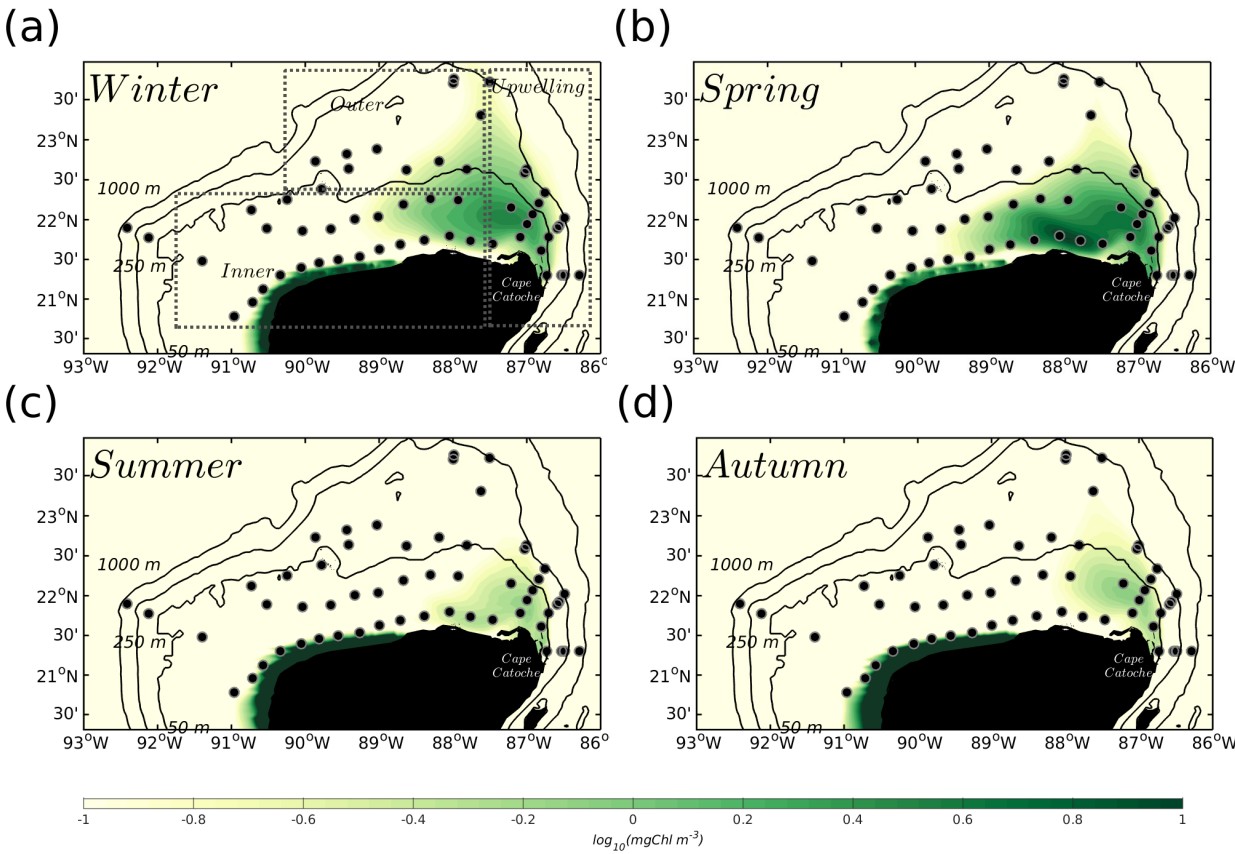

**Figure 2.** Seasonal climatology of surface chlorophyll (mgChl m$^{-3}$) given by the biogeochemical coupled model for: (a) Winter (Jan, Feb, Mar); (b) Spring (Apr, May, Jun); (c) Summer (Jul, Aug, Sep) and (d) Autumn (Oct, Nov, Dec), for the 2002-2010 period. Dashed boxes in (a) denote the three areas in which the validation with observations (black dots) was carried out, i.e., inner shelf, outer shelf and the upwelling region close to Cape Catoche.

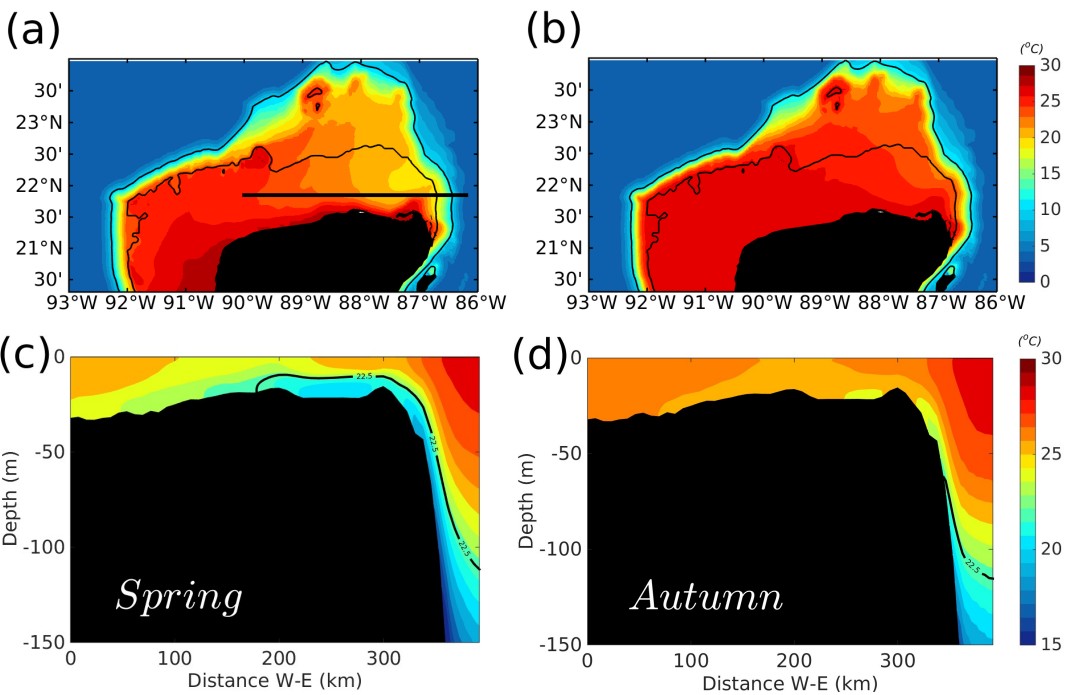

**Figure 3.** Seasonal climatology of bottom temperature (°C) for (a) spring, and (b) autumn, for the period between 2002 and 2010. The corresponding vertical sections, indicated by the zonal black line in (a), for (c) spring and (d) autumn. The contours in (a) and (b) denote the 50 and 250 m isobaths. The black contour in (c) and (d) shows the upwelling isotherm of 22.5°C.

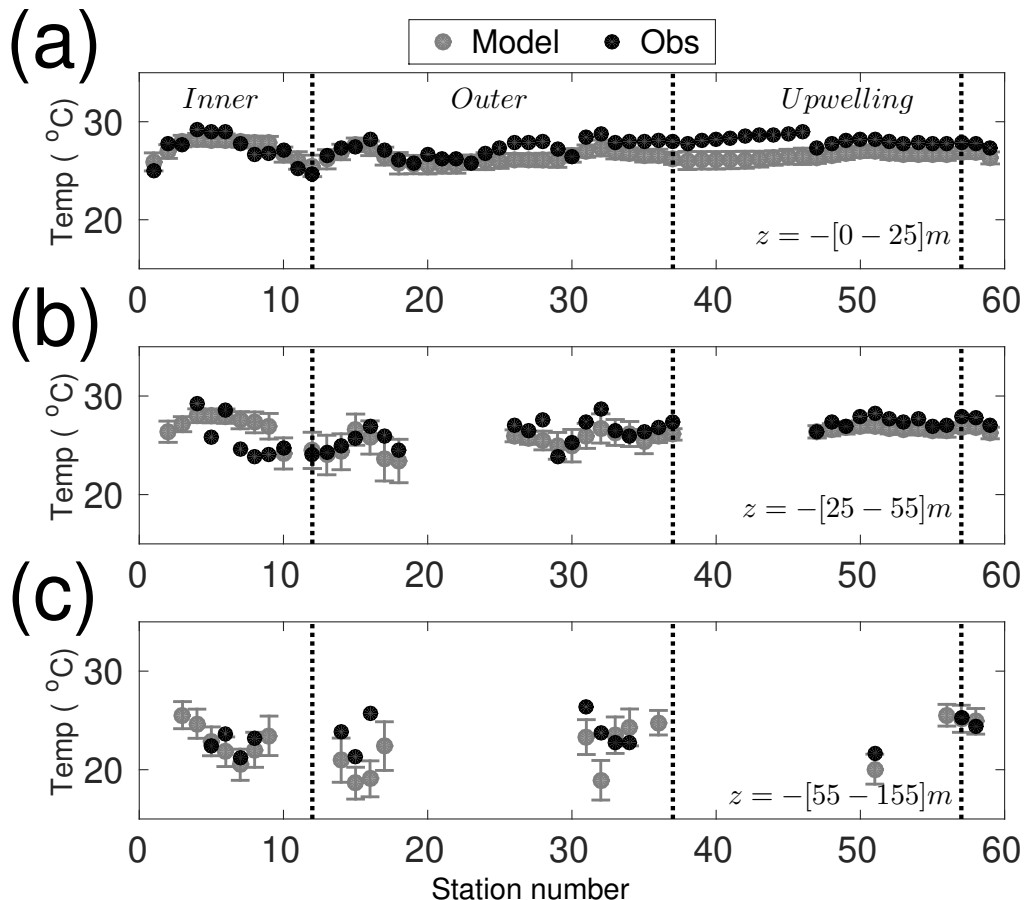

**Figure 4.** Comparison between *in situ* data and simulated temperatures (°C). Temperature values correspond to each hydrographic station, averaged over three depths; (a) between surface and 25 m depth, (b) between 25 and 50 m depth, and (c) between 55 and the deepest measured concentration (z ∼ - 150 m). Black dots correspond to the observed values and open gray circles to the simulation. Vertical gray lines are the temporal standard deviation of the simulated values, as these are temporally averaged over all Novembers from 2002 to 2010. Vertical black lines delimit the group of stations for inner-shelf, outer-shelf and the upwelling area.

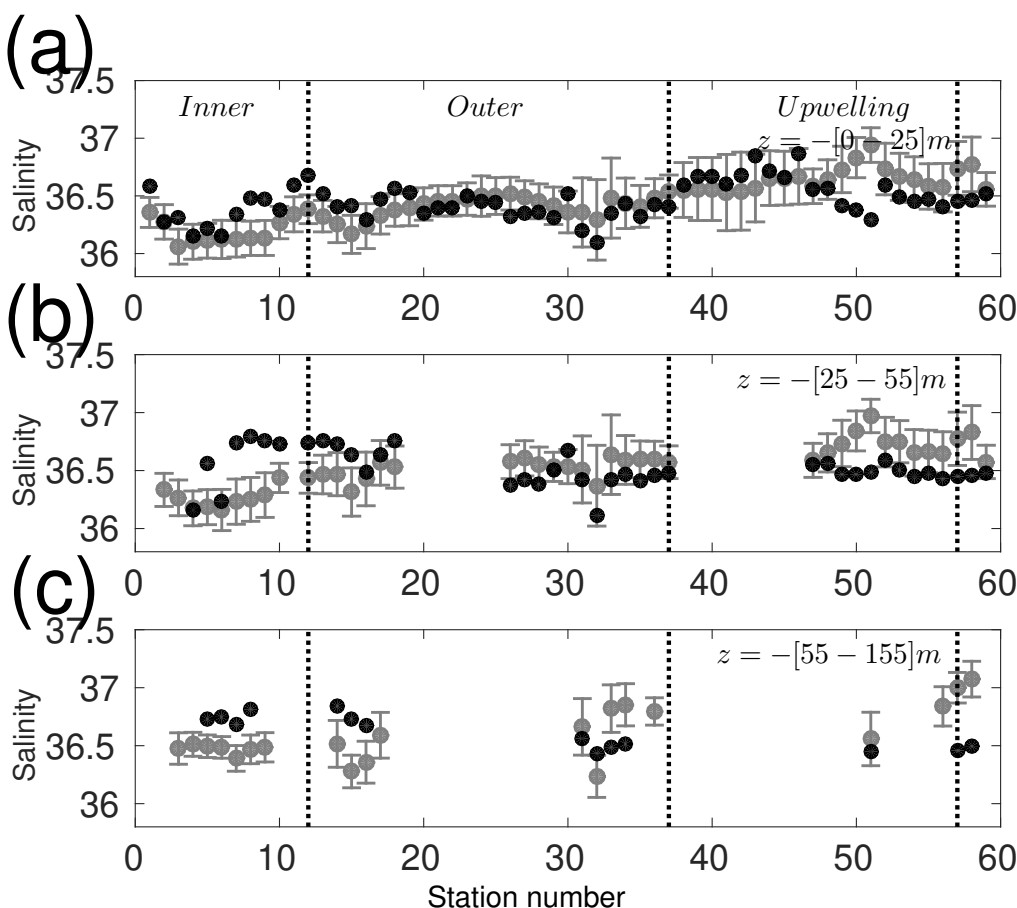

**Figure 5.** Same as Figure 4, but for salinity.

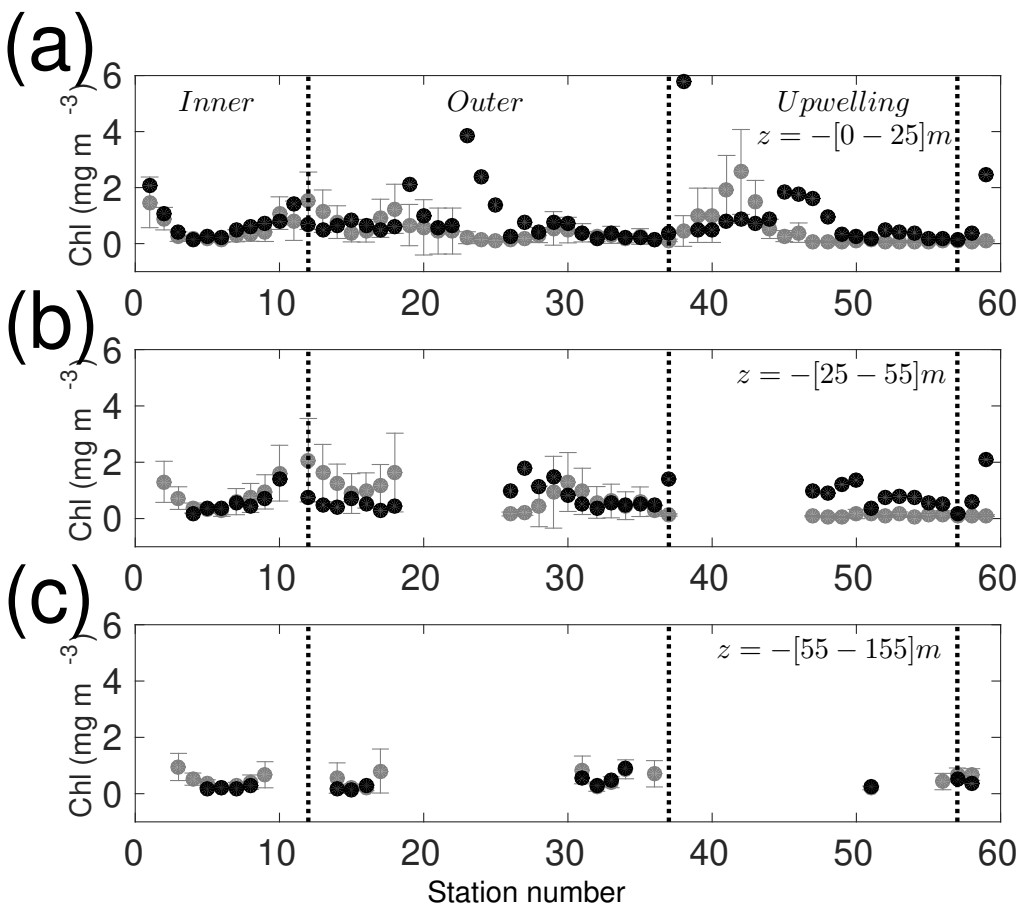

**Figure 6.** Same as Figure 4, but for chlorophyll concentrations (mgChl m$^{-3}$).

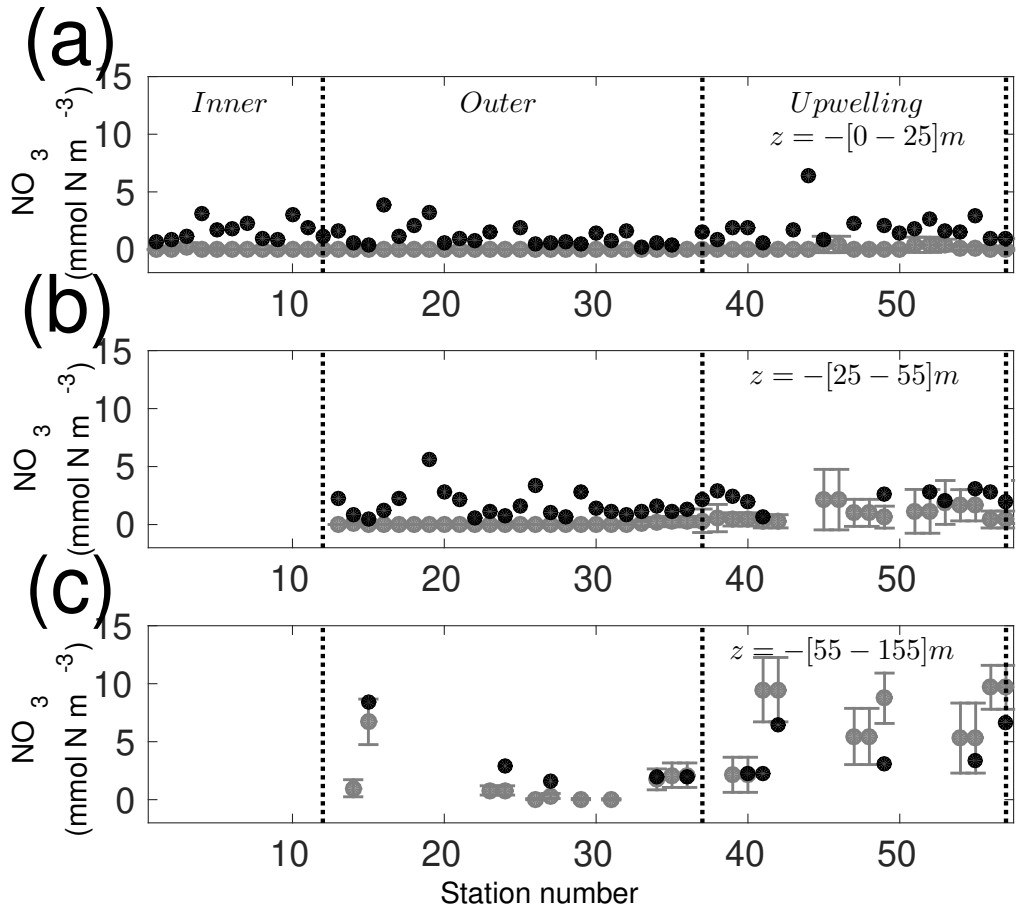

**Figure 7.** Same as Figure 4, but for nitrate concentrations (mmolN m$^{-3}$).

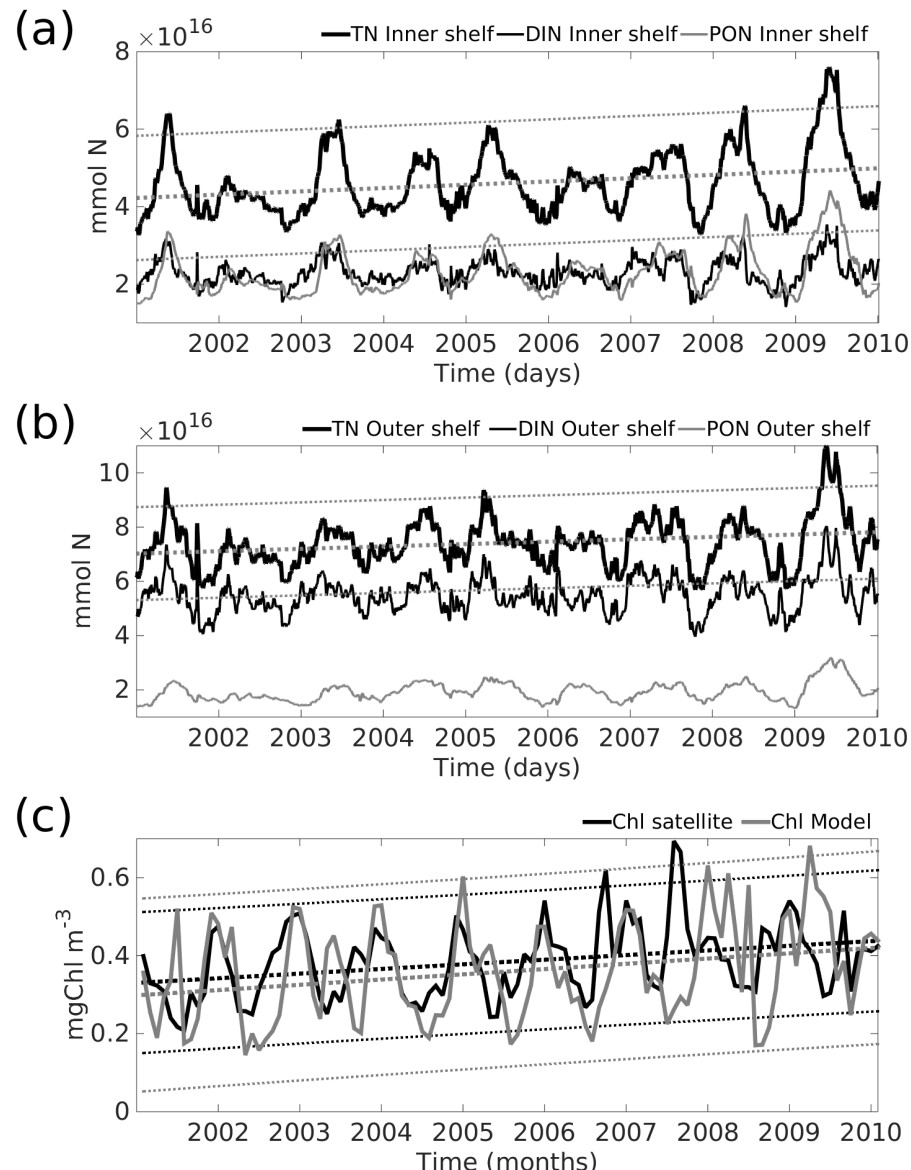

**Figure 8.** Temporal series of TN (thick black line), DIN (thin black line) and DIN (thin gray line) in mmolN, spatially integrated over: (a) the inner shelf, and (b) the outer shelf. (c) are the temporal series from monthly satellite chlorophyll (black, mgChl m$^{-3}$) and from the model outputs (gray) averaged over the whole Yucatan shelf. Dashed thick lines are the trend indicated by the linear fit for the TN or chlorophyll time series, where thiner dashed lines are the respective 95% confidence intervals. Equations of each linear fit are: TN (Inner shelf) = 2.33 $\times 10^{12}$ days + 4.2 $\times 10^{16}$, TN (Outer shelf) = 2.40 $\times 10^{12}$ days + 7.0 $\times 10^{16}$, $Chl$ (satellite) = 0.0010 months + 0.28, and $Chl$ (model) = 0.0010 months + 0.30. Notice that the trend is positive for all the temporal series.

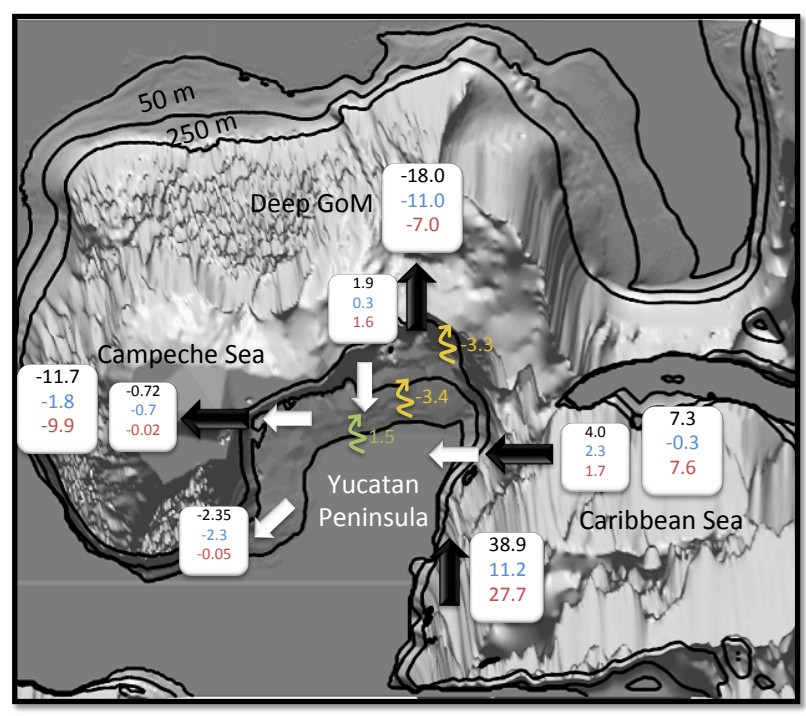

**Figure 9.** Scheme of the TN budget for the Yucatan shelf. Black and gray arrows denote cross-shelf direction flux for the outer and inner shelf, respectively. In blue are the PON; in red the DIN; freshwater DIN sources (Rivers) are in green and sinks of TN due denitrification (DNF) are in yellow. The values are expressed in molN $yr^{-1}$ $\times 10^{10}$. Negative values indicate sink, whereas positive indicates source of TN. The isobaths that delimit the inner (50 m depth) and outer (250 m depth) shelves are also highlighted.

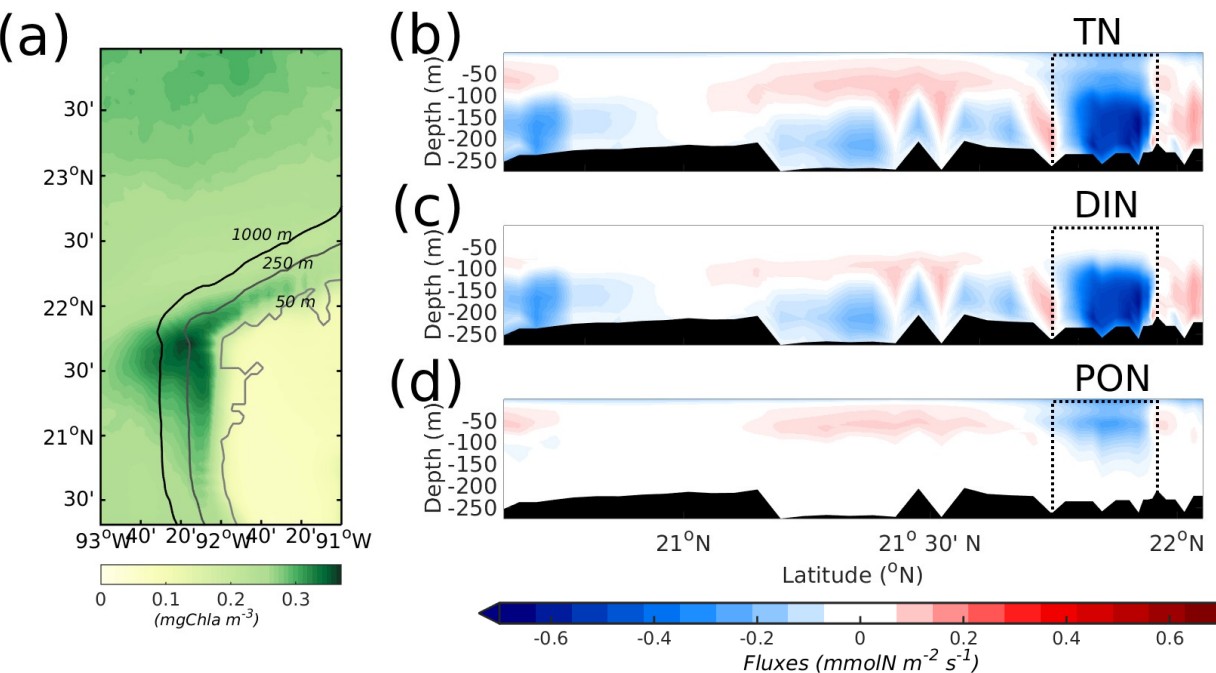

**Figure 10.** (a) Map of surface chlorophyll (mgChl m$^{-3}$), averaged over the nine simulated years. The three characteristic isobaths are denoted. Nine years averaged cross-shelf fluxes along the 250 m isobath at the western boundary of (b) TN, (c) DIN and (d) PON (mmolN m$^{-2}$ s$^{-1}$). Negative values indicate westward flux, i.e., TN flux from the shelf to the Campeche Bay. The area delimited by dashed lines shows the location of the filament depicted in (a), at the NW of the YS.

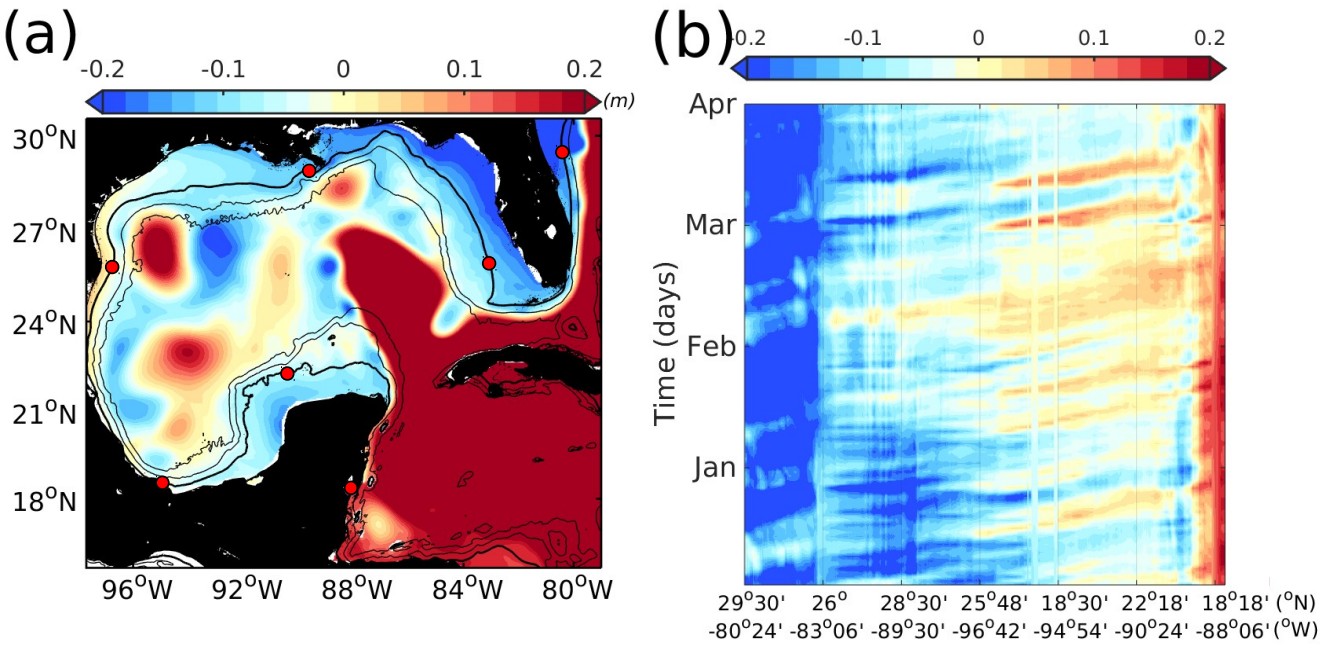

**Figure 11.** (a) Snapshot of sea level anomaly ($\eta$, m) for the simulated year 2005. (b) Hovmöller diagram of $\eta$ along the 50 m isobath from January to April of the 2005 year. Red dots in (a) denote the latitude and longitude shown at the bottom of (b), from Florida to the Yucatan peninsula.

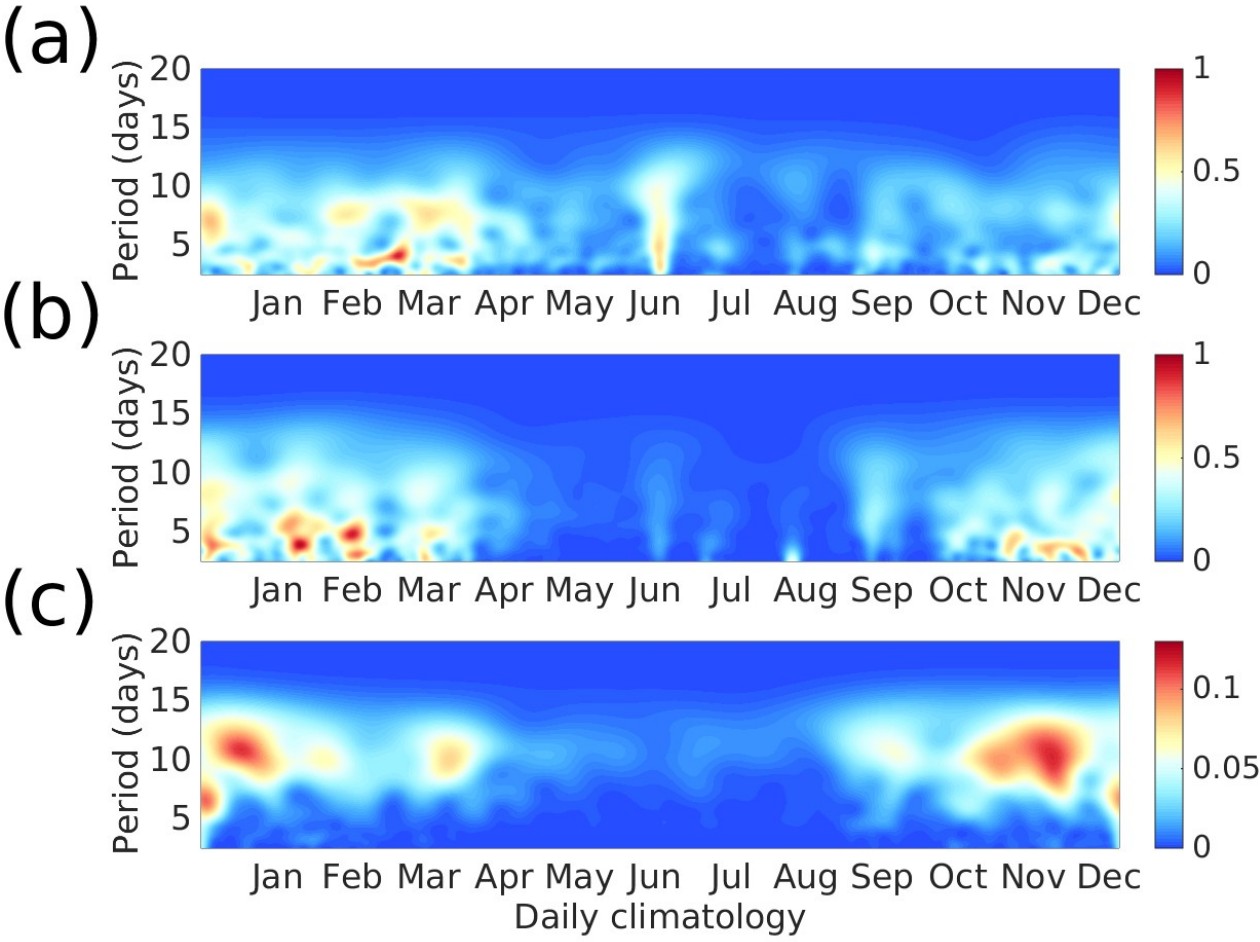

**Figure 12.** Wavelet power spectrum for time series averaged over the western 50 m isobath of: (a) cross-shelf total nitrogen flux vertically integrated (TN, mmolN m$^{-1}$ s$^{-1}$), (b) Along shelf wind stress ($\tau_{along}$, N m$^{-2}$), and (c) Sea level anomaly (SLA, m). The temporal series are detrended, normalized, and filtered by a lancsos high-pass filter with a cut-off of 15 days.

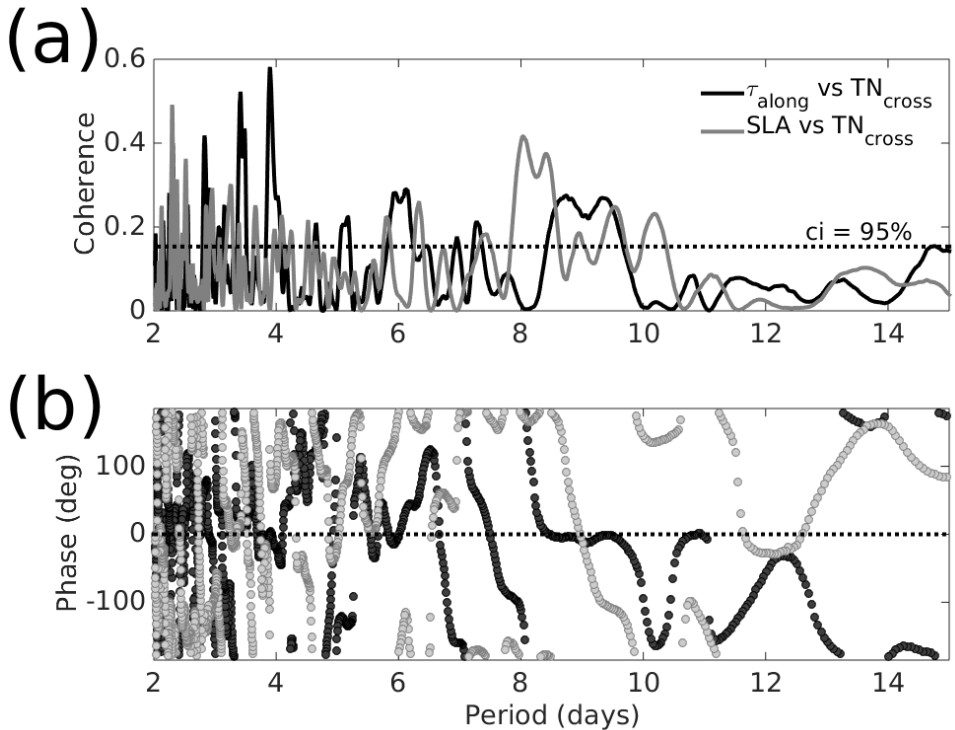

**Figure 13.** (a) Cross-correlation spectral analysis of time series over the western 50 m isobath, indicating the square coherence coefficient between: along shelf wind stress ($\tau_{along}$, N m$^{-2}$) and cross-shelf total nitrogen flux vertically integrated (TN$_{cross}$, mmolN m$^{-1}$ s$^{-1}$) in black; and between Sea level anomaly (SLA, m) and TN$_{cross}$ in gray. The black horizontal line indicates the 95% confidence interval. Analysis for the nine simulated years based on daily outputs with a 30 day window. Before analysis, the temporal series are detrended, normalized, and filtered by a lancsos high-pass filter with a cut-off of 15 days. (b) Shows the phase or anti-phase in degrees of both coherence analysis of (a): $\tau_{along}$ and TN$_{cross}$ in black; and SLA and TN$_{cross}$ in gray.

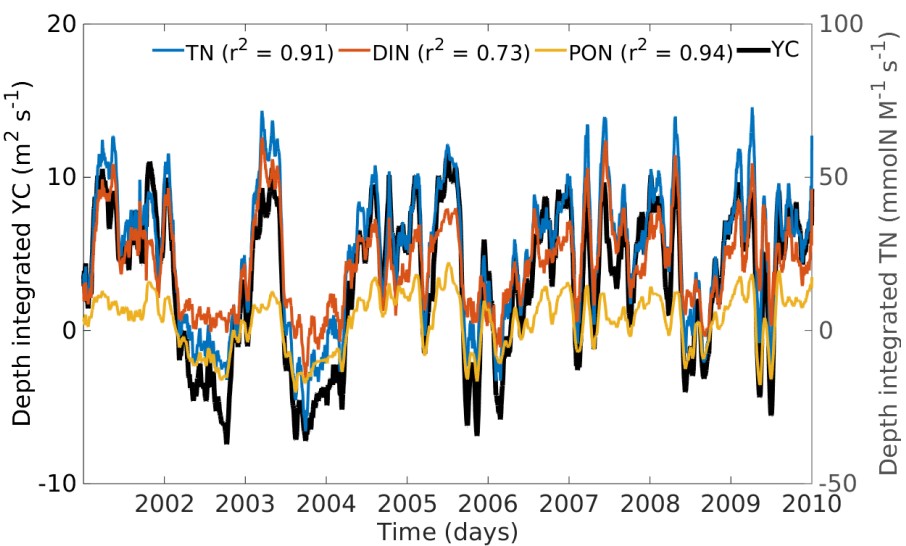

**Figure 14.** Temporal series for the nine simulated years of cross-shelf Yucatan Current component (YC), Total Nitrogen (TN), Dissolved Inorganic Nitrogen (DIN) and, Particulate Organic Nitrogen (PON), vertically integrated and averaged over the isobath 250 m of the eastern boundary. The square of the correlation coefficients ($r^2$), between YC and the biogeochemical variables are shown on top. The temporal series are filtered by a moving average of 30 days to remove daily variability.

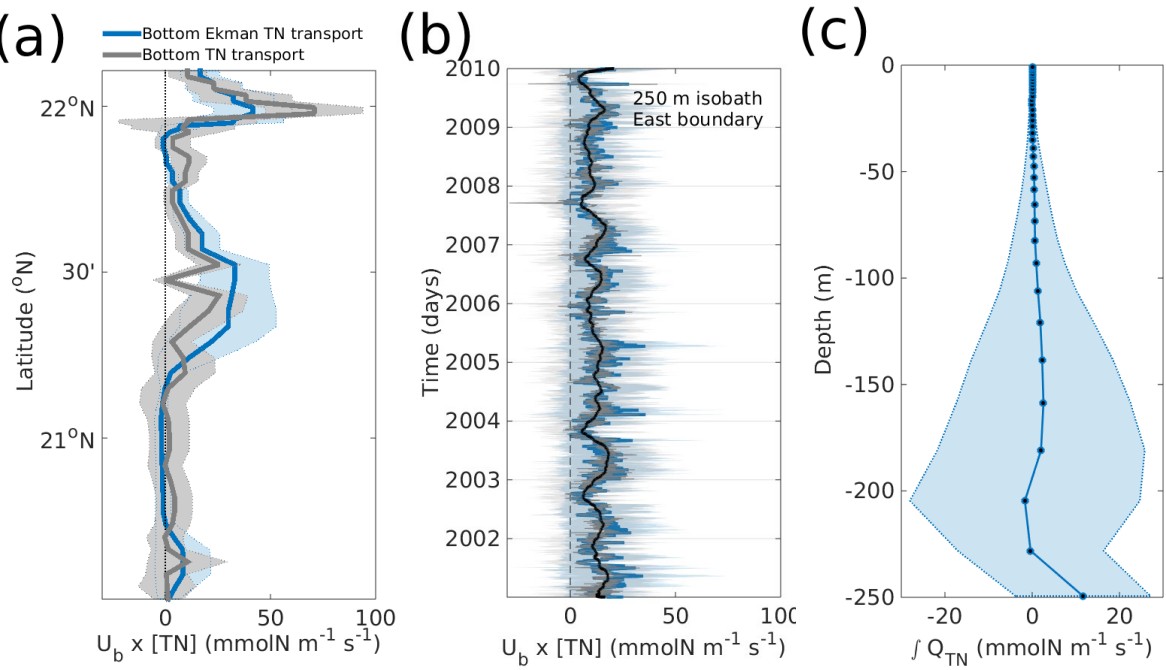

**Figure 15.** Flux of total nitrogen ($Q_{TN}$) computed by the Bottom Ekman transport ($U_{bE}$, m$^2$ s$^{-1}$) for the nine simulated years (blue) compared with the bottom-most layer TN flux (gray, mmolN m$^{-1}$ s$^{-1}$) over the Ekman bottom layer for: (a) temporal averages, and (b) spatial averages over the 250 m isobath, where superimposed black line is the bottom Ekman transport filtered with a 90 day moving average. (c) Vertically integrated TN flux along the eastern 250 m isobath, averaged over latitude and over the nine simulated years in mmolN m$^{-1}$ s$^{-1}$. Shaded areas denote the standard deviation of the averages.

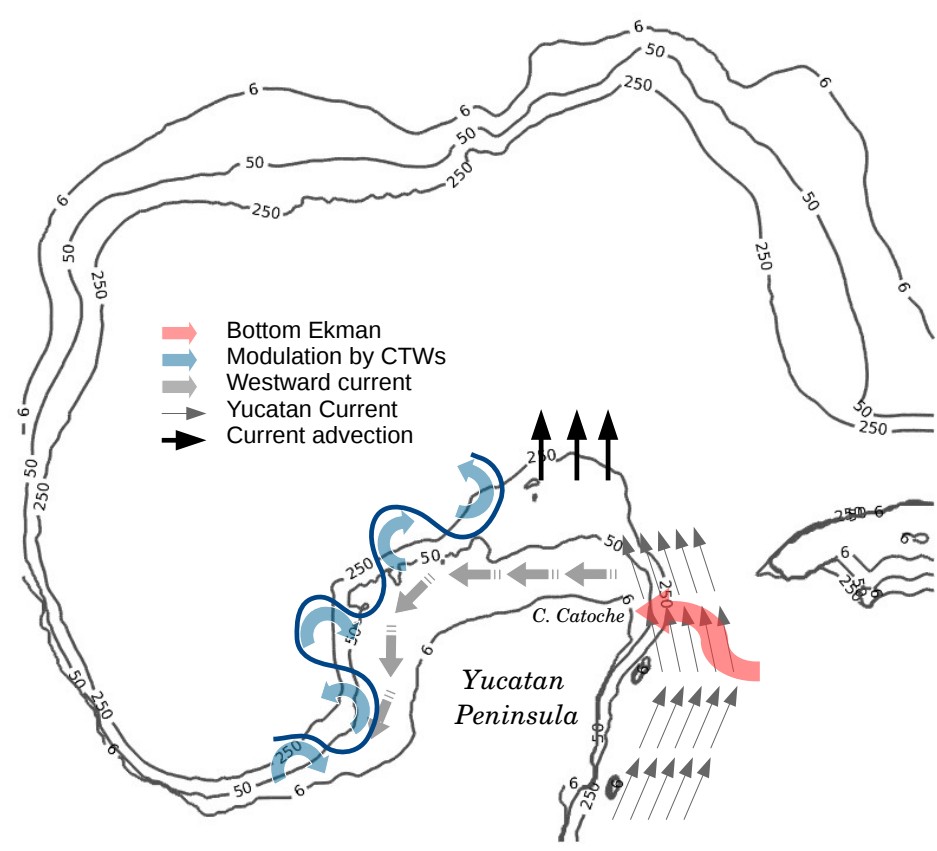

**Figure 16.** Schematic view of the main physical processes that modulate the cross-shelf transport of TN in the Yucatan shelf.

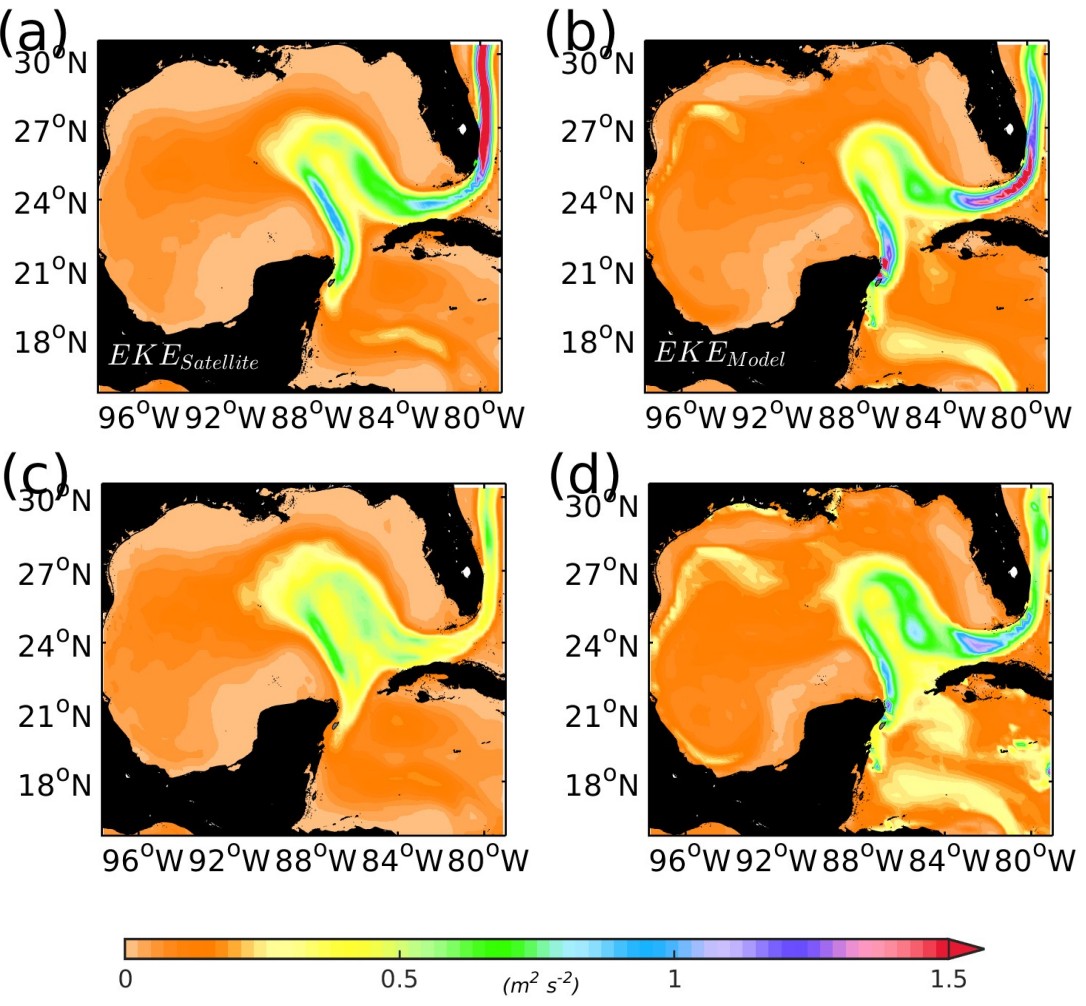

**Figure A1.** Comparison of 17 yr (1995-2012) averaged Eddy Kinetic Energy (EKE, $m^2$ $s^{-2}$) calculated in base on (a) AVISO SSH product and (b) ROMS model simulated SSH. (c) and (d) are the standard deviation for altimeter and model EKE, respectively.

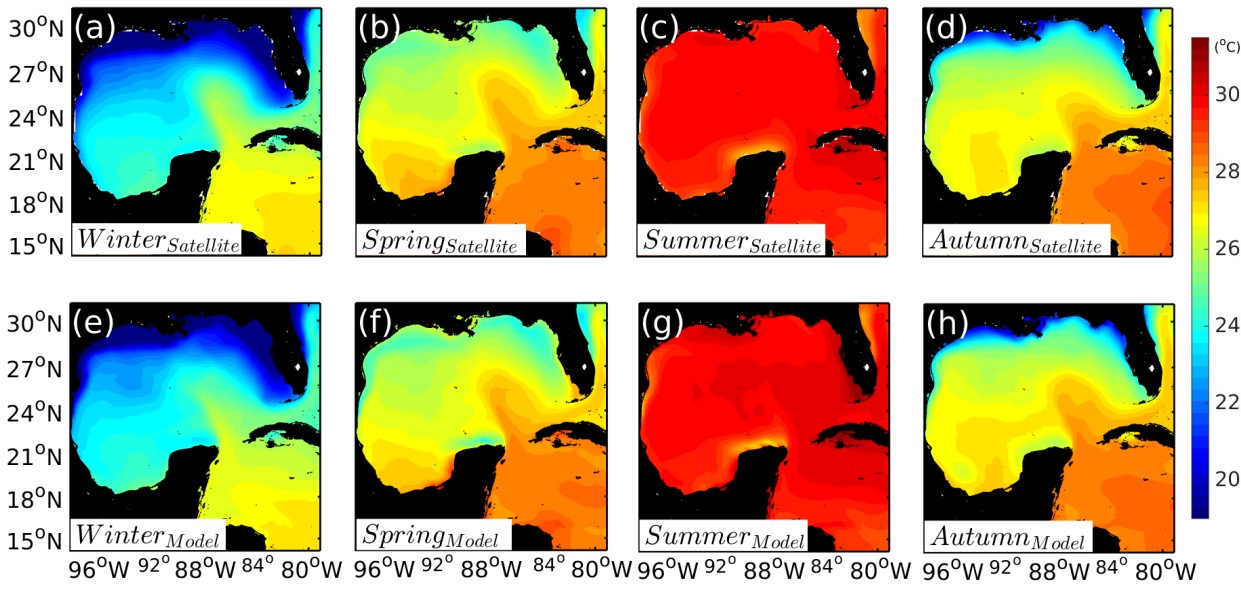

**Figure A2.** Seasonal climatologies of SST (°C) for the GoM (2005 to 2012). Comparison between (a-d) satellital SST product and (e-h) model SST.

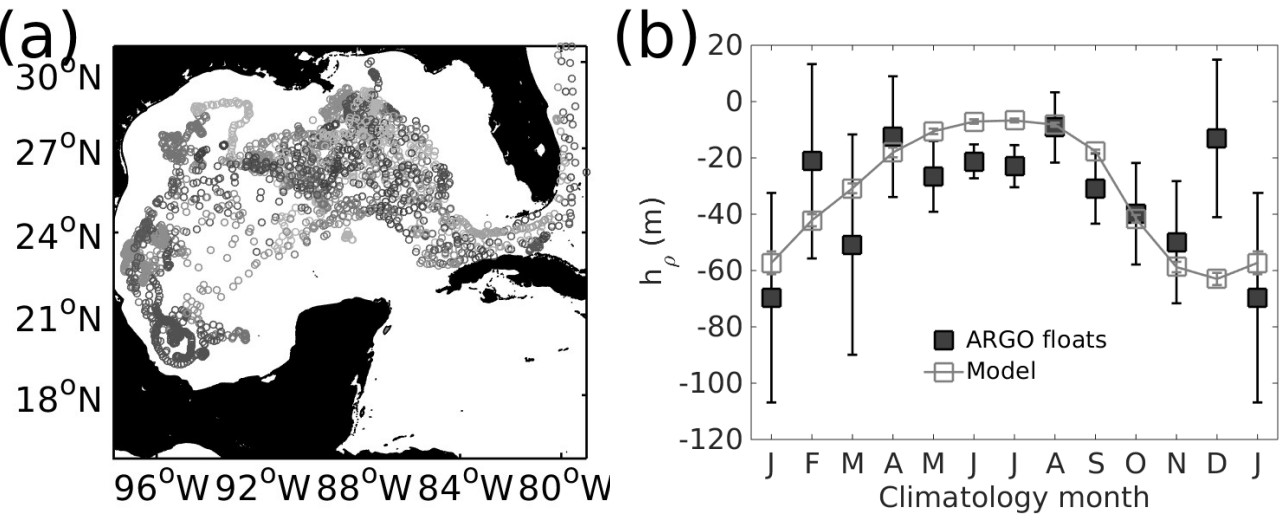

**Figure A3.** (a) Location of the 2629 ARGO profiles used to compute the mixed layer depth ($h_\rho$, m). (b) Climatology comparison of mixed layer depths for ARGO profile floats (black boxes) and the model (gray boxes). Vertical lines in the boxes denote standard deviation.

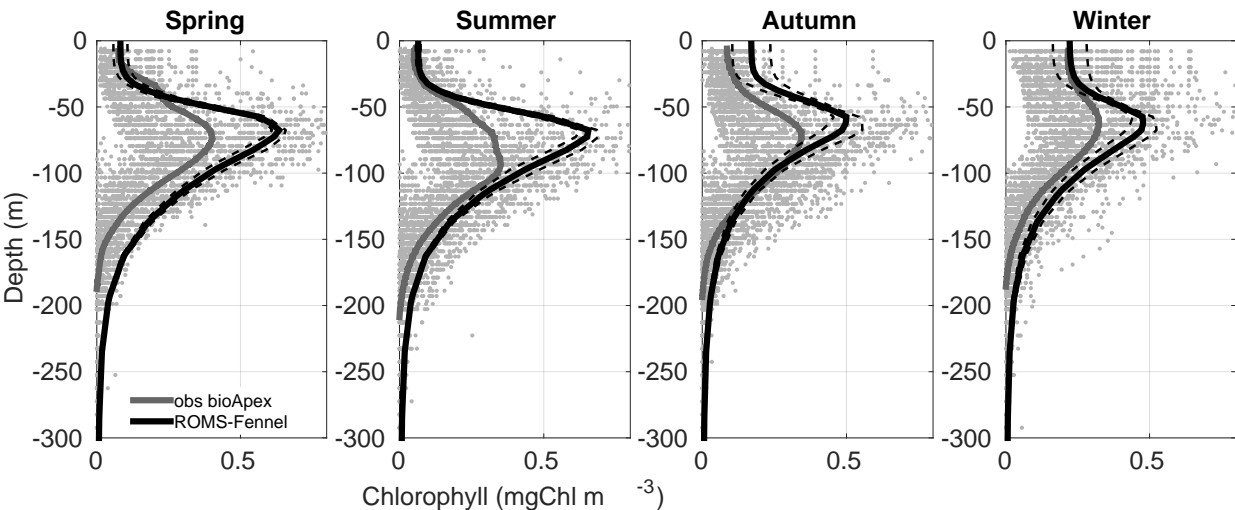

**Figure A4.** Seasonal comparison of chlorophyll profiles in mgChl m$^{-3}$, taking all the available Apex floats (Pasqueron de Fommervault et al., 2015), in order to evaluate the Deep Chlorophyll Maximum (DCM). Grey dots are the data observed from Apex floats; the average profile is shown in grey. In black is the averaged profile of the model data with its respective standard deviation in dashed black lines.

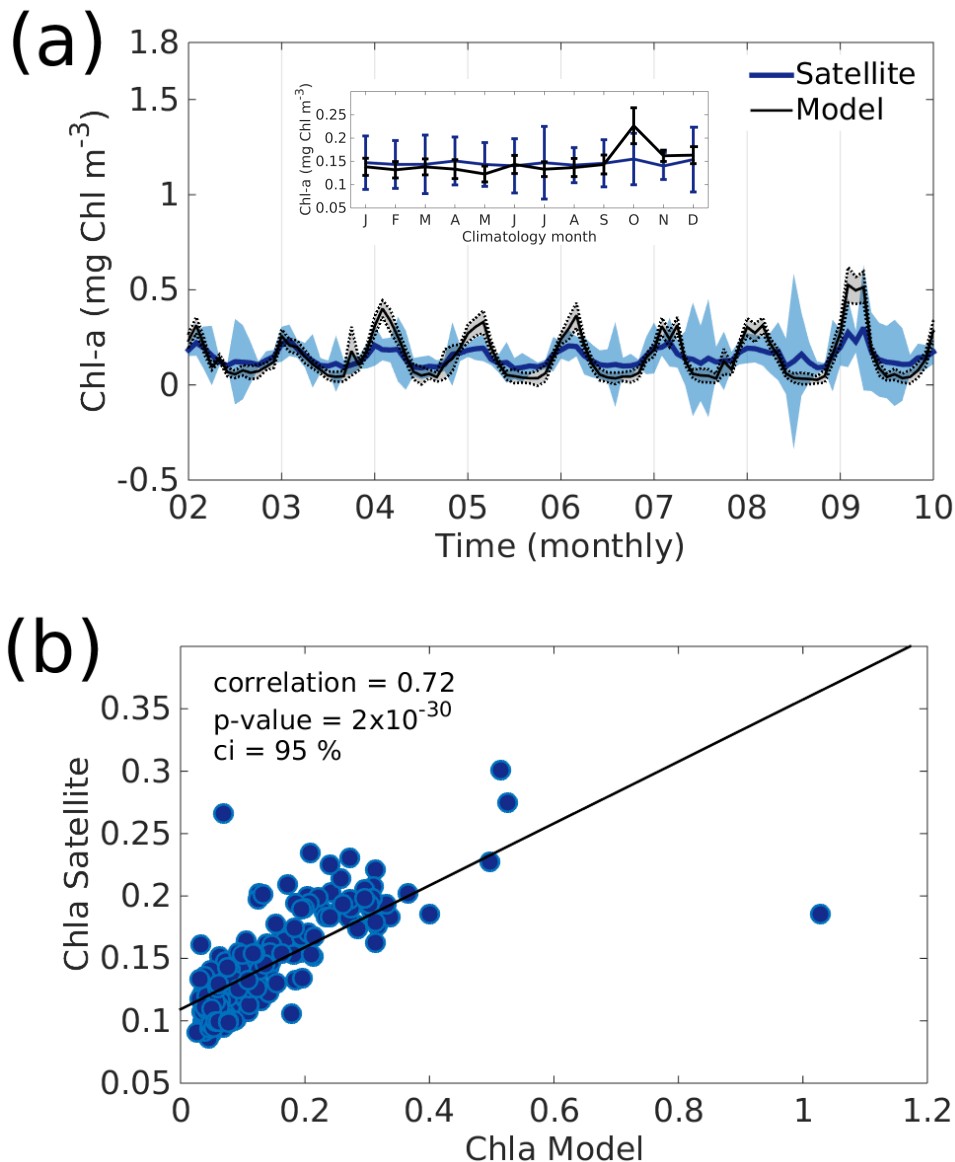

**Figure A5.** (a) Temporal series of surface chlorophyll in mgChl m$^{-3}$ from satellite and model for the whole deep GoM. Standard deviations from the spatial averages are shown in shadow blue areas for satellite and dashed black lines for the model. The monthly climatology of the temporal series is shown at the upper part of the figure, where vertical bars indicate standard deviation from the temporal mean. In (b) are represented the correlation coefficient of both monthly temporal series and their respective linear fit in black line. The slope of the linear fit is 0.25.

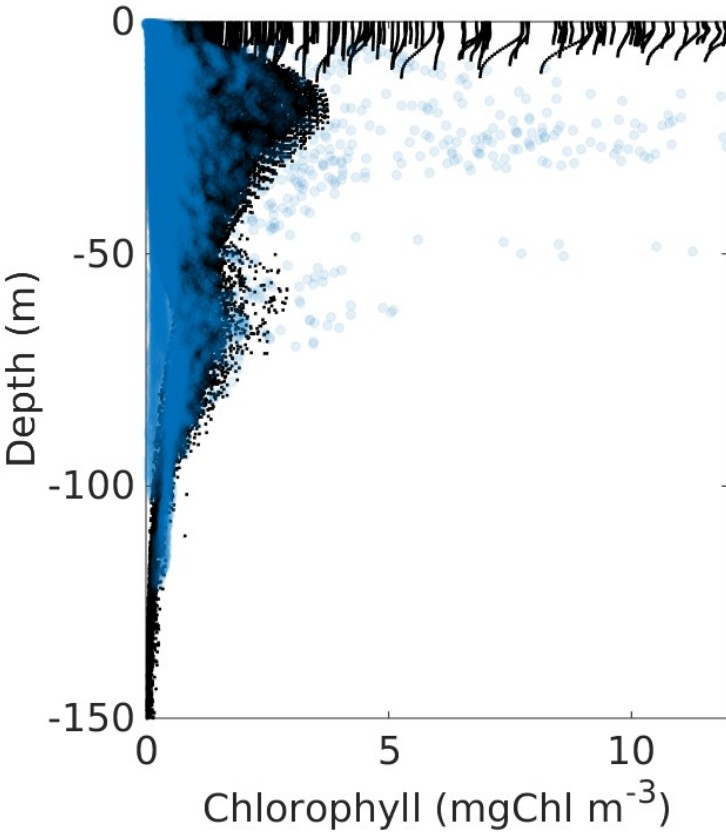

**Figure A6.** Profiles of chlorophyll (mg Chl m$^{-3}$). In black are the model profiles temporally averaged for all the July, August and November months of the nine simulated years. Superimposed in blue are the observed profiles of the three GOMEX cruises carried out during November 2015, August 2016 and July 2018.

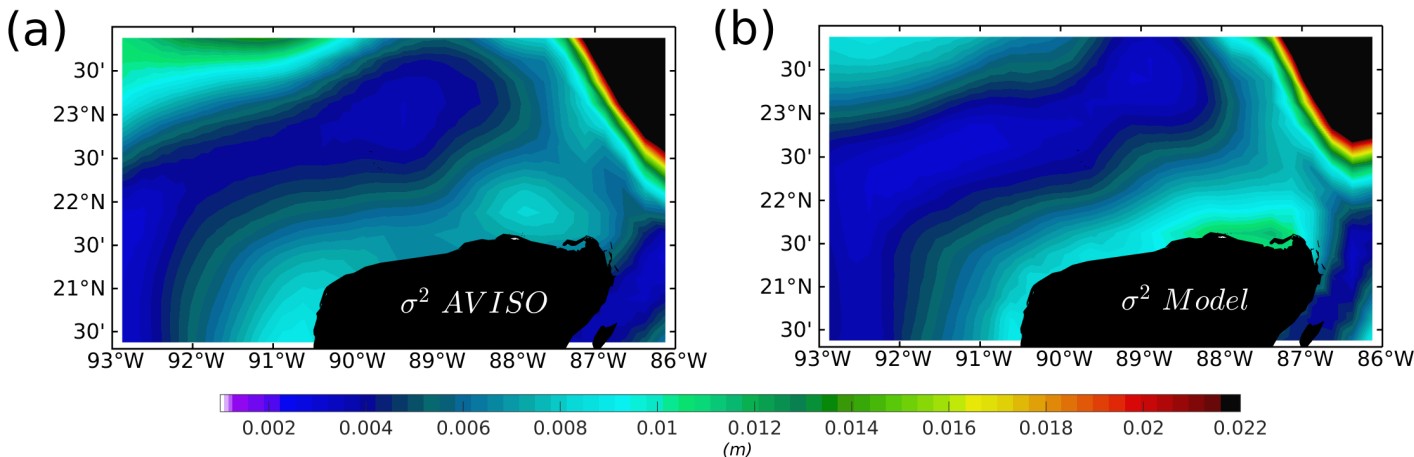

**Figure A7.** Variance ($\sigma^2$, m) for the range of years 2002-2010 of: (a) Absolute dynamic topography (ADT) extracted from the AVISO altimeter product, and (b) Mean sea level ($\eta$) from the ROMS outputs.

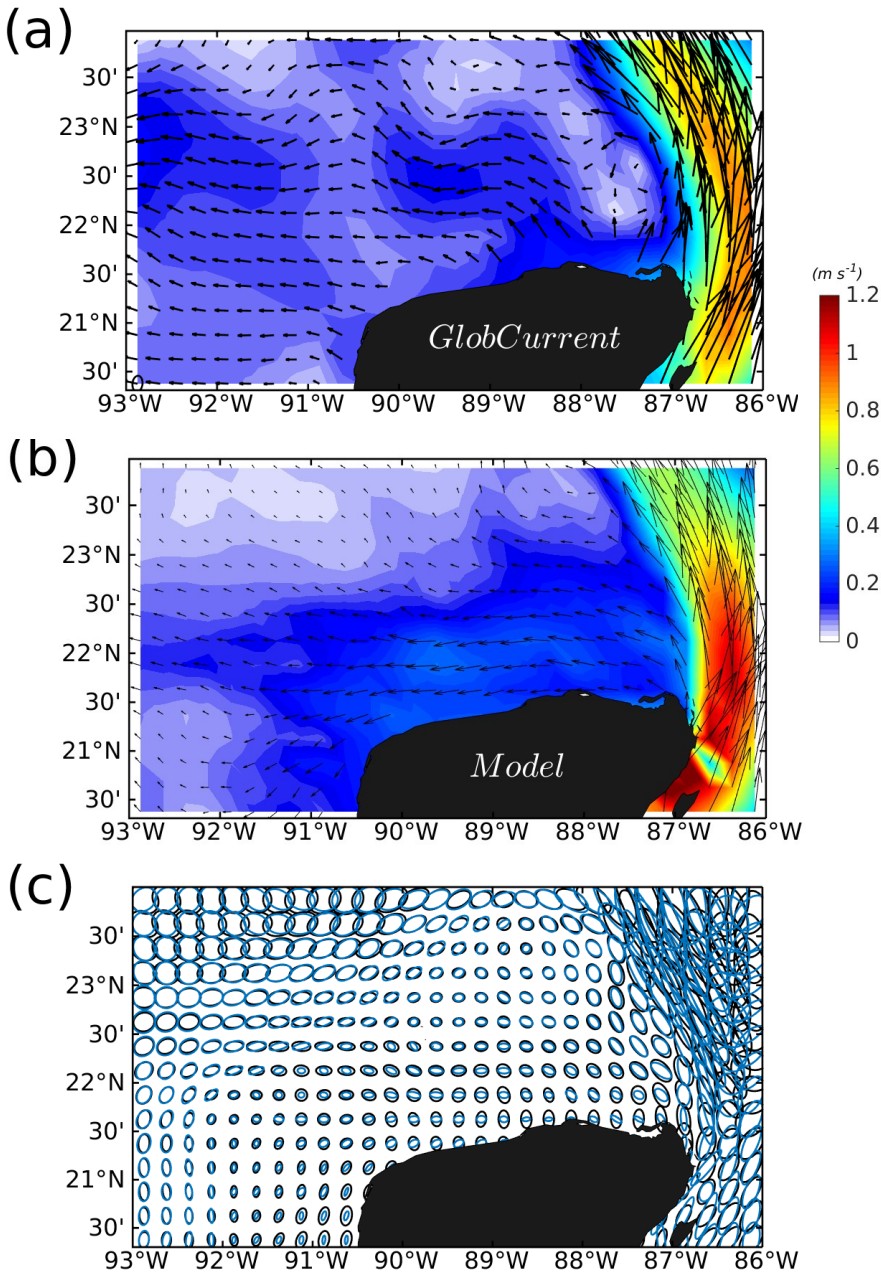

**Figure A8.** Mean current magnitude and velocity vectors averaged for the years 2002 to 2010 in m s$^{-1}$. (a) Mean geostrophic plus Ekman currents from GlobCurrent product; (b) Mean total current from the model; (c) velocity variability ellipses of the GlobCurrent product (black) and model outputs (blue).

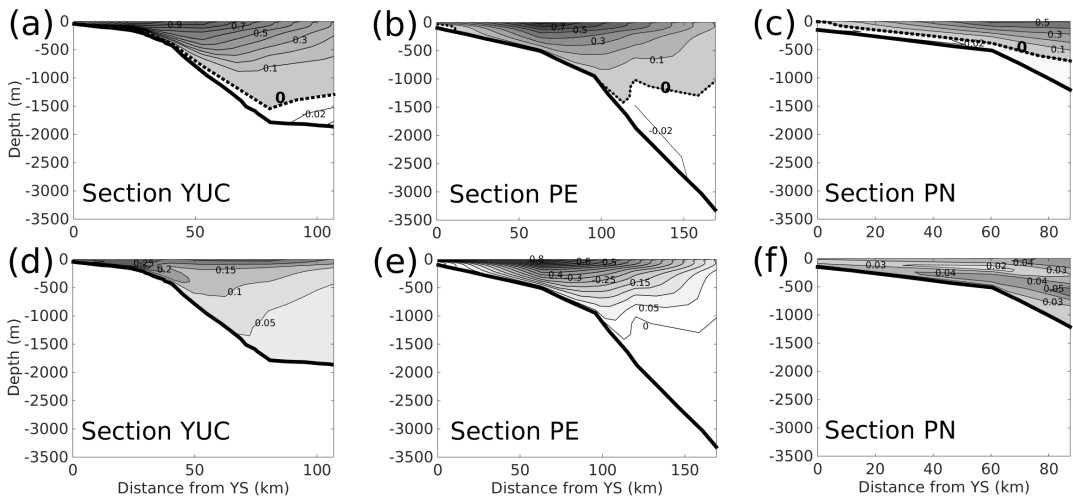

**Figure A9.** Mean model velocity (m s$^{-1}$) component normal to the three sections: (a) YUC; (b) PE; and (c) PN, depicted in Figure 1a, for the years 2009 to 2011, and to be compared with Sheinbaum et al. (2016). Positive velocities are in gray (contours every 0.1 m s$^{-1}$) and negative velocities in white (contours every 0.03 m s$^{-1}$); dashed black contour indicates zero velocity. (d), (e) and (f) shows the standard deviations for each transect (contours every 0.05 m s$^{-1}$ for d and e, and every 0.01 m s$^{-1}$ for f).

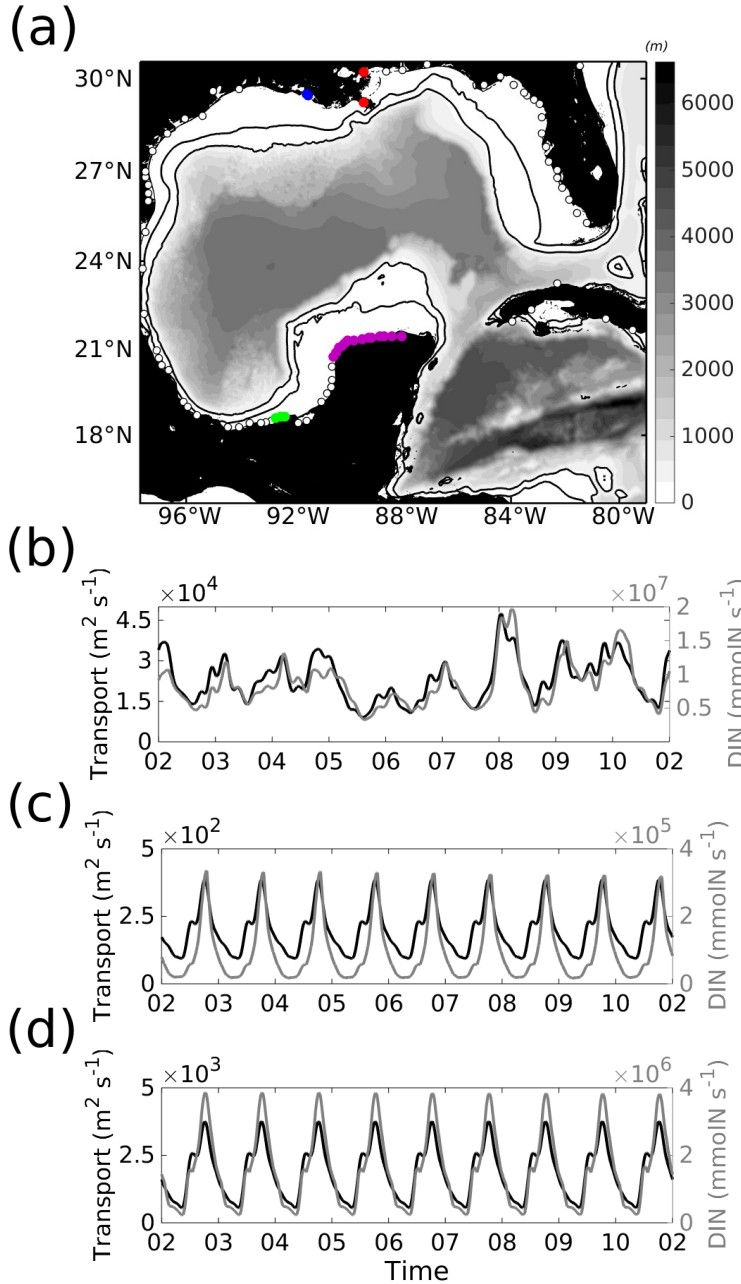

**Figure A10.** (a) Model bathymetry with the location of the input freshwater sources (white points). In red and blue points are the Mississippi and Atchafalaya riverine system, in green is the Usumacinta and Grijalva riverine system and in violet are the freshwater system of the YS. The panels below show the temporal series of water transport ($m^3$ s) and the DIN ($NO_3$ + $NH_4$) fluxes (mmolN s$^{-1}$) for the systems: (b) Mississippi-Atchafalaya; (c) Usumacinta-Grijalva; and (d) Yucatan shelf freshwater.