# Peer review of "Budget of the total nitrogen in the Yucatan Shelf: driving mechanisms through a physical-biogeochemical coupled model"

_Biogeosciences, 2019_

## Referee Comment (RC1) · Anonymous Referee #1 · 3 Jun 2019

Review of Estrada-Ellis et al. "Budget of the total nitrogen in the Yucatan Shelf: driving mechanisms through a physical-biogeochemical coupled model"

Overview:
The authors present an analysis of the total nitrogen budget on the Yucatan shelf as influenced by physical transports, mixing, river inputs, and biogeochemical processes. A coupled physical-biogeochemical model was used to quantify the processes driving nitrogen source and sink terms. On the eastern boundary of the Yucatan shelf, the Yucatan Current is the dominant flux. Bottom Ekman transport towards the shelf is also important in this area. On the western and northwestern boundary, coastal trapped waves drive exchanges with the open Gulf of Mexico. A westward current on the inner shelf results in N exports at the western boundary of the inner shelf. The results of this work are interesting because the Yucatan shelf has been poorly studied and because the geographic setting provides an interesting interplay of different physical processes that are overlaid on one another.

General Comments:
Unfortunately, I cannot recommend publication of the manuscript at this time for three reasons. First, I found the validation of the physical model insufficient. If there are any measurements of ocean currents for the Yucatan shelf, these should be presented and discussed to evaluate the accuracy of the modeled physical transports. If there are no data for currents, which may be likely especially outside the Yucatan current, the model could be validated by presenting comparisons of modeled versus observed salinity data. Second, I find that the manuscript lacks a discussion of how model bias in physical and biogeochemical state variables may influence results and there is no presentation of uncertainty estimates for the calculated budget source and sink terms, which makes it impossible to compare magnitudes of these terms. Third, the sink and source values were not presented as normalized to a unit area (e.g. m2) and thus the results from this study cannot be compared to results from previous N budget work in the Gulf of Mexico or elsewhere.

Specific Comments:
1. Abstract, L11: Is there a reason for choosing the 250 m isobath as the shelf boundary?

Page 1,
2. The first paragraph of the Introduction needs to be rewritten. References should be updated with recent relevant work on shelf carbon and nitrogen budgets. The last sentence in this paragraph incorrectly lists acidification and eutrophication as socio-economic activities. These processes may result from socio-economic activities but are not activities in themselves. Likewise, the processes listed as part of the climate system are not ones that would immediately come to mind. Please rewrite.

Page 2,
3. L3: Probably should cite Walsh et al. 1989 here

4. L12-13: It would be good to provide more detail about these controversies to help the reader understand the motivation for this study.

5. L26: Show the Yucatan Current in Figure 1.

Page 3,
6. L23: Change "was ran" to "was run"

Page 4,
7. L12: Suggest deleting the first sentence and starting the paragraph with something like "The XIXIMI cruises provided profiles of nutrients and …

8. L22: Does the SDet equate to dissolved organic nitrogen (DON) in the model? In the real world, the components of total nitrogen are DIN, DON, and PON (or PN since there's some adsorbed inorganic nitrogen on particles). Dissolved organic nitrogen is often equal to or greater than dissolved inorganic nitrogen in the coastal ocean and in coastal rivers. If the SDet does not equate to DON, then your TN definition is incorrect. If SDet does equate to DON, then the assumption of setting PON in rivers equal to 0.1 mmol N m-3 (see comment 11 below) is incorrect.

9. L26 and equation (2): This equation is only for the water-column. The total nitrogen budget also includes the loss to denitrification and to burial in the sediments. Please clarify.

Page 5,
10. L16-20: More details are required about how the freshwater inputs were calculated. Since the freshwater inputs are unknown, it would be justified to include a time-series figure of these inputs, perhaps in the appendix.

11. L23: Setting the PON to this small value is not justified. I suspect that PON must include DON, else the definition of TN used in this study is incorrect. DON concentrations are generally >> 0.1 mmol N m-3.

12. L26: Provide dates for the November cruise.

13. Section 3.1 seems like it should be in the appendix with the other basin wide modeling results. These results aren't really germane to the analysis except as boundary conditions to the shelf.

Page 6,
14. L20: Why is there no model comparison with salinity data? This should be included to provide confidence the model is accurately representing physical transports.

Page 7,
15. L3-9: Poorly worded paragraph. The explanation of why the model results cannot be compared with other results is incorrect.

The results from this study should be compared to other studies to put the overall budget for the Yucatan shelf into some context in comparison to other more well-studied shelves in the Gulf

such as the West Florida and Louisiana shelves. I recommend normalizing your budget fluxes to area so that they are comparable to other flux estimates.

16. L15: The trend is mentioned here but there's no explanation. Is it real? What is driving the trend? What source/sink terms have changed? The model is deterministic so there's no reason not to get to the bottom of this, especially since the trend suggests that the N budget is not at steady state.

17. L19: I'm not sure what you mean by "a very efficient biological cycle". Please be more specific.

18. L16-17: This logic doesn't make sense to me. Earlier in the ms it was stated that the chlorophyll time series were used in an inverse analysis to prescribe freshwater and N inputs (also see comment 10). Thus, the TN trend and the chlorophyll trend may not really be independent. Please address whether these are completely independent variables.

Page 8,
19. L22: Please report the rates of denitrification (mmol N m$^{-2}$ d$^{-1}$ or something similar) obtained from the model.

20. L24: Fennel et al. (2006) was a study of the Mid-Atlantic and did not address GoM shelves.

Page 9,
21. L29-: This paragraph should be deleted. The last sentence makes it clear that the present analysis cannot address these phenomena.

Page 10,
22. L30: Insert "to" after "due"

23. L31: Change "show" to "shows"

Page 11,
24. L13: Is "2015" a typo?

25. L28: Delete "the" before "unique"

26. Prior to Concluding Remarks there needs to be a discussion of the uncertainties in your budget analysis. How does model bias for N concentrations affect your budget? What is the error (standard deviation of the mean) of each term in the mean budget? Without including this, there is no way to make meaningful judgements about the magnitude of the budget terms.

Page 12,
27. L1-2: Figure 15 shows the physical system but not the biogeochemical system.

28. Table 1: Normalizing the fluxes to a unit area would be more meaningful since the flux estimates presented are driven by the length of the boundaries and the area of the inner and outer shelf.

29. Figure 1: These maps use degrees-minutes whereas other maps use decimal degrees. Be consistent. On Figure 1, the grey contours are difficult to see in panel (a). In panel (b), the vectors are too small to be seen in my copy.

30. Figure 2: It is hard to see the dashed boxes in my copy. Note the isobaths again in this figure caption so the reader knows what these lines are.

31. Figure 3: Should be in appendix with basin-wide results.

32. Figure 4: Should be in appendix. In panel (a), the shadow and dashed line are difficult to differentiate. In panel (b), report the slope of the linear fit.

33. Figures 5, 6, 7: Report model evaluation statistics such as bias and RMSE. The bias in temperature, NO3, and chlorophyll is generally positive with model results being greater than observations. How does this affect the TN budget calculated with the model?

34. Figure 8: For panels a, c, and e report the p-values for the trend lines. The legends are confusing. Perhaps rename them to Inner Shelf TN, Inner Shelf DIN, Inner Shelf PON, etc.

35. Figure 9: There are differences in values and significant digits presented here and Table 1. Double check these values and make corrections. Also some numerical values for fluxes are difficult to read. A simple 2-D map may make a better figure. Plus, the upside down (S-N) orientation is odd for the 3-D figure.

36. Figure 10: Font is too small for gray depths. Latitude is shown in decimal degrees here. "Isobtahs" is misspelled in the caption.

37. Figure 11: I can't see the red dot for the station at Lat = 18.3 and Long = -88.1.

38. Figure 12: Is this figure necessary? It seems to just show a correlation between currents and SLA that could likely be seen with a simple correlation analysis. What is the unit cpd-1 in the y-axis labels?

39. Figure 13: Difficult to see isobaths in panel (a). In panels b, c, and d, change the blue lines to black to match the y-axis label or change the y-axis label to blue.

I did not check references.

END OF REVIEW

---

## Referee Comment (RC2) · Javier Zavala-Garay (Referee) · 4 Jun 2019

*Review of the manuscript* **"Budget of the total nitrogen in the Yucatan Shelf: driving mechanisms through a physical-biogeochemical coupled model"** *by Sheila N. Estrada-Allis et al.*

**General comments**

This work presents an estimation of the Total Nitrogen (TN) budget in the Yucatan Shelf (YS). The estimate is obtained using a coupled physical-biochemical model (ROMS), validated by some in-situ and satellite observations. The model solution is available for 9 year (2002-2010) while the in-situ observations used to validate the solution within the YS are available for Nov 2015. Physical processes that are relevant in explaining the estimated TN budget are identified and described. The main input of N is at the eastern boundary through the interaction of the western boundary current with the shelfbreak, presumably mainly due to Ekman transport at the bottom boundary layer. The imported N is then advected westward by the wind driven-circulation along the shelf. Most the N that enters the inner shelf (depths shallower than 50 m) is consumed by phytoplankton, and part of the N that enters the outer shelf (depths 50-250 m) is exported to the deep ocean in the west and northwest parts of the YS. This export of N is modulated by Coastally Trapped Waves with a typical period of ~10 days.

I think this manuscript addresses a relevant scientific question within the scope of BG, and the modeling results suggest a very interesting case for the relevance of likely physical processes controlling or modulating the import and export of N in and out of the YS. However, I think some revisions are needed in terms of validating the model, justifying the model physics or at least acknowledging the limitations and implications for the estimated budget, and the analysis and overall presentation of the work done. Below I outline some suggested revisions that might guide the authors when improving the manuscript.

**Specific Comments**

Model validation:

While this study focuses on the YS and its vicinity, most of the convincing validation presented is for the whole Gulf of Mexico (GoM). There is a need to validate and characterize the background state in the YS in order to add credibility to the results. While I understand there is a scarcity of in-situ observations in the YS, some vertical sections of temperature and salinity have been reported previously (e. g., Enriquez et al., 2013). There are also lots of in-situ observations in the western boundary current (e.g., Sheinbaum et al., 2002 and 2016) that could be used to convince the reader that the model physics is reliable. Specially those closer to the 250 m isobath if available. There is no need to do an exhaustive analysis of such observational datasets, but to show a congruence between model and observed mean background state.

The in-situ biochemical observations were taken in Nov 2015, while the model solution is available until 2012. Ideally one would have a solution contemporaneous with the observations but if that is not possible then the approach presented is the best next choice: compare the mean and standard deviation for Nov with the observations. But then one wonders is the agreement with observations will hold for other months given that the authors present lon-term means for the budgets.  The authors seem to suggest that seasonality is weak, but I think that needs to be shown. For instance, does the inner shelf remains well mixed throughout the year as it seems to be the case in Nov?, or does some stratification develops during summer and if so, how that affects the budget?.

Similarly, more analysis can be done using satellite products to validate the circulation in the YS. This is specially important to compensate for the scarcity of in-situ observations. How does the observed Sea Level Anomaly correlates with that of the model in the YS? The authors could for instance compare the annual cycles of the aviso SLA and the model (without Mean Dynamic Topography) to convince the readers that the background model physics on the shelf is reliable. Is the pressure gradient across the shelf break well resolved by the model?

A similar comparison could be done with scatterometer winds. The use of OSCAR currents, while very low resolution, might be possible given the wide YS.

SST is not a robust validation set since the authors are using bulk fluxes at the surface and the CFSR model (I assume) is forced by satellite SST. Therefore the model is implicitly nudged to observed SST via the 2 m air temperature.

Model physics:

One the main findings reported in the manuscript is that the input of N in the southeast part of the shelf comes from Ekman transport at the bottom boundary layer. The vertical stretching used was developed to study the surface vorticity balance and while this might be a wise choice for the whole GoM, it is not the best for the YS NT budget given the relevance of the bottom boundary layer. This limitation needs to be further analyzed and acknowledged in the manuscript. The authors could for instance estimate the bottom boundary layer thickness (a common estimate is thickness = 0.4*frictional_velocity/Corilis_paramater, where frictional velocity is estimated from the bottom stress). How this thickness compares to the thickness the the first model sigma layer for the 250 m isobath?. That is, how well is the bottom boundary layer resolved?. If it is not very well resolved I still think the bulk characteristics of the budget will hold (i.e., tha main sources and sinks and the TN pathways), but this limitation needs to be acknowledged.

**Technical Corrections**

I noticed that the quality of the manuscript in terms of typos, clarity of the statements, grammar, etc. degrades towards the end. Please revise it carefully. Below is a list of some of them.

(L = Line, P = Page)

Abstract:

L4: I think it should be "Coastal-Trapped Waves" or "Coastally Trapped Waves"

L8: Define DIN or spell it out.

Introduction:

L6: Processes

Section 2.1:

L23: run

L25: In what sense it is "consistent with the observational data"?

L27: Is the boundary condition daily means? Monthly? What is done with the tides?

L30: Mention that the surface fluxes are computed using the bulk formulae for the marine boundary layer and provide a reference.

Section 2.2

Formula (2) is confusing. While the model is in sigma levels the vertical integration is in depth (I hope so!) where the "dz" is the corresponding sigma layer thickness. The formula also uses summation indexes (x1:xn an y1:yn) as limits for integrals, which is very confusing. One option could be just to do a single integral over area elements dA. Also this expression is equated with the same abbreviation used for expression (1). Maybe use an overbar??

L19: How the unknown groundwater sources were "inversely estimated"? How many? Where? What are their fluxes?

Section 3.1 is good enough but more work is needed in section 3.2 (regional validation) as suggested above.

P7,L3: "Below 55 m the modeled and …"

P7,L4: "since there is no data assimilation". This sentence doesn't makes much sense since you are not comparing contemporaneous values.

P7,L4: "and ARE in"

P7,L5: "To the best of our knowledge this is the first modeling study focusing on the nitrate budget of the YS."

P7,L16: Usually the word "trend" is used for time, so I'll suggest erasing "along time".

P7,L17: How Fig 8e compares to the model equivalent Chl? Does that also show the trend? Given the very large std in the Chl I can't help wonder how big are the error bars for the trend.

P7,L21: "associated with the seasonal cycle AND INTERANNUAL variability".

The trend is presumably related to the interannual part. It could be good to compare that with the physics of the model such as the western boundary current strength, across-shelf pressure gradient, etc.

P7,L31: "inner shelf"

P8,L2-4: Is this the situation all year round? Is it well mixed all year?

P8,L11: Please indicate all the geographical locations mentioned in the text in Fig 1a (Campeche Basin, YS/Campeche Bank, Campeche shelf, etc.).

P9,L6-7: Please rephrase.

P9,L16 onwards. Maybe include a coherence plot of crosh-shelf velocity vs SLA and wind stress?

P10,L1: EASTERN!

P10,L7: "Another". Revise the whole sentence.

P10,L30: "due to"

P10,L31:shows or showed?

P11,L1: erase "produce bottom"

P11,L3:"A similar mechanism has been found for the southern …(Shaeffer et al., 2014)"

P11,L13: revise the value of \rho_0

P11,L14: "We found that" … The formulation used for the bottom Ekman transport implies westward flow for a northward flowing western boundary current, so was that really a finding?

P11,L17: How Fig 14c shows that "the Ekman transport is responsible for 65% of the TN that is entering the shelf."?

P11,L24: Rephrase. Maybe "The high-resolution modeling work of Jouanno et al. (2018) suggest that the bathymetric notch could be responsible for as a much as 50% of the upwelling."?

P11,L28: change "but is not the unique" by "but is not the only process at work."

P12,L17: This is not shown at all. How do we know it is the case?

All figures and figure captions need to be carefully revised.

Figure 8: Is it possible to combine (a)+(b) and (c)+(d)? And maybe include the model equivalent of (e)?

Figure 10: Plot the sections using actual depths instead of sigma levels.

Figure 11: Is this signal visible in the aviso analysis? The signal is strong enough but the temporal variability might be "too fast" for the altimeter constellation to catch it.

---

## Author Comment (AC1) · 29 Jun 2019

**Response to reviewer 1 of "Budget of the total nitrogen in the Yucatan Shelf: driving mechanisms through a physical-biogeochemical coupled model"**

S. Estrada-Allis on behalf of all co-authors.

June 2019

Dear Reviewer, thank you so much for your time reading this paper and providing your suggestions to improving it. Following are responses (in blue) to your detailed comments.

**1   General Comments**

In agreement with the reviewer, we have extended the Yucatan shelf (YS) validation by comparing the transport and current velocity with observations found in the literature (Sheinbaum et al., 2002; Athié et al., 2015; Sheinbaum et al., 2016). The model transport, velocity and its standard deviation are in a good agreement with the observed mean values reported in the literature. A qualitatively comparison is made in terms of salinity with the observations reported in Enriquez et al. (2013). We have added a point-by-point comparison of the salinity from the GOMEX IV cruise and a comparison of the mean salinity and chlorophyll vertical profiles observed for different years in the YS with successful results. In order to prove that the model reproduce the seasonality of the upwelling, the bottom shelf temperature is also shown. Moreover, the model sea level anomaly over the YS is compared with altimetry data from satellite and the model current velocity with the OSCAR current product. We hope that all those new comparisons and model evaluation cal help to convince the readers that the background model physics and dynamics on the shelf is reliable.

Regarding the bias of the model, we have added an extra subsection named 'Impact of model uncertainties'. In general terms, the model tends to overestimate the NO3 and temperature, mainly at the surface. The phytoplankton growth is a function of the temperature through the maximun growth rate (Eppley, 1972). An overestimation of the temperature will produce an increase in the PON concentrations. A consequence of this is that DIN and PON concentrations would be higher than the reality leading to an increase of total nitrogen (TN) budget over shelf. However, the bias is reduced with depth, i.e., the model

uncertainties are important at surface having a little impact over the whole water column.

With respect to the budget units. The main budget for the YS was expressed in mmol N/s in our Figure 9. In the literature, only few papers are focus on the budget quantification enclosing the shelves of the Gulf of Mexico. A recent example is giving in Zhang et al. (2019), in their figure 14 and 16 they shown the sources and sinks of TN for part of the Gulf of Mexico in Gmol N $yr^{-1}$. Xue et al. (2013), their figure 13, denotes the TN budget for the shelves of the Gulf of Mexico (except for the YS) in mol N. Fennel et al. (2006), their Figure 8, show the sources and sinks of N in mmol N $yr^{-1}$, similar as in our Figure 9. None of these papers express the budget in units of area $m^2$. Moreover, notice that in our Table 1, the budget is expressed in molN $yr^{-1}$, such as in Table 1 of Fennel et al. (2006) and Table 2 of Xue et al. (2013), among others studies in the GoM (Fennel, 2010). These are examples of direct comparisons for the same area and with the same physical-biogeochemical coupled model, but different set-up.

**2 Specifics Comments**

1. Abstract, L11: Is there a reason for choosing the 250 m isobath as the shelf boundary?
The reason for choosing the 250 m is explained in P4 L16-20. 200-250 m is the mean depth for shelf break of the Yucatan peninsula (Ruiz-Castillo et al., 2016), please see also our Figure 1. We also pointed out that the YS can be separated into two compartments depending on its dynamics, showed by the mean kinetic energy (see Figure 1b and c).

2. The first paragraph of the Introduction needs to be rewritten. References should be updated with recent relevant work on shelf carbon and nitrogen budgets. The last sentence in this paragraph incorrectly lists acidification and eutrophication as socio-economic activities. These processes may result from socio-economic activities but are not activities in themselves. Likewise, the processes listed as part of the climate system are not ones that would immediately come to mind. Please rewrite.
In agreement with the reviewer comment the Introduction will be rewritten and the most recent work about the nitrogen budget will be also added in the new manuscript.

Page 2,
3. L3: Probably should cite Walsh et al. 1989 here
The cited has been placed in P2 L3.
4. L12-13: It would be good to provide more detail about these controversies to help the reader understand the motivation for this study.
P2 L12-13 will be rewritten in order to provide more detail about the controversies regarding its dynamics which is related with the seasonality and generation

of the upwelling of Cape Catoche, the notch area, influence of the Yucatan Current and the effect of the coastal trapped waves.

5. L26: Show the Yucatan Current in Figure 1.

A better description of the Yucatan Current is included in the new manuscript. A yellow line has been added in Figure 1a to show a zonal transect of the Yucatan Current.

Page 3,

6. L23: Change "was ran" to "was run"

Thanks, this has been corrected in the new manuscript.

Page 4,

7. L12: Suggest deleting the first sentence and starting the paragraph with something like "The XIXIMI cruises provided profiles of nutrients and ...

Thanks, this has been corrected in the new manuscript.

8. L22: Does the SDet equate to dissolved organic nitrogen (DON) in the model? In the real world, the components of total nitrogen are DIN, DON, and PON (or PN since there's some adsorbed inorganic nitrogen on particles). Dissolved organic nitrogen is often equal to or greater than dissolved inorganic nitrogen in the coastal ocean and in coastal rivers. If the SDet does not equate to DON, then your TN definition is incorrect. If SDet does equate to DON, then the assumption of setting PON in rivers equal to 0.1 mmol N m-3 (see comment 11 below) is incorrect.

Here we want to clarify two important things, the first is that the YS does not have rivers. The freshwater sources come from a complex cave system called "cenotes". The second is that the Fennel model does not have DON as a state variable, hence cannot be equated with SDetN. The model variables are described in the manuscript in P4 subsesction 2.2. For a more detailed description of the model you must refer to Fasham et al. (1990); Fennel et al. (2006, 2011). For freshwater sources, the particulated Nitrogen fluxes are assumed to enter as the pool of SDet (Fennel et al., 2011) and, together with the freshwater DON, is set with a small and constant value of 0.1 N m$^{-3}$, see answer to comment 11 below for more details. The definition of TN is the combination of DIN and PON, with DIN the sum of NO3 and NH4, and PON the sum of Phy, Zoo and the two detritus pools. This definition is the one used in Xue et al. (2013), which until now, is the only study that quantify a long-term budget of TN for the whole GoM.

9. L26 and equation (2): This equation is only for the water-column. The total nitrogen budget also includes the loss to denitrification and to burial in the sediments. Please clarify.

In agreement with your comment this lines has been clarified in the new manuscript.

Page 5,

10. L16-20: More details are required about how the freshwater inputs were calculated. Since the freshwater inputs are unknown, it would be justified to include a time-series figure of these inputs, perhaps in the appendix.

In agreement with your comment, a time-series for the most important river systems and freshwater sources (Mississippi/Atchafalaya, Usumacinta-Grijalva, and two representative sources over the YS) will be included in a new Appendix B.

11. L23: Setting the PON to this small value is not justified. I suspect that PON must include DON, else the definition of TN used in this study is incorrect. DON concentrations are generally >> 0.1 mmol N m-3.

This question is related with previous comment 8. As we explained, DON is not included in the Fennel model as state variable. PON is taken as the sum of Phy, Zoo and the detritus pools, which together to DIN (NO3 and NH4), are the definition of TN as in Xue et al. (2013). PON were initialized with a small constant value of 0.1 mmol $N^{-3}$. This approach works well due that the physical-biogeochemical coupled model will evolve with time until reach a distribution representative of the model dynamics. In order to get this adjustment simulations began after a 30 year model spin-up. In fact, Fennel et al. (2006) argue that the adjustment timescales for biogeochemical variables are on the order of days to weeks. This of course does not guaranties that the biological model is validated with respect to a real ocean, but we can ensure that the initial conditions are far from the outputs values used in this study.

12. L26: Provide dates for the November cruise.

The dates for November cruise are now included in the new manuscript.

13. Section 3.1 seems like it should be in the appendix with the other basin wide modeling results. These results aren't really germane to the analysis except as boundary conditions to the shelf.

In agreement with your comment, section 3.1. is now included in the appendix A. Figures are now renamed as Figure A4, A5 and A6.

Page 6,

14. L20: Why is there no model comparison with salinity data? This should be included to provide confidence the model is accurately representing physical transports.

We have added a Figure and description of root mean square, mean and standard deviation in the appendix for salinity comparison with GOMEX IV observations. Moreover, zonal sections of salinity are qualitatively compared with observations shown in Enriquez et al. (2013). A mean vertical profile for salinity observed over the shelf during different years is also compared with the model salinity.

Page 7,

15. L3-9: Poorly worded paragraph. The explanation of why the model results cannot be compared with other results is incorrect. The results from this study should be compared to other studies to put the overall budget for the Yucatan shelf into some context in comparison to other more well-studied shelves in the Gulf such as the West Florida and Louisiana shelves. I recommend normalizing your budget fluxes to area so that they are comparable to other flux estimates.

Comparison with other shelves, even with those from GoM, must be take with care. For instance, the Yucatan shelf is influenced for a complex cave system,

the upwelling at Cape Catoche, the variability of the intense Yucatan Current and the rich mesoscale activity around the area, which is not present in the Florida shelf for example. Sources and sinks of nitrogen are not necessarily the same for the different shelf of the GoM. Please, refer to previous responses regarding the units and normalization of the budget.

16. L15: The trend is mentioned here but there's no explanation. Is it real? What is driving the trend? What source/sink terms have changed? The model is deterministic so there's no reason not to get to the bottom of this, especially since the trend suggests that the N budget is not at steady state.

The explanation of the trend is in P7 L 15-22. It is not an artifact of the model since chlorophyll from satellite products exhibit the same positive trend over nine years. We will further investigate the possible reason of this trend in the new manuscript.

17. L19: I'm not sure what you mean by "a very efficient biological cycle". Please be more specific.

This is related with the efficiency of the inner shelf in that sources and sinks of DIN are in balance with PON, i.e., almost all the NO3 is consumed by phytoplankton or remineralized being in balance with the particulate organic nitrogen.

18. L16-17: This logic doesn't make sense to me. Earlier in the ms it was stated that the chlorophyll time series were used in an inverse analysis to prescribe freshwater and N inputs (also see comment 10). Thus, the TN trend and the chlorophyll trend may not really be independent. Please address whether these are completely independent variables.

Please, see response to previous comment 17.

Page 8,

19. L22: Please report the rates of denitrification (mmol N m-2 d-1 or something similar) obtained from the model.

Thank you, the rates of denitrification are reported in the new manuscript.

20. L24: Fennel et al. (2006) was a study of the Mid-Atlantic and did not address GoM shelves.

This ha been corrected in the new manuscript.

Page 9,

21. L29-: This paragraph should be deleted. The last sentence makes it clear that the present analysis cannot address these phenomena. This has been corrected in the new manuscript.

Page 10,

22. L30: Insert "to" after "due" Thank you, this has been corrected in the new manuscript.

23. L31: Change "show" to "shows" Thank you, this has been corrected in the new manuscript.

Page 11,

24. L13: Is "2015" a typo? Thank you, yes, is a typo and has been corrected in the new manuscript.

25. L28: Delete "the" before "unique" Thank you, this has been corrected in the new manuscript.

26. Prior to Concluding Remarks there needs to be a discussion of the uncertainties in your budget analysis. How does model bias for N concentrations affect your budget? What is the error (standard deviation of the mean) of each term in the mean budget? Without including this, there is no way to make meaningful judgements about the magnitude of the budget terms.

Thank you, this has been rewritten and a new subsection about the impact of the model uncertainties has been added in the new manuscript.

Page 12,

27. L1-2: Figure 15 shows the physical system but not the biogeochemical system 28. Table 1: Normalizing the fluxes to a unit area would be more meaningful since the flux estimates presented are driven by the length of the boundaries and the area of the inner and outer shelf.

Figure 15 shows the physical processes that affect/modulate the biogeochemical system in the YS. Figure 9 shows the biogeochemical system. The units in Table 1 are according to other similar references, for example Fennel et al. (2006) or Xue et al. (2013), among others ...

29. Figure 1: These maps use degrees-minutes whereas other maps use decimal degrees. Be consistent. On Figure 1, the grey contours are difficult to see in panel (a). In panel (b), the vectors are too small to be seen in my copy.

According to the reviewer, the maps with decimal degrees have been modified to degrees-minutes. The grey contours of Figure 1 are now in black and we have increased the line width to improve the visualization. The vectors of panel b of Figure 1 has been also modified to be in black. It is not recommendable to use a higher vector size since the Yucatan Current is characterized to be more intense than the surrounding currents of the shelf. Higher vectors in panel b will distort the figure making it impossible to visualize.

30. Figure 2: It is hard to see the dashed boxes in my copy. Note the isobaths again in this figure caption so the reader knows what these lines are.

According to the reviewer we have increased the width of the dashed boxes and we note the three isobaths in Figure 2.

31. Figure 3: Should be in appendix with basin-wide results.

Figure 3 is now Figure A4 in the new manuscript.

32. Figure 4: Should be in appendix. In panel (a), the shadow and dashed line are difficult to differentiate. In panel (b), report the slope of the linear fit.

Figure 4, both panel (a) and (b), has been modified according your suggestion and it is now Figure A5 in the new manuscript.

33. Figures 5, 6, 7: Report model evaluation statistics such as bias and RMSE. The bias in temperature, NO3, and chlorophyll is generally positive with model results being greater than observations. How does this affect the TN budget calculated with the model?

In agreement with this comment, additional statistical metrics (bias, RMSE)

are now included in the new manuscript. We consider that due the focus of this study, a critical area that needs to be evaluated is the upwelling region. More attention is paid to its validation with observations in the new manuscript as we explained before in the general comments and comment 14.

34. Figure 8: For panels a, c, and e report the p-values for the trend lines. The legends are confusing. Perhaps rename them to Inner Shelf TN, Inner Shelf DIN, Inner Shelf PON, etc.

The p-values are now added and the legends have been changed accordingly.

35. Figure 9: There are differences in values and significant digits presented here and Table 1. Double check these values and make corrections. Also some numerical values for fluxes are difficult to read. A simple 2-D map may make a better figure. Plus, the upside down (S-N) orientation is odd for the 3-D figure.

Please, notice that the differences in Figure 9 and Table 1 are due the units. Table 1 is in mmolN yr$^{-1}$ in order to be comparable with the budgets for the GoM shown in Fennel et al. (2006) and Xue et al. (2013). Numerical values of Table 1 are now bigger. Note that the orientation of the rotated 3-D diagram is marked by the cardinal rose at the left up corner. Now, Figure 9 has been rotated to a N-S diagram. Numbers are higher and isobaths are highlighted.

36. Figure 10: Font is too small for gray depths. Latitude is shown in decimal degrees here. "Isobtahs" is misspelled in the caption.

Thank you, Figure 10 has been modified accordingly in the new manuscript.

37. Figure 11: I can't see the red dot for the station at Lat = 18.3 and Long = -88.1.

The dots are now bigger in panel a, and the panel b is in degree instead of decimal latitude and longitude map.

38. Figure 12: Is this figure necessary? It seems to just show a correlation between currents and SLA that could likely be seen with a simple correlation analysis. What is the unit cpd-1 in the yaxis labels?

Figure 12 is a relevant Figure since shows the correlation between SLA and cross-shelf transport modulation. This can be only achieved by a more specific analysis rather than a simple Fourier spectral analysis. In fact, the analysis of wavelets were also useful to show correlations in the Cape Catoche and Notch area in the study of Jouanno et al. (2018).

39. Figure 13: Difficult to see isobaths in panel (a). In panels b, c, and d, change the blue lines to black to match the y-axis label or change the y-axis label to blue.

Isobaths in panel (a) are now thicker and in black. In panels b, c and d, each color lines match with the y-axis label.

**References**

Athié, G., J. Sheinbaum, R. Leben, J. Ochoa, M. R. Shannon, and J. Candela, 2015: Interannual variability in the yucatan channel flow. *Geophys. Res. Lett.*, **42**, 1496–1503, doi:10.1002/2014GL062674.

Enriquez, C., I. Mariño Tapia, G. Jeronimo, and L. Capurro-Filograsso, 2013: Thermohaline processes in a tropical coastal zone. *Continental Shelf Research*, **69**, 101–109, doi:10.1016/j.csr.2013.08.018.

Eppley, R. W., 1972: Temperature and phytoplankton growth in the sea. *Fish. Bull.*, **70**, 1063–1085.

Fasham, M. J. R., H. W. Ducklow, and S. M. McKelvie, 1990: A nitrogen based model of plankton dynamics in the oceanic mixed layer. *Jorunal of Marine Resesearch*, **48**, 591 – 639.

Fennel, K., 2010: The role of continental shelves in nitrogen and carbon cycling: Northwestern north atlantic case study. *Ocean Sci.*, **6**, 539–548, doi:10.5194/os-6-539-2010,2010.

Fennel, K., R. Hetland, Y. Feng, and S. DiMarco, 2011: A coupled physical-biological model of the northern gulf of mexico shelf: Model description, validation and analysis of phytoplankton variability. *Biogeosciences*, **8**, 1881–1899.

Fennel, K., J. Wilkin, J. Levin, J. Moisan, J. O'Reilly, and D. Haidvogel, 2006: Nitrogen cycling in the middle atlantic bight: results from a three-dimensional model and implications for the north atlantic nitrogen budget. *Global Biogeochemical Cycles*, **20**, doi:10.1029/2005GB002456.

Jouanno, J., E. Pallàs-Sanz, and J. Sheinbaum, 2018: Variability and dynamics of the yucatan upwelling: High-resolution simulations. *Journal of Geophysical Research: Oceans*, **123**, doi:10.1002/2017JC013535.

Ruiz-Castillo, E., J. Gomez-Valdes, J. Sheinbaum, and R. Rioja-Nieto, 2016: Wind-driven coastal upwelling and westward circulation in the yucatan shelf. *Continental Shelf Research*, **118**, 63–76, doi:10.1016/j.csr.2016.02.010.

Sheinbaum, J., G. Athié, J. Candela, J. Ochoa, and R.-A. A., 2016: Structure and variability of the yucatan and loop currentsalong the slope and shelf break of the yucatan channel andcampeche bank. *Dynamics of Atmospheres and Oceans*, **76**, 217–239, doi:10.1016/j.dynatmoce.2016.08.001.

Sheinbaum, J., J. Candela, A. Badan, and J. Ochoa, 2002: Flow structure and transport in the yucatan channel. *Geophys. Res. Lett.*, **29 (1–6)**.

Xue, Z., R. He, K. Fennel, W.-J. Cai, S. Lohrenz, and C. Hopkinson, 2013: Modeling ocean circulation and biogeochemical variability in the gulf of mexico. *Biogeosciences*, **10**, 7219–7234, doi:10.5194/bg-10-7219-2013.

Zhang, S., C. A. Stock, E. N. Curchitser, and R. Dussin, 2019: A numerical model analysis of the mean and seasonal nitrogen budget on the northeast u.s. shelf. *Journal of Geophysical Research: Oceans*, **124**, doi:10.1029/2018JC014308, URL https://agupubs.onlinelibrary.wiley.com/doi/abs/10.1029/2018JC014308, https://agupubs.onlinelibrary.wiley.com/doi/pdf/10.1029/2018JC014308.

---

## Author Comment (AC2) · 29 Jun 2019

**Response to reviewer 2, of "Budget of the total nitrogen in the Yucatan Shelf: driving mechanisms through a physical-biogeochemical coupled model"**

S. Estrada-Allis on behalf of all co-authors.

June 2019

Dear Reviewer, thank you so much for your time reading this paper and providing your valuable suggestions to improve it. Following are responses (in blue) to your detailed comments.

**1 General Comments**

In agreement with the reviewer general comment we have extended the model validation, especially for the Yucatan shelf (YS). More specifically, we compare the transport in Sv and current velocity with three sections of moorings located at the Yucatan Channel, and on the northern side of the shelf (Sheinbaum et al., 2002; Athié et al., 2015; Sheinbaum et al., 2016). The model transport, velocity and its standard deviation are in good agreement with the observed mean values reported in the literature. We added a salinity qualitative comparison against the observations reported in Enriquez et al. (2013). We also included a point-by-point comparison with salinity, temperature and chlorophyll profiles from 3 GOMEX cruises covering different periods. These are all the existing *in situ* observations available for the YS to the best of our knowledge. In order to show that the model reproduce the seasonality of the upwelling, the bottom shelf temperature is also shown. Moreover, the model sea level anomaly over the YS is compared with the merged satellite altimetry product from AVISO. We hope that all these new comparisons and model evaluations improve the robustness and reliability on the background model physics and dynamics. Additionally, a new subsection is added in order to acknowledging the impact of the model uncertainties in the quantification of the total nitrogen budget for the YS.

**2 Specific Comments**

**2.1 Model Validation**

While this study focuses on the YS and its vicinity, most of the convincing validation presented is for the whole Gulf of Mexico (GoM). There is a need to validate and characterize the background state in the YS in order to add credibility to the results. While I understand there is a scarcity of *in situ* observations in the YS, some vertical sections of temperature and salinity have been reported previously (e. g., Enriquez et al., 2013). There are also lots of *in situ* observations in the western boundary current (e.g., Sheinbaum et al., 2002, 2016) that could be used to convince the reader that the model physics is reliable. Specially those closer to the 250 m isobath if available. There is no need to do an exhaustive analysis of such observational datasets, but to show a congruence between model and observed mean background state.

In agreement with the reviewer, there are not many *in situ* observations available for the YS. To expand the model results validation in the YS, a comparison to T-S profiles for August 2016 and July 2018 are included. These are all the existing profiles for the YS to the best of our knowledge. A cross-section of the Yucatan Current is extracted from the model and compared to **?**.

The *in situ* biochemical observations were taken in Nov 2015, while the model solution is available until 2012. Ideally one would have a solution contemporaneous with the observations but if that is not possible then the approach presented is the best next choice: compare the mean and standard deviation for Nov with the observations. But then one wonders is the agreement with observations will hold for other months given that the authors present long-term means for the budgets. The authors seem to suggest that seasonality is weak, but I think that needs to be shown. For instance, does the inner shelf remains well mixed throughout the year as it seems to be the case in Nov?, or does some stratification develops during summer and if so, how that affects the budget?.

As in the comment above for the T-S profiles, we have added a comparison between chlorophyll profiles from different years (August 2016 and July 2018) temporal and spatially averaged for the whole YS. The mean and standard deviation profile of chlorophyll and salinity are presented to show the degree of variability within the shelf. Unfortunately, nutrient and particulate organic nitrogen observations are scarce.

A temporal series of the bottom shelf temperature is also added to the new manuscript to show the model is capable of representing the seasonal variability. More details about the stratification conditions of the shelf are also included. This is a relevant question in the sense that a more stratified system will prevent the uplift of nutrient-rich deep waters that enters into the shelf.

Similarly, more analysis can be done using satellite products to validate the

circulation in the YS. This is specially important to compensate for the scarcity of *in situ* observations. How does the observed Sea Level Anomaly correlates with that of the model in the YS? The authors could for instance compare the annual cycles of the aviso SLA and the model (without Mean Dynamic Topography) to convince the readers that the background model physics on the shelf is reliable. Is the pressure gradient across the shelf break well resolved by the model? A similar comparison could be done with scatterometer winds. The use of OSCAR currents, while very low resolution, might be possible given the wide YS. SST is not a robust validation set since the authors are using bulk fluxes at the surface and the CFSR model (I assume) is forced by satellite SST. Therefore the model is implicitly nudged to observed SST via the 2 m air temperature.

Following the reviewer suggestion, we improved the model validation by comparing Sea Level Anomaly from AVISO and the model. However, one should be careful when using the AVISO product in shallow areas and close to the shore. In addition, we use OSCAR (Ocean Surface Current Analysis Real-time) currents to evaluate the shelf model velocity. Thank you for your comment regarding the SST, this is taken into consideration for the basin-scale model validation.

**2.2 Model Physics**

One the main findings reported in the manuscript is that the input of N in the southeast part of the shelf comes from Ekman transport at the bottom boundary layer. The vertical stretching used was developed to study the surface vorticity balance and while this might be a wise choice for the whole GoM, it is not the best for the YS NT budget given the relevance of the bottom boundary layer. This limitation needs to be further analyzed and acknowledged in the manuscript. The authors could for instance estimate the bottom boundary layer thickness (a common estimate is thickness 0.4*frictionalvelocity/Corilisparamater, where frictional velocity is estimated from the bottom stress). How this thickness compares to the thickness the the first model sigma layer for the 250 m isobath?. That is, how well is the bottom boundary layer resolved?. If it is not very well resolved I still think the bulk characteristics of the budget will hold (i.e., the main sources and sinks and the TN pathways), but this limitation needs to be acknowledged.

We agree with the reviewer in the sense that the chosen vertical stretching (Azevedo Correia de Souza et al., 2015) was developed to provide higher resolution near the surface. In fact, in most areas of the study region important fluxes are concentrated near the surface (see for eg. manuscript figure 10). Therefore, the chosen vertical scheme is justified. That said, we agree with the reviewer that the bottom Ekman layer needs to be well presented for us to be able to draw conclusions on the balance and the physical processes modulating it. To verify the model is able to represent the this layer, its thickness was calculated

[Figure]

Figure 1: Bottom Ekman thickness, temporally averaged for the Yucatan Shelf. In blue is the Ekman thickness computed theoretically as shown in the legend. The friction velocity ($u^*$) is obtained from $\sqrt{\tau_{by}/\rho_o}$, where $\tau_{by}$ is the along-shelf bottom-stress, $\rho_o$ is the reference density of 1025 km m$^{-3}$, and $f$ is the Coriolis parameter. In black, is the bottom-most layer ($dz$) resolved by the model. Shadow areas denote the temporal standard deviation for one simulated year.

and compared to the model vertical grid (see Figure 1 below). It is shown that, despite the stretching used the model can properly reproduce both the surface and bottom Ekman layers. Even in periods when the velocity is very low, the bottom Ekamn layer is still larger than the model last layer. In these particular cases, the Ekman transport will not be important. Following the reviewer suggestion, comments are added to the text to make clear the limitations in the vertical resolution and the possible consequences.

**2.3 Technical Corrections**

Abstract:

L4: I think it should be "Coastal-Trapped Waves" or "Coastally Trapped Waves"

Thank you, this has been corrected as "Coastal-Trapped Waves" in the revised version.

L8: Define DIN or spell it out.

The DIN term is spell it out in the Abstract in the revised version.

Introduction:

L6: Processes

Thank you, this has been corrected in the revised version.

Section 2.1: L23: run

Thank you, this has been corrected in the revised version.

L25: In what sense it is "consistent with the observational data"?

To be time-consistent, i.e., in the sense that the model and observations match in the same time range, as far as possible.

L27: Is the boundary condition daily means? Monthly? What is done with the tides?

The boundary conditions are daily averaged. The tides are hourly and added as a separate spectral forcing at the boundaries. This is clarified in the revised version.

L30: Mention that the surface fluxes are computed using the bulk formulae for the marine boundary layer and provide a reference.

We mentioned this in the revised version, thanks.

Section 2.2 Formula (2) is confusing. While the model is in sigma levels the vertical integration is in depth (I hope so!) where the "dz" is the corresponding sigma layer thickness. The formula also uses summation indexes (x1:xn an y1:yn) as limits for integrals, which is very confusing. One option could be just to do a single integral over area elements dA. Also this expression is equated with the same abbreviation used for expression (1). Maybe use an overbar??

Effectively, the model is in sigma levels and the vertical integration is in the corresponding depth for each sigma level. We agree that formula (2) is confusing. In order to make the text more readable we have avoided this equation, which can be easily explained in the text of the revised manuscript as follows "Accordingly, the total budget is obtained as the integral over the area of the shelf and over the depth of the water column for the inner and the outer shelf".

L19: How the unknown groundwater sources were "inversely estimated"? How many? Where? What are their fluxes?

With "inversely" we mean that the groundwater sources are estimated by fitting the freshwater model sources to the scarce transport, salinity, temperature and nutrient data cited in the literature. We have avoided the term "inversely" to clarify this sentence. Moreover, we will add an extra appendix B to present examples of the monthly climatology of Mexican rivers and freshwater sources.

Section 3.1 is good enough but more work is needed in section 3.2 (regional validation) as suggested above.

Thank you for this suggestion. More validation is presented in the new version of the manuscript.

P7,L3: "Below 55 m the modeled and . . . "

Thank you, this has been corrected in the revised version.

P7,L4: "since there is no data assimilation". This sentence doesn't makes much sense since you are not comparing contemporaneous values.

Thank you, this sentence has been erased in the revised version.

P7,L4: "and ARE in"

Thank you, this has been corrected.

P7,L5: "To the best of our knowledge this is the first modeling study focusing on the nitrate budget of the YS."

Thank you, this has been corrected.

P7,L16: Usually the word "trend" is used for time, so I'll suggest erasing "along time".

Thank you, this has been corrected.

P7,L17: How Fig 8e compares to the model equivalent Chl? Does that also show the trend? Given the very large std in the Chl I can't help wonder how big are the error bars for the trend.

The model Chl compares well with the satellite Chl as shown in Figure 2. The model also exhibit the positive trend. The large std for the Chl is related with the strong variability given by upwelling episodes in the shelf. To better show the Chl trend one option is to avoid the std of the time series and show the values in the text as the temporal mean and its std, i.e., The value for the satellite Chl averaged over the shelf area and over the period of time between 2002-2010 is $0.38 \pm 0.09$ mgChl m$^{-3}$, similarly, the model shows a spatially-temporal mean of $0.36 \pm 0.13$ mgChl m$^{-3}$ for the same area and time period.

P7,L21: "associated with the seasonal cycle AND INTERANNUAL variability". The trend is presumably related to the interannual part. It could be good to compare that with the physics of the model such as the western boundary current strength, across-shelf pressure gradient, etc.

We agree with the reviewer, and further details on the causes of this trend will be investigated in the revised version of the manuscript, thank you.

P7,L31: "inner shelf"

Thank you, this has been corrected.

P8,L2-4: Is this the situation all year round? Is it well mixed all year?

The mixing state of the shelf will be analyzed in more detail in the revised manuscript.

P8,L11: Please indicate all the geographical locations mentioned in the text in Fig 1a (Campeche Basin, YS/Campeche Bank, Campeche shelf, etc.).

The geographical locations will be indicated in Fig. 1a.

P9,L6-7: Please rephrase.

The sentence has been rephrase as: "The CTWs are remotely forced by alongshore winds, propagating from the western GoM (Dubranna et al., 2011;

[Figure]

Figure 2: Temporal series of surface chlorophyll (mgChl m$^{-3}$) given by the satellite (thick gray line) and by the model (thick black line), averaged for the period of years between 2002-2010 and for the YS area. Thick dashed lines are the fitted trends for each temporal series, and thin dashed lines are the corresponding 95% confidence interval. Equations for the linear fits are $Chl_{trend}$ = 0.0010 month + 0.28 for satellite, and $Chl_{trend}$ = 0.0011 month + 0.30 for the model chlorophyll trend.

Jouanno et al., 2016)."
P9,L16 onwards. Maybe include a coherence plot of crosh-shelf velocity vs SLA and wind stress?

We will consider to include a coherence plot between the three variables.
P10,L1: EASTERN!

This has been corrected.
P10,L7: "Another". Revise the whole sentence.

The whole sentence is revised.
P10,L30: "due to"

This has been corrected.
P10,L31:shows or showed?

shows.
P11,L1: erase "produce bottom"

This has been erased, thank you.
P11,L3:"A similar mechanism has been found for the southern ...(Shaeffer et al., 2014)"

This has been modified, thank you.
P11,L13: revise the value of rho 0

This has been revised.
P11,L14: "We found that" ...  The formulation used for the bottom Ekman

transport implies westward flow for a northward flowing western boundary current, so was that really a finding?

This has been revised.

P11,L17: How Fig 14c shows that "the Ekman transport is responsible for 65 % of the TN that is entering the shelf."?

The sentence is rephrase since the Figure 14 c does not show the percentage.

P11,L24: Rephrase. Maybe "The high-resolution modeling work of Jouanno et al. (2018) suggest that the bathymetric notch could be responsible for as a much as 50 % of the upwelling."?

The sentence has been rephrased, thank you.

P11,L28: change "but is not the unique" by "but is not the only process at work."

The sentence has been rephrased, thank you.

P12,L17: This is not shown at all. How do we know it is the case? All figures and figure captions need to be carefully revised. The sentence has been erased. All the figures and captions will be revised in the revised version of the manuscript.

Figure 8: Is it possible to combine (a)+(b) and (c)+(d)? And maybe include the model equivalent of (e)?

We will combine panels (a+b) and (c+d). The model equivalent is now included in the revised version (see previous Figure 2).

Figure 10: Plot the sections using actual depths instead of sigma levels.

Actual depths are shown in each sigma level. We will plot the sections using actual depths however.

Figure 11: Is this signal visible in the aviso analysis? The signal is strong enough but the temporal variability might be "too fast" for the altimeter constellation to catch it.

The signal from AVISO altimetry will be compared.

**References**

Athié, G., J. Sheinbaum, R. Leben, J. Ochoa, M. R. Shannon, and J. Candela, 2015: Interannual variability in the yucatan channel flow. *Geophys. Res. Lett.*, **42**, 1496–1503, doi:10.1002/2014GL062674.

Azevedo Correia de Souza, J., B. Powell, A. C. Castillo-Trujillo, and P. Flament, 2015: The vorticity balance of the ocean surface in hawaii from a regional reanalysis. *Journal of Physical Oceanography*, **45 (2)**, 424–440, doi:10.1175/JPO-D-14-0074.1, URL https://doi.org/10.1175/JPO-D-14-0074.1, https://doi.org/10.1175/JPO-D-14-0074.1.

Dubranna, J., P. Pérez-Brunius, M. López, and J. Candela, 2011: Circulation over the continental shelf of the western and southwestern gulf of mexico. *Journal of Geophysical Research*, **116 (C08009)**, doi:10.1029/2011JC007007.

Enriquez, C., I. Mariño Tapia, G. Jeronimo, and L. Capurro-Filograsso, 2013: Thermohaline processes in a tropical coastal zone. *Continental Shelf Research*, **69**, 101–109, doi:10.1016/j.csr.2013.08.018.

Jouanno, J., K. Ochoa, E. Pallàs-Sanz, J. Sheinbaum, F. Andrade, J. Candela, and J. Molines, 2016: Loop current frontal eddies: Formation along the campeche bank and impact of coastally trapped waves. *Journal of Physical Oceanography*, **46 (11)**, 3339–3363, doi:10.1175/JPO-D-16-0052.1.

Sheinbaum, J., G. Athié, J. Candela, J. Ochoa, and R.-A. A., 2016: Structure and variability of the yucatan and loop currentsalong the slope and shelf break of the yucatan channel andcampeche bank. *Dynamics of Atmospheres and Oceans*, **76**, 217–239, doi:10.1016/j.dynatmoce.2016.08.001.

Sheinbaum, J., J. Candela, A. Badan, and J. Ochoa, 2002: Flow structure and transport in the yucatan channel. *Geophys. Res. Lett.*, **29 (1–6)**.

---

## Author Response (AR1)

**Response to reviewer 1 of "Budget of the total nitrogen in the Yucatan Shelf: driving mechanisms through a physical-biogeochemical coupled model"**

S. Estrada-Allis on behalf of all co-authors.

August 2019

We would like to thank the reviewer for the comments to the paper and the suggestions for improving it. Following are our responses (in blue) to all comments. The modifications to the original manuscript can be found in the attached document named Tracking-changes.pdf, where new text is marked in blue and removed text in red.

**Overview:**

The authors present an analysis of the total nitrogen budget on the Yucatan shelf as influenced by physical transports, mixing, river inputs, and biogeochemical processes. A coupled physical biogeochemical model was used to quantify the processes driving nitrogen source and sink terms. On the eastern boundary of the Yucatan shelf, the Yucatan Current is the dominant flux. Bottom Ekman transport towards the shelf is also important in this area. On the western and northwestern boundary, coastal trapped waves drive exchanges with the open Gulf of Mexico. A westward current on the inner shelf results in N exports at the western boundary of the inner shelf. The results of this work are interesting because the Yucatan shelf has been poorly studied and because the geographic setting provides an interesting interplay of different physical processes that are overlaid on one another.

**1 General Comment**

Unfortunately, I cannot recommend publication of the manuscript at this time for three reasons. First, I found the validation of the physical model insufficient. If there are any measurements of ocean currents for the Yucatan shelf, these should be presented and discussed to evaluate the accuracy of the modeled physical transports. If there are no data for currents, which may be likely especially outside the Yucatan current, the model could be validated by presenting comparisons of modeled versus observed salinity data. Second, I find that the manuscript lacks a discussion of how model bias in physical and biogeochemical state variables may influence results and there is no presentation of uncertainty estimates for the calculated budget source and sink terms, which makes it impossible to compare magnitudes of these terms. Third, the sink and source values were not presented as normalized to a unit area (e.g. m2) and thus the results from this study cannot be compared to results from previous N budget work in the Gulf of Mexico or elsewhere.

The issues raised by the reviewer in this first general comment have been addressed in the following way:

We have extended the Yucatan shelf (YS) model validation by comparing the model current velocity with observations reported in the literature (Sheinbaum et al., 2002; Athié et al., 2015; Sheinbaum et al., 2016). The model velocity structure, mean and standard deviation are in good agreement with these observations (see appendix A, subsection A4 and new Figure A9). A qualitative comparison is carried out for salinity with observations reported in Enriquez et al. (2013), see Figure 1 included here for comparison with Figure 9 in Enriquez et al. (2013).

We have added a point-by-point comparison of the salinity from the GOMEX IV cruise (see P9 L11-15, and new Figure 5) and a comparison of the mean chlorophyll vertical profiles observed for three different years in the YS (see new appendix A, subsection A2 and new Figure A6) with successful results. In order to prove that the model is able to reproduce the seasonality of the upwelling, the bottom shelf temperature is also shown as well as the seasonality of the upwelling marked by isotherm 22.5 °C (see P8 L28-32, new Figure 3) for comparison with Merino (1997). Moreover, the model sea level anomaly over the YS is compared with satellite altimetry data and the model's current velocity with the Global Current product (see new appendix A, subsection A3 and new Figure A7). We hope that all these new comparisons between model and observations are enough to convince the reviewers that the model results on the YS shelf are reliable and capture the main features of the observed variability. Note that the size and number of figures in the new version of this paper has increased considerably. Those for validation are included in appendix A which may be shortened or added instead as supporting information rather than an appendix if necessary.

Regarding the bias of the model, we have added appendix B named "Model statistics". In general, the model tends to overestimate the NO3 and temperature, mainly at the surface. A possible consequence of this is that the total nitrogen (TN) budget over the shelf may be overestimated by the model although differences are less than 2% (see new appendix B). It is important to mention a couple of extra things about this: firstly, the bias is reduced in the upwelling area, which is one of the main objectives of this study and is the region where most of the TN is entering the shelf. Secondly, despite a possible NO3 overestimation, the main nitrogen flux pathways remain unchanged.

Regarding the issue of budget units there are only a few published papers that address quantification of net biogeochemical budgets of the shelves of the Gulf of Mexico. A recent example Zhang et al. (2019) (see their figures 14 and 16) show the sources and sinks of TN for part of the Gulf of Mexico in Gmol N  $yr^{-1}$ . Xue et al. (2013), (their figure 13), report the TN budget for the shelves of the Gulf of Mexico (except for the YS) in mol N. In other regions also, (Fennel et al. (2006), (their Figure 8)), the sources and sinks of N are expressed in mmol N yr $^{-1}$ , similar to our Figure 9. None of these papers express the net budgets in units per unit area  $m^2$ . We understand the the concern of the reviewer regarding this issue, but prefer to express our results in the units used by other relevant works in the region to ease comparison. We certainly agree that in doing so, the results depend on the size of each region or shelf but believe they provide useful information to the overall GoM TN budget. For all these reasons we have kept the net budget units as in the original manuscript. Note that in our Table 1, the budget is expressed in molN  $yr^{-1}$ , same as in Table 2 of Xue et al. (2013) who compute budgets in the other GoM shelves using a similar model but different set-up. We are convinced this allows a better comparison of different model results in the region. We can easily compute budgets per unit area or length (fluxes) but would not be able to compare them with results available in the literature.

**2 Specifics Comments**

1. Abstract, L11: Is there a reason for choosing the 250 m isobath as the shelf boundary?

The reason for choosing the 250 m is explained in P4 L16-20 of the previous manuscript and remains in the new one: 200-250 m is the mean depth of the shelf break of the Yucatan peninsula (e.g., Ruiz-Castillo et al., 2016), please see Figure 1 as well. We also pointed out that the YS can be separated into two compartments based on quite different values of the mean kinetic energy (see Figure 1b and c).

2. The first paragraph of the Introduction needs to be rewritten. References should be updated with recent relevant work on shelf carbon and nitrogen budgets. The last sentence in this paragraph incorrectly lists acidification and eutrophication as socio-economic activities. These processes may result from socio-economic activities but are not activities in themselves. Likewise, the processes listed as part of the climate system are not ones that would immediately come to mind. Please rewrite.

Following the reviewer's comment the first paragraph of the Introduction has been rewritten and the most recent work about nitrogen budget is also referenced. Please, check section 1 of the Introduction in the new version

Page 2,

3. L3: Probably should cite Walsh et al. 1989 here Citation has been placed in P2 L20.

4. L12-13: It would be good to provide more detail about these controversies to help the reader understand the motivation for this study.

The introduction section has been modified to make it clearer and easier to understand our motivations which are now at the end of the section, after a short explanation of the controversies regarding the dynamics that control the YS variability(section 1 of Introduction).

5. L26: Show the Yucatan Current in Figure 1.

A better description of the Yucatan Current is included in the new manuscript (P3 L10-13). A yellow line has been added in Figure 1a to show a zonal transect of the Yucatan Current.

Page 3,

6. L23: Change "was ran" to "was run"

Thanks, this has been corrected in the new manuscript (P5 L2). Page 4,

7. L12: Suggest deleting the first sentence and starting the paragraph with something like "The XIXIMI cruises provided profiles of nutrients and "

Thanks, this has been corrected in the new manuscript (P5 L26-27).

8. L22: Does the SDet equate to dissolved organic nitrogen (DON) in the model? In the real world, the components of total nitrogen are DIN, DON, and PON (or PN since there's some adsorbed inorganic nitrogen on particles). Dissolved organic nitrogen is often equal to or greater than dissolved inorganic nitrogen in the coastal ocean and in coastal rivers. If the SDet does not equate to DON, then your TN definition is incorrect. If SDet does equate to DON, then the assumption of setting PON in rivers equal to 0.1 mmol N m-3 (see comment 11 below) is incorrect.

Here we want to clarify two important things. The first is that the YS does not have rivers. The freshwater sources or more specific the Submarine Groundwater Discharges (SGD) come from a complex cave system called "cenotes", lagoons and other sources. The second is that the Fennel model does not have DON as a state variable, hence cannot be equated with SDetN. The model variables are described in the old manuscript in P4 subsection 2.2. For a more detailed description of the model refer to Fasham et al. (1990); Fennel et al. (2006, 2011). For freshwater sources, the particulated Nitrogen fluxes are assumed to enter as the pool of SDet (Fennel et al., 2011) and, together with the freshwater DON, is set with a small and constant value of 0.1 N m-3, see answer to comment 11 below for more details. The definition of TN is the combination of DIN and PON, with DIN the sum of NO3 and NH4, and PON the sum of Phy, Zoo and the two detritus pools. This definition is the one used in Xue et al. (2013), which until now, is the only study that quantifies a long-term budget of TN for the whole GoM.

Biogeochemical models are approximations to reality and make several assumptions and simplifications depending on their complexity. The above mentioned published papers use empirical values and ad-hoc approximations that have shown merit in reproducing the main features of the biogeochemistry of the GoM. Our model parameters are in tune with those cited papers and are modified only if observations are available or local conditions suggest they should be "tuned" within reasonable values to better reproduce the few available observa-

**tions.**

9. L26 and equation (2): This equation is only for the water-column. The total nitrogen budget also includes the loss to denitrification and to burial in the sediments. Please clarify.

We agree with your comment. Thanks. This has been corrected in the new manuscript (P6 L11-13).

**Page 5,**

10. L16-20: More details are required about how the freshwater inputs were calculated. Since the freshwater inputs are unknown, it would be justified to include a time-series figure of these inputs, perhaps in the appendix.

Thank you for the suggestion. A time-series for the most important river systems and freshwater sources (Mississippi/Atchafalaya, Usumacinta-Grijalva, and freshwater sources of the Yucatan Shelf) are included in a new appendix C.

11. L23: Setting the PON to this small value is not justified. I suspect that PON must include DON, else the definition of TN used in this study is incorrect. DON concentrations are generally >> 0.1 mmol N m-3.

This question is related to previous comment 8. As we explained, DON is not included in the Fennel model as state variable. PON is taken as the sum of Phy, Zoo and the detritus pools, which together to DIN (NO3 and NH4), are the definition of TN as in Xue et al. (2013). PON were initialized with a small constant value of 0.1 mmol  $N^{-3}$ . This a numerical approach that works well due to the fact that the physical-biogeochemical coupled model evolves with time until it reaches a distribution representative of the model dynamics in accordance with observations. As we also mentioned, it is used in all the cited literature. In order to get this adjustment to reasonable values the analysis and budgets presented in the paper are carried out after a 30 year model spin-up. In fact, Fennel et al. (2006) argues that the adjustment timescales for near surface biogeochemical variables are on the order of days to weeks. Therefore the model "forgets" these initial values relatively quickly and produces realistic values based on its internal dynamics. Again, it is just a numerical technique used to produce more realistic results after model spin-up.

12. L26: Provide dates for the November cruise.

The dates for November cruise are now included in the new manuscript (P7 L17).

13. Section 3.1 seems like it should be in the appendix with the other basin wide modeling results. These results aren't really germane to the analysis except as boundary conditions to the shelf.

Following your comment, section 3.1. is now included in appendix A which deals with model validation. Figures are now renamed as Figure A4, A5 and A6.

**Page 6,**

14. L20: Why is there no model comparison with salinity data? This should be included to provide confidence the model is accurately representing physical transports.

We have added Figure 5 and description of the salinity comparison with GOMEX IV observations in P9 L11-15. Moreover, a T-S diagram is qualitatively compared (here, Figure 1) with observations shown in Enriquez et al. (2013) (see the first General comments section)

**Page 7,**

15. L3-9: Poorly worded paragraph. The explanation of why the model results cannot be compared with other results is incorrect. The results from this study should be compared to other studies to put the overall budget for the Yucatan shelf into some context in comparison to other more well-studied shelves in the Gulf such as the West Florida and Louisiana shelves. I recommend normalizing your budget fluxes to area so that they are comparable to other flux estimates. We addressed this issue in our reply to the first general comment. Text has been changed accordingly. We do believe that even if integrated budgets were given normalized by area, one should be aware that dynamics are very different on each shelf so having similar or different values per unit area may not help interpretation of what causes those similarities or differences, which was the intention of the last phrase referred to by the reviewer in the comment above. Again, Zhang et al. (2019) and Xue et al. (2013) do not report net budgets per unit area and therefore our Table 1 can be directly compared to their results. We stress again that we could easily compute budgets by unit area but that would not help comparison with results from other shelves available in the literature

16. L15: The trend is mentioned here but there's no explanation. Is it real? What is driving the trend? What source/sink terms have changed? The model is deterministic so there's no reason not to get to the bottom of this, especially since the trend suggests that the N budget is not at steady state.

The explanation of the trend is in P7 L15-22 of the old manuscript. It is not an artifact of the model since chlorophyll from satellite products exhibits the same positive trend over nine years. Having said that, finding the cause or causes of this trend is not trivial. Neither the biogeochemical input data nor the freshwater inputs have this trend. The trend may be related to physical processes, for example the variability/strength of the Yucatan Current, as suggested by the study of ?. However, many observational uncertainties remain. The main goal of this study is the description of the general budget of the Yucatan Shelf. The observations have such a trend and the model is able to reproduce it, so we are confident of our results. Finding out what is or are the mechanisms behind such trend is certainly a very interesting problem, but is out of the scope of the present study and further research is needed in order to get to the bottom of this. This is now better explained in P10 L26-31 of the revised manuscript.

17. L19: I'm not sure what you mean by "a very efficient biological cycle". Please be more specific.

This is related to the efficiency of the inner shelf in that sources and sinks of DIN are in balance with PON, i.e., almost all the NO3 is consumed by phytoplankton or remineralized and in balance with the particulate organic nitrogen. This sentence has been clarified in P10 L31-32.

18. L16-17: This logic doesn't make sense to me. Earlier in the ms it was

stated that the chlorophyll time series were used in an inverse analysis to prescribe freshwater and N inputs (also see comment 10). Thus, the TN trend and the chlorophyll trend may not really be independent. Please address whether these are completely independent variables.

We sincerely regret the use of the wording "inverse method" in the original manuscript. It has unfortunately caused a lot of unnecessary confusion. Chlorophyll was not used to determine fresh water or nutrient fluxes. All we did was to compile as much data as possible including literature references regarding fresh water and nutrient data to build a climatology of fresh water and nutrient fluxes to force the model at the southern GoM boundary. Again, no chlorophyll data were used for this. Whilst in the northern GoM there are long time series of nutrient, salinity, temperature and volume transports for most of the rivers, at the southern GoM boundary data are very sparse and scarce. We used some data from near coast stations to infer nutrient fluxes (e.g. near coast stations of the GOMEX IV cruise). We have added the appendix C to explain this in more detail including figures and have changed the wording to avoid confusion. Therefore the trend in model TN is not caused by the trend in observed chlorophyll data.

**Page 8,**

19. L22: Please report the rates of denitrification (mmol N m-2 d-1 or something similar) obtained from the model.

The rates of denitrification obtained from the model are given in mmolN m-2 d-1 and included in Table 1 in the revised manuscript. The rates of denitrification are averaged for the nine simulated years and reported in P11 L14-15.

20. L24: Fennel et al. (2006) was a study of the Mid-Atlantic and did not address GoM shelves.

This has been corrected in the new manuscript, reference now is to (Xue et al., 2013) instead. Thank you (see P12 L14).

Page 9,

21. L29: This paragraph should be deleted. The last sentence makes it clear that the present analysis cannot address these phenomena. According to the reviewer suggestion (see also response to comment regarding the coherence plot of Reviewer 2), we have modified the analysis of the CTWs of sesction 4.1. Now, the wavelet power spectrum analysis is performed to show the climatology (daily averages from the 9 year results) of SLA, TN fluxes and along-shelf wind-stress (new Fig. 12) and concentrate on seasonal variations or higher frequency. Although we do not show now the year to year variability in the wavelet spectrum, we could not avoid mentioning the coincidence between the years of highly energetic events and the occurrence of relevant climatic signals (El Niño). The paragraph was modified mentioning this coincidence but recognizing that longer time-series are needed to investigate such variability.

**Page 10,**

22. L30: Insert "to" after "due" Thank you, this has been corrected in the new

**manuscript.**

23. L31: Change "show" to "shows" Thank you, this has been corrected in the new version of the manuscript.

**Page 11,**

24. L13: Is "2015" a typo? Yes, thank you, is a typo and has been corrected in the new version P16 L28.

25. L28: Delete "the" before "unique" Thank you, this has been corrected in the new manuscript P17 L19.

26. Prior to Concluding Remarks there needs to be a discussion of the uncertainties in your budget analysis. How does model bias for N concentrations affect your budget? What is the error (standard deviation of the mean) of each term in the mean budget? Without including this, there is no way to make meaningful judgements about the magnitude of the budget terms.

Thank you, the impact of the model uncertainties has been added to the revised version of the manuscript in appendix B.

**Page 12,**

27. L1-2: Figure 15 shows the physical system but not the biogeochemical system. Table 1: Normalizing the fluxes to a unit area would be more meaningful since the flux estimates presented are driven by the length of the boundaries and the area of the inner and outer shelf.

Figure 15 shows the physical processes that affect/modulate the biogeochemical system in the YS. Figure 9 shows the biogeochemical system. As we explained before, units in Table 1 are according to other similar studies (Xue et al. (2013) or Zhang et al. (2019) to ease comparison. As in the references, numbers represent net (integrated) values/contributions for the YS. Perhaps we should add here that the goal of the paper is to understand the main processes controlling the YS TN budget. The units used allow comparison with published results but detailed comparison of our results with other GoM shelves is a subject to be addressed in the future

29. Figure 1: These maps use degrees-minutes whereas other maps use decimal degrees. Be consistent. On Figure 1, the grey contours are difficult to see in panel (a). In panel (b), the vectors are too small to be seen in my copy.

Following the reviewer suggestion, the maps with decimal degrees have been modified to degrees-minutes. The grey contours of Figure 1 are now in black and we have increased the line width to improve the visualization. The vectors of panel b of Figure 1 have been changed to black. It is not straightforward to use larger vector sizes to improve the figure since the Yucatan Current is much more intense than the surrounding currents over the shelf larger vectors in panel b will distort the figure making it impossible to visualize. We have done our best to improve the figure hoping it is easier to see

30. Figure 2: It is hard to see the dashed boxes in my copy. Note the isobaths again in this figure caption so the reader knows what these lines are.

We have increased the width of the dashed boxes and we note the three isobaths in Figure 2.

31. Figure 3: Should be in appendix with basin-wide results.

Figure 3 is now Figure A4 of appendix A in the revised version.

32. Figure 4: Should be in appendix. In panel (a), the shadow and dashed line are difficult to differentiate. In panel (b), report the slope of the linear fit.

Figure 4, both panel (a) and (b), has been modified according to your suggestion and it is now Figure A5 of appendix A in the revised version.

33. Figures 5, 6, 7: Report model evaluation statistics such as bias and RMSE. The bias in temperature, NO3, and chlorophyll is generally positive with model results being greater than observations. How does this affect the TN budget calculated with the model?

additional statistical metrics (bias, RMSE) are now included in the revised version (section 3.1, P8 and 9). We consider that given the focus of this study, a critical area that needs to be evaluated is the upwelling region. More attention is paid to its validation with observations in the new manuscript as we explained before in answering your general comments and comment 14. The model may overestimate or underestimate some values but we believe it captures the basic mechanisms determining the balances

34. Figure 8: For panels a, c, and e report the p-values for the trend lines. The legends are confusing. Perhaps rename them to Inner Shelf TN, Inner Shelf DIN, Inner Shelf PON, etc.

The p-values are now added and the legends have been changed accordingly, thank you (Figure 8).

35. Figure 9: There are differences in values and significant digits presented here and Table 1. Double check these values and make corrections. Also some numerical values for fluxes are difficult to read. A simple 2-D map may make a better figure. Plus, the upside down (S-N) orientation is odd for the 3-D figure. Thank you for the suggestion, the values of Table 1 and Figure 9 are now expressed in mmolN  $yr^{-1}$  in order to be comparable with the budgets for the GoM shown in Fennel et al. (2006) and Xue et al. (2013). Numerical values of Table 1 are now bigger. Figure 9 has been rotated to a N-S diagram. Numbers are higher and isobaths are highlighted.

36. Figure 10: Font is too small for gray depths. Latitude is shown in decimal degrees here. "Isobtahs" is misspelled in the caption.

Thank you, Figure 10 has been modified accordingly in the new manuscript.

37. Figure 11: I can't see the red dot for the station at Lat = 18.3 and Long = -88.1.

The dots are now bigger in panel a, and the panel b is in degree instead of decimal latitude and longitude map.

38. Figure 12: Is this figure necessary? It seems to just show a correlation between currents and SLA that could likely be seen with a simple correlation analysis. What is the unit cpd-1 in the y-axis labels?

Thank you for the suggestion. As explained in comment 21, we have modified section 4.1. Now Fig. 12, along with new Fig. 13, show that the high-frequency variability of the TN fluxes at the western YS border are modulated by surface Ekman transport and propagation of CTWs. We wanted to capture these variations in time but following your comments we concentrate on seasonal and

higher frequency variations. That is the reason why we use the "climatology" of the wavelet spectrum and coherence analysis (as suggested also by reviewer 2). Capturing this time changes can not be obtained from a simple correlation analysis. Our analysis allows us to capture the increased variability during winter at the western YS border.

39. Figure 13: Difficult to see isobaths in panel (a). In panels b, c, and d, change the blue lines to black to match the y-axis label or change the y-axis label to blue.

Isobaths in panel (a) are now thicker and in black. In panels b, c and d, each color lines match with the y-axis label.

Figure 1: Spatially and temporally averaged over nine simulated years T-S diagram for the whole YS. Color boxes denote the three main water masses found on the shelf: the Caribbean Subtropical Underwater (CSUW), the high salinity, warm and surface Yucatan Sea Water (YSW) and Gulf Common Water (GCS). For more information refer to Enriquez et al. (2013).

**References**

- Athié, G., J. Sheinbaum, R. Leben, J. Ochoa, M. R. Shannon, and J. Candela, 2015: Interannual variability in the yucatan channel flow. *Geophys. Res. Lett.*, 42, 1496–1503, doi:10.1002/2014GL062674.
- Enriquez, C., I. Mariño Tapia, G. Jeronimo, and L. Capurro-Filograsso, 2013: Thermohaline processes in a tropical coastal zone. *Continental Shelf Research*, 69, 101–109, doi:10.1016/j.csr.2013.08.018.
- Fasham, M. J. R., H. W. Ducklow, and S. M. McKelvie, 1990: A nitrogen based model of plankton dynamics in the oceanic mixed layer. *Jorunal of Marine Research*, 48, 591 – 639.
- Fennel, K., R. Hetland, Y. Feng, and S. DiMarco, 2011: A coupled physicalbiological model of the northern gulf of mexico shelf: Model description, validation and analysis of phytoplankton variability. *Biogeosciences*, 8, 1881–1899.
- Fennel, K., J. Wilkin, J. Levin, J. Moisan, J. O'Reilly, and D. Haidvogel, 2006: Nitrogen cycling in the middle atlantic bight: results from a three-dimensional model and implications for the north atlantic nitrogen budget. *Global Biogeochemical Cycles*, 20, doi:10.1029/2005GB002456.
- Merino, M., 1997: Upwelling on the yucatan shelf: hydrographic evidence. Journal of marine systems, 13, 101–121.
- Ruiz-Castillo, E., J. Gomez-Valdes, J. Sheinbaum, and R. Rioja-Nieto, 2016: Wind-driven coastal upwelling and westward circulation in the yucatan shelf. *Continental Shelf Research*, **118**, 63–76, doi:10.1016/j.csr.2016.02.010.
- Sheinbaum, J., G. Athié, J. Candela, J. Ochoa, and R.-A. A., 2016: Structure and variability of the yucatan and loop currentsalong the slope and shelf break of the yucatan channel and campeche bank. *Dynamics of Atmospheres* and Oceans, **76**, 217–239, doi:10.1016/j.dynatmoce.2016.08.001.
- Sheinbaum, J., J. Candela, A. Badan, and J. Ochoa, 2002: Flow structure and transport in the yucatan channel. *Geophys. Res. Lett.*, **29** (1–6).
- Xue, Z., R. He, K. Fennel, W.-J. Cai, S. Lohrenz, and C. Hopkinson, 2013: Modeling ocean circulation and biogeochemical variability in the gulf of mexico. *Biogeosciences*, **10**, 7219–7234, doi:10.5194/bg-10-7219-2013.
- Zhang, S., C. A. Stock, E. N. Curchitser, and R. Dussin, 2019: A numerical model analysis of the mean and seasonal nitrogen budget on the northeast u.s. shelf. *Journal of Geophysical Research: Oceans*, **124**, doi:10.1029/2018JC014308, URL https://agupubs.onlinelibrary.wiley.com/doi/abs/10.1029/2018JC014308, https://agupubs.onlinelibrary.wiley.com/doi/pdf/10.1029/2018JC014308.

**Response to reviewer 2 of: "Budget of the total nitrogen in the Yucatan Shelf: driving mechanisms through a physical-biogeochemical coupled model"**

S. Estrada-Allis on behalf of all co-authors.

**August 2019**

We thank the reviewer for the comments to our paper and for providing valuable suggestions to improve it. Our responses to each one of them follow (in blue). Modifications to the manuscript can be found in the attached document Tracking-changes.pdf, where new text is marked in blue and removed text in red.

**1 General Comments**

This work presents an estimation of the Total Nitrogen (TN) budget in the Yucatan Shelf (YS). The estimate is obtained using a coupled physical-biochemical model (ROMS), validated by some in-situ and satellite observations. The model solution is available for 9 year (2002-2010) while the in-situ observations used to validate the solution within the YS are available for Nov 2015. Physical processes that are relevant in explaining the estimated TN budget are identified and described. The main input of N is at the eastern boundary through the interaction of the western boundary current with the shelf break, presumably mainly due to Ekman transport at the bottom boundary layer. The imported N is then advected westward by the wind driven-circulation along the shelf. Most the N that enters the inner shelf (depths shallower than 50 m) is consumed by phytoplankton, and part of the N that enters the outer shelf (depths 50-250 m) is exported to the deep ocean in the west and northwest parts of the YS. This export of N is modulated by Coastally Trapped Waves with a typical period of 10 days.

I think this manuscript addresses a relevant scientific question within the scope of BG, and the modeling results suggest a very interesting case for the relevance of likely physical processes controlling or modulating the import and export of N in and out of the YS. However, I think some revisions are needed in terms of validating the model, justifying the model physics or at least acknowledging the limitations and implications for the estimated budget, and the

analysis and overall presentation of the work done. Below I outline some suggested revisions that might guide the authors when improving the manuscript.

Thanks for your comment. We have extended the Yucatan shelf (YS) validation by comparing the structure and velocity values with observations found in the literature (Sheinbaum et al., 2002; Athié et al., 2015; Sheinbaum et al., 2016). The velocity and its standard deviation are in a good agreement with the observed mean values reported in the literature (see P10 L22 and new appendix A4, Figure A9). A qualitatively comparison is made in terms of salinity with the observations reported in Enriquez et al. (2013), see Figure A6 in the appendix A of the revised manuscript and compared to Figure 9 from Enriquez et al. (2013).

We have added a point-by-point comparison of the salinity from the GOMEX IV cruise (see P9 L11-15) and a comparison of the mean chlorophyll vertical profiles observed during three different years in the YS (see new appendix A2, Figure A6) with successful results. In order to prove that the model reproduces the seasonality of the upwelling, the bottom shelf temperature is also shown and the seasonality of the upwelling isotherm of 22.5 °C (see P8 L27-31, Figure 3). Moreover, the model sea level anomaly over the YS is compared with altimetry data from satellite and the model current velocity with the Global Current product (see new appendix A3, Figure A7). Most of the content related to model validation is contained in section 3 and appendix A of the revised version. We hope that all these new model-data comparisons convince the reviewers of the capability of the model to reproduce main features of the observed variability in the region. Note that the size and number of figures in the new version of this paper has increased considerably. Those for validation are included in appendix A which may be shortened or added as supporting information instead of an appendix if necessary.

**2 Specific Comments**

**2.1 Model Validation**

While this study focuses on the YS and its vicinity, most of the convincing validation presented is for the whole Gulf of Mexico (GoM). There is a need to validate and characterize the background state in the YS in order to add credibility to the results. While I understand there is a scarcity of *in situ* observations in the YS, some vertical sections of temperature and salinity have been reported previously (e. g., Enriquez et al., 2013). There are also lots of *in situ* observations in the western boundary current (e.g., Sheinbaum et al., 2002, 2016) that could be used to convince the reader that the model physics is reliable. Specially those closer to the 250 m isobath if available. There is no need to do an exhaustive analysis of such observational data-sets, but to show a congruence between model and observed mean background state.

In agreement with the reviewer we have extended the model YS validation, see our previous response to the general comment and all section 3 and appendix

**A of the revised version.**

The *in situ* biochemical observations were taken in Nov 2015, while the model solution is available until 2012. Ideally one would have a solution contemporaneous with the observations but if that is not possible then the approach presented is the best next choice: compare the mean and standard deviation for Nov with the observations. But then one wonders is the agreement with observations will hold for other months given that the authors present long-term means for the budgets. The authors seem to suggest that seasonality is weak, but I think that needs to be shown. For instance, does the inner shelf remains well mixed throughout the year as it seems to be the case in Nov?, or does some stratification develops during summer and if so, how that affects the budget?.

The reviewer is right and our goal is just to show that the model is capable to reproduce basic statistics from available observations (mean profiles and standard deviations). To extend our validation of basic statistics (see also comment above for the T-S profiles), we have added a comparison between chlorophyll profiles from other years (August 2016 and July 2018) temporally and spatially averaged for the whole YS (see appendix A2 and Figure A6). The mean and standard deviation profiles of chlorophyll and salinity are presented to show the degree of variability within the shelf. Unfortunately, nutrient and particulate organic nitrogen observations are scarce.

Horizontal and vertical sections of the bottom shelf temperature are also added to the new manuscript to show that the model is capable of representing the seasonal variability (P8 L27-31, Figure 3). In Figures 3c and d one can see that the isotherm of 22.5 °C, which represents the upwelling waters at Cape Catoche outcrops to the shelf during spring. Whereas during autumn months, the isotherm is not able to reach the shelf in agreement with observations reported in (Merino, 1997) and consistent with the strengthening/weakening of the upwelling of nutrients and chlorophyll during those seasons. The stratification of the shelf is a complex issue that depends on the proximity of oceanic waters with the coast and freshwater sources. However, it is a very interesting question that merits a more comprehensive study using higher horizontal resolution than the one used presently.

Similarly, more analysis can be done using satellite products to validate the circulation in the YS. This is specially important to compensate for the scarcity of *in situ* observations. How does the observed Sea Level Anomaly correlates with that of the model in the YS? The authors could for instance compare the annual cycles of the aviso SLA and the model (without Mean Dynamic Topography) to convince the readers that the background model physics on the shelf is reliable. Is the pressure gradient across the shelf break well resolved by the model? A similar comparison could be done with scatterometer winds. The use of OSCAR currents, while very low resolution, might be possible given the wide YS. SST is not a robust validation set since the authors are using bulk fluxes at the surface and the CFSR model (I assume) is forced by satellite SST. Therefore the model is implicitly nudged to observed SST via the 2 m air temperature.

Following the reviewer suggestion, we improved the model validation by comparing Sea Level Anomaly from AVISO and the model (see new APPENDIX A, and Figure A7). However, one should be careful when using the AVISO product in shallow areas and close to the shore. Instead of OSCAR, we use GlobalCurrent products (http://www.globcurrent.org) to evaluate the shelf model velocity. This choice is based on the fact that this product includes both Ekman and geostrophic velocities, with Ekman being an important contribution to the circulation on the shelf, Those comparisons suggest pressure gradient across the shelf break is properly resolved. Validation of the CFSR reanalysis winds in the GoM has been carried by other authors (e.g Chawla et al 2013, https://doi.org/10.1016/j.ocemod.2012.07.005). Thank you for your comment regarding the SSTs for validation, this is taken into consideration for the basinscale model validation.

**2.2 Model Physics**

One the main findings reported in the manuscript is that the input of N in the southeast part of the shelf comes from Ekman transport at the bottom boundary layer. The vertical stretching used was developed to study the surface vorticity balance and while this might be a wise choice for the whole GoM, it is not the best for the YS NT budget given the relevance of the bottom boundary layer. This limitation needs to be further analyzed and acknowledged in the manuscript. The authors could for instance estimate the bottom boundary layer thickness (a common estimate is thickness 0.4\*frictionalvelocity/Corilisparamater, where frictional velocity is estimated from the bottom stress). How this thickness compares to the thickness the the first model sigma layer for the 250 m isobath?. That is, how well is the bottom boundary layer resolved?. If it is not very well resolved I still think the bulk characteristics of the budget will hold (i.e., the main sources and sinks and the TN pathways), but this limitation needs to be acknowledged.

We agree with the reviewer. The chosen vertical stretching (Azevedo Correia de Souza et al., 2015) was developed to provide higher resolution near the surface. In fact, in most areas of the study region important fluxes are concentrated near the surface (e.g. Figure 10, previous manuscript). In shallow areas there is high resolution (because of the use of sigma-s coordinates) but one may expect some issues on the slopes close to the shelf but deep. Certainly, we agree that the bottom Ekman layer needs to be as well represented as possible, particularly if we think bottom Ekman layer transports may be important for the dynamics. We estimated the size of the bottom Ekman layer using standard formulas based on bottom stress and friction velocity for homogeneous and stratified fluids (Cushman-Roisin and Beckers, 2011). Analysis (not shown), indicates the bottom Ekman layer width is on average between 10-30 m whereas the model bottom layer width is about 20 m. Since we estimate bottom Ekman layer transports using the vertical size of the near-bottom

cells, our estimates could be somewhat biased. We acknowledge this in the new version of the manuscript (P16 L32-35).

**2.3 Technical Corrections**

Abstract:

L4: I think it should be "Coastal-Trapped Waves" or "Coastally Trapped Waves"

Thank you, this has been corrected as "Coastal-Trapped Waves" in the revised version.

L8: Define DIN or spell it out.

The DIN term is spelled out in the Abstract in the revised version. Introduction:

L6: Processes

Thank you, this has been corrected in the revised version (see P2 L23) Section 2.1: L23: run

Thank you, this has been corrected in the revised version (see P5 L2)

L25: In what sense it is "consistent with the observational data"?

To be time-consistent, i.e., in the sense that the model and observations match in the same time range, as far as possible (P5 L3).

L27: Is the boundary condition daily means? Monthly? What is done with the tides?

The boundary conditions are daily averaged. The tides are hourly and added as a separate spectral forcing at the boundaries. This is clarified in the revised version (see P5 L7)

L30: Mention that the surface fluxes are computed using the bulk formulae for the marine boundary layer and provide a reference.

We mentioned this in the revised version, thank you (see P5 L312)

Section 2.2

Formula (2) is confusing. While the model is in sigma levels the vertical integration is in depth (I hope so!) where the ?dz? is the corresponding sigma layer thickness. The formula also uses summation indexes (x1:xn an y1:yn) as limits for integrals, which is very confusing. One option could be just to do a single integral over area elements dA. Also this expression is equated with the same abbreviation used for expression (1). Maybe use an overbar??

Indeed, the model is in sigma levels and the vertical integration is in depth from layer thicknesses. We agree that formula (2) is confusing. To make the text more readable we decided to remove this equation and is only described in the text of the revised manuscript (see P6 L11-13)

L19: How the unknown groundwater sources were "inversely estimated"? How many? Where? What are their fluxes?

We regret the use of the wording "inversely estimated" which has caused a lot of unnecessary confusion. All we did was to use all possible available information that we were aware of regarding fresh water and nutrient fluxes (temperature and salinity too) at the Mexican GoM coast. We also used near coastal nutrient measurements. Based on that limited information we computed a monthly climatology to force the model. It is important to mention that no chlorophyll data were used for this (see reply to comment 18 of reviewer 1). We have added a new appendix C to present examples of the monthly climatology of Mexican rivers (Usumacinta and Grijalva) and freshwater sources ("cenotes" and lagoons in the YS) used in this study and compared with the Mississippi and Atchafalaya river system for which we have more information. We have rephrased the description of the freshwater sources on P7 L11-14, and added appendix C for details in the revised version.

Section 3.1 is good enough but more work is needed in section 3.2 (regional validation) as suggested above.

Thank you for this suggestion. More validation is presented in the new version of the manuscript (see response to the general comments above, new section 3 and appendix A).

P7,L3: "Below 55 m the modeled and "?

This paragraph has been changed.

P7,L4: "since there is no data assimilation". This sentence doesn't make much sense since you are not comparing contemporaneous values.

Thank you, this sentence has been erased in the revised version (see P10 L7).

P7,L4: "and ARE in"

This paragraph has been changed.

P7,L5: "To the best of our knowledge this is the first modeling study focusing on the nitrate budget of the YS."

This paragraph has been changed.

P7,L16: Usually the word "trend" is used for time, so I'll suggest erasing "along time".

Thank you, this has been corrected in (see P10 L26).

P7,L17: How Fig 8e compares to the model equivalent Chl? Does that also show the trend? Given the very large std in the Chl I can't help wonder how big are the error bars for the trend.

The model Chl compares well with the satellite Chl, as shown in Figure 1. This figure is added to Figure 8c in the revised version. The model also exhibits a positive trend. The large std of the Chl is probably related to the strong variability given by upwelling episodes over the shelf. Even in the presence of this high variability a trend in both observations and model (just for the simulation period) can be inferred although with large confidence intervals indeed, as shown in the next figure for the linear fit of the time series averaged over the YS of satellite Chl measurements and corresponding model Chl values. The dotted lines show the linear fit to the series and the thinner dotted lines the 95% confidence interval of the fits. Overall satellite Chl averaged over the YS and over the period of time between 2002-2010 is  $0.38 \pm 0.09 \text{ mgChl m}^{-3}$  with corresponding values for the model being a mean of  $0.36 \pm 0.13 \text{ mgChl m}^{-3}$  for the same area and time period.

P7,L21: "associated with the seasonal cycle AND INTERANNUAL variability". The trend is presumably related to the interannual part. It could be good to compare that with the physics of the model such as the western boundary

Figure 1: Time series of satellite surface Chl (mgChl m-3) (thick gray line) and model (thick black line), averaged for the period of years between 2002-2010 and over the YS area. Thick dashed lines are the fitted trends for each temporal series, and thin dashed lines are the corresponding 95% confidence interval. Equations for the linear fits are  $Chl_{trend} = 0.0010$  month + 0.28 for satellite, and  $Chl_{trend} = 0.0011$  month + 0.30 for the model chlorophyll trend.

current strength, across-shelf pressure gradient, etc.

We agree with the reviewer. Determining the origins of this trend is an interesting problem that requires further research. Based on the study of (Varela et al., 2018) we tried to establish a link between the trend and dynamic indices for example, the strength of the Yucatan Current. This index (Yucatan Current strength) has no clear seasonality in the model (Figure 2 first panel), and depicts a large std higher than the mean. No significant correlation was found with the positive trend of nitrogen or chlorophyll (Figure 2 second panel). We also investigated the closeness of the Yucatan Current core to the shelf but that was also uncorrelated with the chlorophyll trend (Figure 2 third panel). The same happens with changes in the cross-shelf sea level anomaly. Therefore the question needs to be addressed in future work. This is now mentioned in P10 L28-31 of the revised version.

---

## Referee Report (RR1)

Review of Estrada-Ellis et al. "Budget of the total nitrogen in the Yucatan Shelf: driving mechanisms through a physical-biogeochemical coupled model"

General comments:
I reviewed a previous version of this manuscript and noted that the current version of the manuscript was greatly improved from the earlier draft. I commend the authors for the significant revision. In particular, they have added many additional figures to demonstrate the model's skill at reproducing observations in the Yucatan shelf region. I still have some problems with the manuscript though. In particular, I am still not satisfied with the description of how total nitrogen (TN) in the model equates to total nitrogen in the real world. My concern remains that the model is missing nitrogen in the form of dissolved organic nitrogen (DON). The model description continues to be unclear in this regard. How did you set the boundary condition concentrations at the edges of the shelf modeling domain for LDet and Sdet state variables? How did you set the LDet and SDet in rivers and freshwater inputs? In the model, the TN seems to be comprised of DIN and PON (see figure 8). Where is the DON pool accounted for? Please clarify this in the text. Also, now that I see the model-data comparisons for NO3 (Figure 6) it appears to me that the modeled NO3 may be 2-3x lower than the observed NO3 values. It is reported that the mean bias is on the order of -1.7 mmol m-3. The discussion of how this bias may affect the magnitudes of the estimated N budget fluxes is addressed in the appendix. I think these uncertainties should be included in the main text prior to the 'Concluding Remarks' section.

Specific Comments:
pg 2, line 28: replace 'responsible of' with 'responsible for'

Pg 3, line 12: perhaps rephrase this to 'Regarding freshwater inflow, a significant source to the YS is related to submarine groundwater discharge (SGD) …

Pg 3, line 22: insert 'of' between 'some the'

Pg 3, line 26: missing period at end of sentence

Pg 4, lines 19-20: The sources of data for initial and boundary NO3, NH3, and Chl are reported. How did you specify boundary conditions for LDet, SDet, Phy, and Zoo?

Pg 5, line 17: What about LDet and SDet in freshwater and river inputs? Similar to the comment above about the boundary conditions, how did you estimate river inputs?

Pg 7, lines 21-33: move the paragraph discussing model-data comparisons of NO3 before discussing Chl to be consistent with figure numbering and presentation.

Pg 7, lines 32-33: There are other studies besides Xue et al that do report N budgets normalized by area or length. For example, see Walsh et al. 1989 or Lehrter et al. 2013. At a minimum, you could provide the spatial area of your inner shelf and outer shelf domains shown in Table 1or for the boxes shown in Fig. 2a so that a reader could calculate area normalized rates.

Pg 10, line 7: I don't recall seeing SLA defined

END OF REVIEW

---

## Referee Report (RR2)

*Review of the manuscript* **"Budget of the total nitrogen in the Yucatan Shelf: driving mechanisms through a physical-biogeochemical coupled model"** *by Sheila N. Estrada-Allis et al.*

**General comments**

This revised manuscript presents an estimation of the Total Nitrogen (TN) budget in the Yucatan Shelf (YS). The estimate is obtained using a coupled physical-biochemical model (ROMS), validated by in-situ and satellite observations. The model solution is available for 9 year (2002-2010) while the in-situ observations used to validate the solution within the YS are available for Nov 2015. Physical processes that are relevant in explaining the estimated TN budget are identified and described. The main input of N is at the eastern boundary through the interaction of the western boundary current with the shelfbreak, presumably mainly due to Ekman transport at the bottom boundary layer. The imported N is then advected westward by the wind driven-circulation along the shelf. Most the N that enters the inner shelf (depths shallower than 50 m) is consumed by phytoplankton, and part of the N that enters the outer shelf (depths 50-250 m) is exported to the deep ocean in the west and northwest parts of the YS. This export of N is modulated by intraseasonal wind and Coastally Trapped Waves.

In the revision of the earlier version of this manuscript I expressed my concerns about the validation of the physical component of the model and the need to justify or acknowledge the limitations of the model configuration used. Both concerns have been addressed in this second revision.

The model validation of the revised manuscript includes comparison with EKE and variance maps from aviso, satellite SST, mixed layer depth from ARGO, satellite-derived surface Chl, Chl from profiling floats, surface currents form the GlobCurrent product, and comparison with some hydrographic profiles and sections of mean velocity from moorings along the eastern side of the YS. In addition, it is now acknowledged that the model vertical resolution at the shelf break (~20 m) cannot resolve the details of the bottom boundary layer. However I do not believe this to be a limiting factor for this exploratory study which aims to provide a first order approximation to the TN budget. The bulk properties of the bottom Ekman transport can be inferred just as surface wind stress is used to provide a bulk estimate of the Ekman transport near the surface.

As mentioned before, the manuscript addresses a relevant scientific question within the scope of BG and the modeling results suggest a very interesting case for the relevance of likely physical processes controlling or modulating the import and export of N in and out of the YS. While the new validation provides more confidence on the model results, I still think the manuscript needs to be highly revised for grammatical and redaction errors. I noticed that the quality of the

manuscript in terms of typos, clarity of the statements, grammar, etc. degrades towards the end. Please revise it carefully.

P1, L11: Maybe change to "due to enhanced bottom Ekman transport"?
Figure 1. The caption says "The seas of the Deep Gulf of Mexico, Campeche and Caribbean are also shown in (a). The inner and outer Yucatan Shelf is denoted in (c)." However the names are not shown in the figure panels.

Is Fig 12 of any use?. Not much information can be extracted from the time-series plots. The wavelet power spectrum is somehow useful but maybe a better colormap could help to emphasize the energy peaks. Revise "lanksos".

Fig 13. Maybe plot just the amplitude, not the phase?

P12- L14: Revise. Maybe "We present results from a 9-year simulation of a physical-biochemical coupled model for the GoM, focusing on the YS."?

P16, L16: Erase or revise this sentence.

---

## Author Response (AR2)

**Response to reviewer of "Budget of the total nitrogen in the Yucatan Shelf: driving mechanisms through a physical-biogeochemical coupled model"**

S. Estrada-Allis on behalf of all co-authors.

November 2019

We would like to thank the reviewer for this second round of revision, which helped us to improve the article quality. As in the previous revision, following are our responses (in blue) to all comments. The modifications to the original manuscript can be found in the attached document named Tracking-Changes-v2.pdf, where new text is marked in blue and removed text in red.

**1    General Comments**

I reviewed a previous version of this manuscript and noted that the current version of the manuscript was greatly improved from the earlier draft. I commend the authors for the significant revision. In particular, they have added many additional figures to demonstrate the model's skill at reproducing observations in the Yucatan shelf region. I still have some problems with the manuscript though. In particular, I am still not satisfied with the description of how total nitrogen (TN) in the model equates to total nitrogen in the real world. My concern remains that the model is missing nitrogen in the form of dissolved organic nitrogen (DON).

The biological model used in this study is an N-cycle model based on modifications of the Fasham's model (Fasham et al., 1990) made by Fennel et al. (2006) and Fennel et al. (2011). The biological model includes seven state variables: phytoplankton, zooplankton, nitrate, ammonium, small and large detritus, and chlorophyll. The interaction of these variables can be seen in the scheme of Figure 1 in Fennel et al. (2006). Since we do not modify the original model equations and assumptions we refer the readers to Fennel et al. (2006) for a detailed description of the model.

In essence, the model takes TN as the sum of DIN and PON, that is TN = NO3 + NH4 + LdetN + LdetC + SdetN + SdetC + Zoo + Phy. Denitrification is the main sink of nitrogen, mainly in shelves (Fennel et al., 2006; Xue et al., 2013). In the sediment compartment of the model, the organic matter is remineralized immediately as an influx of ammonium at the sediment-water interface (see Appendix A of Fennel et al. (2006)) and represents a key feature of this model. The model does not include an explicit compartment for nitrogen in the form of DON although it can be included as in the work of Druon et al. (2010) which adds semi-labile DOC and DON as state variables to the original Fennel et al. (2006) model. They comment on the difficulties of validating the model with observations and highlight open questions even in the definition of both DOC and DON pools (see also ?). Considering these difficulties and uncertainties, our approach is to use, initially, more basic models to understand their capabilities and build/employ more comprehensive ones upon them later on; so the inclusion of DON and/or DOC compartments is left for future studies.

To clarify this issue, the new manuscript recognizes the absence of a DON compartment and includes information about initial and boundary conditions. Pages and lines where the text has been modified are indicated in the comments below.

The model description continues to be unclear in this regard. How did you set the boundary condition concentrations at the edges of the shelf modeling domain for LDet and Sdet state variables? How did you set the LDet and SDet in rivers and freshwater inputs?

The boundary conditions for both the physical and biological model are set at the edges of the model domain ("box"), that is, at the North, South, East and West boundaries of the Gulf of Mexico shown in Figure 1(a) of the manuscript. Figure 1 has been modified and now includes the boundaries used to compute the TN cross-shelf fluxes over the Yucatan shelf. In the case of LDet and SDet state variables, a small and constant concentration of 0.1 mmol N m$^{-3}$ following (e.g., Fennel et al., 2006, 2011; Xue et al., 2013) is used initially and at the boundaries of the whole GoM domain. We tried to clarify this in our previous reply. It is related to the rapid adjustment of these variables to more realistic values produced by the model biogeochemical dynamics. These boundaries are far away from the Yucatan shelf and therefore the fluxes across the inner and outer Yucatan shelf determined internally in the model (not imposed) are not impacted by possible inconsistencies at the GoM open boundaries. This is now explicitly written in the manuscript (P5 L18-25). Given the lack of data for Mexican rivers and ground water fluxes, the same approach is followed for freshwater inputs as done also by Xue et al. (2013). This is now explicitly explained in P6 L19-22 of the revised version.

In the model, the TN seems to be comprised of DIN and PON (see figure 8). Where is the DON pool accounted for? Please clarify this in the text.

As the reviewer noted, the TN is the sum of DIN and PON compartments. DON is not included as state variable in the Fennel model (see previous repky to general comments). Following the reviewer's suggestion we have clarified this aspect of the model in the text (P5 L27-31 and P6 L1-2).

Also, now that I see the model-data comparisons for NO3 (Figure 6) it appears to me that the modeled NO3 may be 2-3x lower than the observed

NO3 values. It is reported that the mean bias is on the order of -1.7 mmol m-3. The discussion of how this bias may affect the magnitudes of the estimated N budget fluxes is addressed in the appendix. I think these uncertainties should be included in the main text prior to the 'Concluding Remarks' section.

Following the reviewer's suggestion the discussion of how the bias may affect the budget of the TN is now included in a new section prior of 'Concluding Remarks' (P13 new Section 5). As the reviewer noted, some observed values may exceed 2 or 3 times the modeled NO3 concentrations. However, notice that: (1) this is a point-by-point comparison, and the model does not replicate the real hydrodynamic conditions at the moment that the surveys were taken; (2) the standard deviations of modeled and observed NO3 are in good agreement, indicating that observed and model statistics are consistent; and (3) the model bias estimate is a mean taken for the whole shelf. The upwelling area, which is in fact our area of interest, shows better agreement with observed NO3 profiles and a bias of -0.7 mmolN $m^{-3}$. This is now included in revised manuscript P8 L15-18.

**2 Specific Comments**

pg 2, line 28: replace 'responsible of' with 'responsible for' Thank you, this has been modified in P2 L30.

Pg 3, line 12: perhaps rephrase this to 'Regarding freshwater inflow, a significant source to the YS is related to submarine groundwater discharge (SGD) ... Thank you, this has been modified in P3 L15-16.

Pg 3, line 22: insert 'of' between 'some the' Thank you, this has been modified in P3 L26.

Pg 3, line 26: missing period at end of sentence Thank you, this has been modified in P3 L22.

Pg 4, lines 19-20: The sources of data for initial and boundary NO3, NH3, and Chl are reported. How did you specify boundary conditions for LDet, SDet, Phy, and Zoo? The values set for the boundary conditions of LDet, SDet, Phy and Zoo, are reported in P5 L20-22. These are specified as small and constant positive value of 0.1 mmolN $m^-$3.

Pg 5, line 17: What about LDet and SDet in freshwater and river inputs? Similar to the comment above about the boundary conditions, how did you estimate river inputs? As explained in general comments, we have set the detritus pool as a small and constant quantity of 0.1 mmolN $m^{-3}$, following the studies of Fennel et al. (2011) and Xue et al. (2013). This is now explicitly noted in the text in P6 L20-22.

Pg 7, lines 21-33: move the paragraph discussing model-data comparisons of NO3 before discussing Chl to be consistent with figure numbering and presentation. Thank you for notice this, we have moved the figure regarding the NO3 comparison after the figure of the Chl comparison, in order to keep consistence within the text.

Pg 7, lines 32-33: There are other studies besides Xue et al that do report

N budgets normalized by area or length. For example, see Walsh et al. 1989 or Lehrter et al. 2013. At a minimum, you could provide the spatial area of your inner shelf and outer shelf domains shown in Table 1or for the boxes shown in Fig. 2a so that a reader could calculate area normalized rates. In agreement with the reviewer suggestion we have added the area of the inner and outer shelf of Yucatan in the text, in P8 L23-25. Moreover, we also added the mean TN concentration in mmolN for the inner and outer shelf in the same paragraph.

Pg 10, line 7: I don't recall seeing SLA defined Thank you for notice this, the SLA definition is now in P11 L3.


First of all, we would like to thank the reviewer for this second round of revision. His comments helped us to improve the article quality considerably. As in the previous revision, following are our responses (in blue) to all comments. The modifications to the original manuscript can be found in the attached document named Tracking-Changes-v2.pdf, where new text is marked in blue and removed text in red.

**1 General Comments**

This revised manuscript presents an estimation of the Total Nitrogen (TN) budget in the Yucatan Shelf (YS). The estimate is obtained using a coupled physical-biochemical model (ROMS), validated by in-situ and satellite observations. The model solution is available for 9 year (2002-2010) while the in-situ observations used to validate the solution within the YS are available for Nov 2015. Physical processes that are relevant in explaining the estimated TN budget are identified and described. The main input of N is at the eastern boundary through the interaction of the western boundary current with the shelfbreak, presumably mainly due to Ekman transport at the bottom boundary layer. The imported N is then advected westward by the wind driven-circulation along the shelf. Most the N that enters the inner shelf (depths shallower than 50 m) is consumed by phytoplankton, and part of the N that enters the outer shelf (depths 50-250 m) is exported to the deep ocean in the west and northwest parts of the YS. This export of N is modulated by intraseasonal wind and Coastally Trapped Waves. In the revision of the earlier version of this manuscript I expressed my concerns about the validation of the physical component of the model and the need to justify or acknowledge the limitations of the model configuration used. Both concerns have been addressed in this second revision. The model validation of

the revised manuscript includes comparison with EKE and variance maps from aviso, satellite SST, mixed layer depth from ARGO, satellite-derived surface Chl, Chl from profiling floats, surface currents form the GlobCurrent product, and comparison with some hydrographic profiles and sections of mean velocity from moorings along the eastern side of the YS. In addition, it is now acknowledged that the model vertical resolution at the shelf break ( 20 m) cannot resolve the details of the bottom boundary layer. However I do not believe this to be a limiting factor for this exploratory study which aims to provide a first order approximation to the TN budget. The bulk properties of the bottom Ekman transport can be inferred just as surface wind stress is used to provide a bulk estimate of the Ekman transport near the surface. As mentioned before, the manuscript addresses a relevant scientific question within the scope of BG and the modeling results suggest a very interesting case for the relevance of likely physical processes controlling or modulating the import and export of N in and out of the YS. While the new validation provides more confidence on the model results, I still think the manuscript needs to be highly revised for grammatical and redaction errors. I noticed that the quality of the manuscript in terms of typos, clarity of the statements, grammar, etc. degrades towards the end. Please revise it carefully.

The manuscript has been revised carefully in terms of English style and correction of typos. The new version and the corresponding changes are in the file Tracking-Changes-v2.pdf. Than you for your suggestions and comments.

P1, L11: Maybe change to "due to enhanced bottom Ekman transport"? Thank you, the phrase has been changed in the Abstract P1 L12.

Figure 1. The caption says "The seas of the Deep Gulf of Mexico, Campeche and Caribbean are also shown in (a). The inner and outer Yucatan Shelf is denoted in (c)." However the names are not shown in the figure panels. Thank you for notice this, the name are now shown in Figure 1, panel (a) and (c).

Is Fig 12 of any use?. Not much information can be extracted from the time-series plots. The wavelet power spectrum is somehow useful but maybe a better colormap could help to emphasize the energy peaks. Revise "lanksos". We are in agreement with the reviewer in that the time-series plots does not give substantial information. Only wavelet spectrum are keep with different colorbar to emphasize the energy peaks as suggested by the reviewer. Thank you for your comment, the typo has been also corrected.

Fig 13. Maybe plot just the amplitude, not the phase? Although the amplitude is part of this type of analysis, we have decided to plot the phase to show that the signals are not only statistically coherent each other but also their significantly peaks are in-phase (0 degrees) or anti-phase (180 degrees out of phase). This is relevant since indicates that SLA and TN cross-shelf fluxes can have coherent energy spectrum peaks at the same frequency band but in anti-phase, that is, when SLA turn on negative values,, the TN cross-shelf fluxes are positive, i.e., offshore fluxes, and vice-versa. The energy spectrum peaks between along-shelf wind-stress and TN cross-shelf flux, are in phase, suggesting that an increase in the wind-stress leads to positive TN cross-shelf fluxes offshore by surface Ekman transport, which is an expected result in the western boundary of the Yucatan shelf.

[revised manuscript text omitted]

S. Estrada-Allis on behalf of all co-authors.

November 2019

We would like to thank the reviewer for this second round of revision, which helped us to improve the article quality. As in the previous revision, following are our responses (in blue) to all comments. The modifications to the original manuscript can be found in the attached document named Tracking-Changes-v2.pdf, where new text is marked in blue and removed text in red.

**1 General Comments**

I reviewed a previous version of this manuscript and noted that the current version of the manuscript was greatly improved from the earlier draft. I commend the authors for the significant revision. In particular, they have added many additional figures to demonstrate the model's skill at reproducing observations in the Yucatan shelf region. I still have some problems with the manuscript though. In particular, I am still not satisfied with the description of how total nitrogen (TN) in the model equates to total nitrogen in the real world. My concern remains that the model is missing nitrogen in the form of dissolved organic nitrogen (DON).

The biological model used in this study is an N-cycle model based on modifications of the Fasham's model (Fasham et al., 1990) made by Fennel et al. (2006) and Fennel et al. (2011). The biological model includes seven state variables: phytoplankton, zooplankton, nitrate, ammonium, small and large detritus, and chlorophyll. The interaction of these variables can be seen in the scheme of Figure 1 in Fennel et al. (2006). Since we do not modify the original model equations and assumptions we refer the readers to Fennel et al. (2006) for a detailed description of the model.

In essence, the model takes TN as the sum of DIN and PON, that is TN = NO3 + NH4 + LdetN + LdetC + SdetN + SdetC + Zoo + Phy. Denitrification is the main sink of nitrogen, mainly in shelves (Fennel et al., 2006; Xue et al., 2013). In the sediment compartment of the model, the organic matter is remineralized immediately as an influx of ammonium at the sediment-water interface (see Appendix A of Fennel et al. (2006)) and represents a key feature of this model. The model does not include an explicit compartment for nitrogen in the form of DON although it can be included as in the work of Druon et al. (2010) which adds semi-labile DOC and DON as state variables to the original Fennel et al. (2006) model. They comment on the difficulties of validating the model with observations and highlight open questions even in the definition of both DOC and DON pools (see also ?). Considering these difficulties and uncertainties, our approach is to use, initially, more basic models to understand their capabilities and build/employ more comprehensive ones upon them later on; so the inclusion of DON and/or DOC compartments is left for future studies.

To clarify this issue, the new manuscript recognizes the absence of a DON compartment and includes information about initial and boundary conditions. Pages and lines where the text has been modified are indicated in the comments below.

The model description continues to be unclear in this regard. How did you set the boundary condition concentrations at the edges of the shelf modeling domain for LDet and Sdet state variables? How did you set the LDet and SDet in rivers and freshwater inputs?

The boundary conditions for both the physical and biological model are set at the edges of the model domain ("box"), that is, at the North, South, East and West boundaries of the Gulf of Mexico shown in Figure 1(a) of the manuscript. Figure 1 has been modified and now includes the boundaries used to compute the TN cross-shelf fluxes over the Yucatan shelf. In the case of LDet and SDet state variables, a small and constant concentration of 0.1 mmol N m$^{-3}$ following (e.g., Fennel et al., 2006, 2011; Xue et al., 2013) is used initially and at the boundaries of the whole GoM domain. We tried to clarify this in our previous reply. It is related to the rapid adjustment of these variables to more realistic values produced by the model biogeochemical dynamics. These boundaries are far away from the Yucatan shelf and therefore the fluxes across the inner and outer Yucatan shelf determined internally in the model (not imposed) are not impacted by possible inconsistencies at the GoM open boundaries. This is now explicitly written in the manuscript (P5 L18-25). Given the lack of data for Mexican rivers and ground water fluxes, the same approach is followed for freshwater inputs as done also by Xue et al. (2013). This is now explicitly explained in P6 L19-22 of the revised version.

In the model, the TN seems to be comprised of DIN and PON (see figure 8). Where is the DON pool accounted for? Please clarify this in the text.

As the reviewer noted, the TN is the sum of DIN and PON compartments. DON is not included as state variable in the Fennel model (see previous repky to general comments). Following the reviewer's suggestion we have clarified this aspect of the model in the text (P5 L27-31 and P6 L1-2).

Also, now that I see the model-data comparisons for NO3 (Figure 6) it appears to me that the modeled NO3 may be 2-3x lower than the observed

NO3 values. It is reported that the mean bias is on the order of -1.7 mmol m-3. The discussion of how this bias may affect the magnitudes of the estimated N budget fluxes is addressed in the appendix. I think these uncertainties should be included in the main text prior to the 'Concluding Remarks' section.

Following the reviewer's suggestion the discussion of how the bias may affect the budget of the TN is now included in a new section prior of 'Concluding Remarks' (P13 new Section 5). As the reviewer noted, some observed values may exceed 2 or 3 times the modeled NO3 concentrations. However, notice that: (1) this is a point-by-point comparison, and the model does not replicate the real hydrodynamic conditions at the moment that the surveys were taken; (2) the standard deviations of modeled and observed NO3 are in good agreement, indicating that observed and model statistics are consistent; and (3) the model bias estimate is a mean taken for the whole shelf. The upwelling area, which is in fact our area of interest, shows better agreement with observed NO3 profiles and a bias of -0.7 mmolN m$^{-3}$. This is now included in revised manuscript P8 L15-18.

**2 Specific Comments**

pg 2, line 28: replace 'responsible of' with 'responsible for' Thank you, this has been modified in P2 L30.

Pg 3, line 12: perhaps rephrase this to 'Regarding freshwater inflow, a significant source to the YS is related to submarine groundwater discharge (SGD) . . . Thank you, this has been modified in P3 L15-16.

Pg 3, line 22: insert 'of' between 'some the' Thank you, this has been modified in P3 L26.

Pg 3, line 26: missing period at end of sentence Thank you, this has been modified in P3 L22.

Pg 4, lines 19-20: The sources of data for initial and boundary NO3, NH3, and Chl are reported. How did you specify boundary conditions for LDet, SDet, Phy, and Zoo? The values set for the boundary conditions of LDet, SDet, Phy and Zoo, are reported in P5 L20-22. These are specified as small and constant positive value of 0.1 mmolN m$^-$3.

Pg 5, line 17: What about LDet and SDet in freshwater and river inputs? Similar to the comment above about the boundary conditions, how did you estimate river inputs? As explained in general comments, we have set the detritus pool as a small and constant quantity of 0.1 mmolN m$^{-3}$, following the studies of Fennel et al. (2011) and Xue et al. (2013). This is now explicitly noted in the text in P6 L20-22.

Pg 7, lines 21-33: move the paragraph discussing model-data comparisons of NO3 before discussing Chl to be consistent with figure numbering and presentation. Thank you for notice this, we have moved the figure regarding the NO3 comparison after the figure of the Chl comparison, in order to keep consistence within the text.

Pg 7, lines 32-33: There are other studies besides Xue et al that do report

N budgets normalized by area or length. For example, see Walsh et al. 1989 or Lehrter et al. 2013. At a minimum, you could provide the spatial area of your inner shelf and outer shelf domains shown in Table 1or for the boxes shown in Fig. 2a so that a reader could calculate area normalized rates. In agreement with the reviewer suggestion we have added the area of the inner and outer shelf of Yucatan in the text, in P8 L23-25. Moreover, we also added the mean TN concentration in mmolN for the inner and outer shelf in the same paragraph.

Pg 10, line 7: I don't recall seeing SLA defined Thank you for notice this, the SLA definition is now in P11 L3.

First of all, we would like to thank the reviewer for this second round of revision. His comments helped us to improve the article quality considerably. As in the previous revision, following are our responses (in blue) to all comments. The modifications to the original manuscript can be found in the attached document named Tracking-Changes-v2.pdf, where new text is marked in blue and removed text in red.

**1 General Comments**

This revised manuscript presents an estimation of the Total Nitrogen (TN) budget in the Yucatan Shelf (YS). The estimate is obtained using a coupled physical-biochemical model (ROMS), validated by in-situ and satellite observations. The model solution is available for 9 year (2002-2010) while the in-situ observations used to validate the solution within the YS are available for Nov 2015. Physical processes that are relevant in explaining the estimated TN budget are identified and described. The main input of N is at the eastern boundary through the interaction of the western boundary current with the shelfbreak, presumably mainly due to Ekman transport at the bottom boundary layer. The imported N is then advected westward by the wind driven-circulation along the shelf. Most the N that enters the inner shelf (depths shallower than 50 m) is consumed by phytoplankton, and part of the N that enters the outer shelf (depths 50-250 m) is exported to the deep ocean in the west and northwest parts of the YS. This export of N is modulated by intraseasonal wind and Coastally Trapped Waves. In the revision of the earlier version of this manuscript I expressed my concerns about the validation of the physical component of the model and the need to justify or acknowledge the limitations of the model configuration used. Both concerns have been addressed in this second revision. The model validation of

the revised manuscript includes comparison with EKE and variance maps from aviso, satellite SST, mixed layer depth from ARGO, satellite-derived surface Chl, Chl from profiling floats, surface currents form the GlobCurrent product, and comparison with some hydrographic profiles and sections of mean velocity from moorings along the eastern side of the YS. In addition, it is now acknowledged that the model vertical resolution at the shelf break ( 20 m) cannot resolve the details of the bottom boundary layer. However I do not believe this to be a limiting factor for this exploratory study which aims to provide a first order approximation to the TN budget. The bulk properties of the bottom Ekman transport can be inferred just as surface wind stress is used to provide a bulk estimate of the Ekman transport near the surface. As mentioned before, the manuscript addresses a relevant scientific question within the scope of BG and the modeling results suggest a very interesting case for the relevance of likely physical processes controlling or modulating the import and export of N in and out of the YS. While the new validation provides more confidence on the model results, I still think the manuscript needs to be highly revised for grammatical and redaction errors. I noticed that the quality of the manuscript in terms of typos, clarity of the statements, grammar, etc. degrades towards the end. Please revise it carefully.

The manuscript has been revised carefully in terms of English style and correction of typos. The new version and the corresponding changes are in the file Tracking-Changes-v2.pdf. Than you for your suggestions and comments.

P1, L11: Maybe change to "due to enhanced bottom Ekman transport"? Thank you, the phrase has been changed in the Abstract P1 L12.

Figure 1. The caption says "The seas of the Deep Gulf of Mexico, Campeche and Caribbean are also shown in (a). The inner and outer Yucatan Shelf is denoted in (c)." However the names are not shown in the figure panels. Thank you for notice this, the name are now shown in Figure 1, panel (a) and (c).

Is Fig 12 of any use?. Not much information can be extracted from the time-series plots. The wavelet power spectrum is somehow useful but maybe a better colormap could help to emphasize the energy peaks. Revise "lanksos". We are in agreement with the reviewer in that the time-series plots does not give substantial information. Only wavelet spectrum are keep with different colorbar to emphasize the energy peaks as suggested by the reviewer. Thank you for your comment, the typo has been also corrected.

Fig 13. Maybe plot just the amplitude, not the phase? Although the amplitude is part of this type of analysis, we have decided to plot the phase to show that the signals are not only statistically coherent each other but also their significantly peaks are in-phase (0 degrees) or anti-phase (180 degrees out of phase). This is relevant since indicates that SLA and TN cross-shelf fluxes can have coherent energy spectrum peaks at the same frequency band but in anti-phase, that is, when SLA turn on negative values,, the TN cross-shelf fluxes are positive, i.e., offshore fluxes, and vice-versa. The energy spectrum peaks between along-shelf wind-stress and TN cross-shelf flux, are in phase, suggesting that an increase in the wind-stress leads to positive TN cross-shelf fluxes offshore by surface Ekman transport, which is an expected result in the western boundary of the Yucatan shelf.

---

## Author Response (AR3)

**Response to reviewer of "Budget of the total nitrogen in the Yucatan Shelf: driving mechanisms through a physical-biogeochemical coupled model"**

January 2020

We would like to thank the reviewer for this final suggestion regarding the title. We accept to change the title by: "DIN and PON budget in the Yucatan Shelf: driving mechanisms through a physical-biogeochemical coupled model".

The abstract and main text have been modified accordingly as you can see in the following tracking-changes document. The modifications to the original manuscript can be found in the following document where, as usual, new text is marked in blue and removed text in red.

Sheila Estrada-Allis on behalf of all co-authors.

**References**

**DIN and PON budget in the Yucatan Shelf: driving mechanisms through a physical-biogeochemical coupled model**

Sheila N. Estrada-Allis[1], Julio Sheinbaum[1], Joao M. Azevedo Correia de Souza[2], Cecilia Enríquez Ortiz[4], Ismael Mariño Tapia[3], and Jorge A. Herrera-Silveira[3]

[1]Physical Oceanography Department, CICESE, Ensenada, Baja California, Mexico
[2]MetOcean Solutions / MetService. Raglan. New Zealand.
[3]Departamento Recursos del Mar, CINVESTAV, Merida, Yucatan, Mexico.
[4]ENES-Merida, Facultad de Ciencias, Campus Yucatan, UNAM, Mexico.

**Correspondence:** Sheila N. Estrada-Allis (sheila@cicese.mx)

**Abstract.**

Continental shelves are the most productive areas in the seas with strongest implications for global  nitrogen cycling. The Yucatan  Shelf (YS) is the largest shelf in the Gulf of Mexico (GoM), however, its  nitrogen budget has not been quantified. This is largely due to the lack of significant spatio-temporal $in\ situ$ measurements

5 and the complexity of the shelf dynamics, including coastal upwelling, Coastal-Trapped Waves (CTWs) and influence of the Yucatan Current (YC) via bottom Ekman transport and dynamic uplift. In this paper,  we investigate and quantify the nitrogen budget of Dissolved Inorganic Nitrogen (DIN) and Particluate Organic Nitrogen (PON) in the YS using a nine-year output from a coupled physical-biogeochemical model of the GoM. The sum of DIN and PON is here referred to as Total Nitrogen (TN). Results indicate that the main entrance of  DIN

[revised manuscript text omitted]